# The climatology of Brewer-Dobson circulation and the contribution of gravity waves

Kaoru Sato[1], Soichiro Hirano[1]

[1]Department of Earth and Planetary Science, the University of Tokyo, Tokyo 113-0033, Japan

*Correspondence to*: Kaoru Sato (kaoru@eps.s.u-tokyo.ac.jp)

**Abstract.** The climatology of residual mean circulation—a main component of Brewer-Dobson circulation—and the potential contribution of gravity waves (GWs) are examined for the annual mean state and each season in the whole stratosphere based on the transformed-Eulerian mean zonal momentum equation using four modern reanalysis datasets. Resolved and unresolved waves in the datasets are respectively designated as Rossby waves and GWs, although resolved waves may contain some GWs.

First, the potential contribution of Rossby waves (RWs) to residual mean circulation is estimated from Eliassen-Palm flux divergence. The rest of residual-mean circulation, from which the potential RW contribution and zonal mean zonal wind tendency are subtracted, is examined as the potential GW contribution, assuming that the assimilation process assures sufficient accuracy of the three components used for this estimation. The GWs contribute to drive not only the summer hemispheric part of the winter deep branch and low-latitude part of shallow branches, as indicated by previous studies, but they also cause a

higher-latitude extension of the deep circulation in all seasons except for summer. This GW contribution is essential to determine the location of the turn-around latitude. The autumn circulation is stronger and wider than that of spring in the equinoctial seasons, regardless of almost symmetric RW and GW contributions around the equator. This asymmetry is attributable to the existence of the spring-to-autumn pole circulation corresponding to the angular momentum transport associated with seasonal variation due to the radiative process. The potential GW contribution is larger in September-to-

November than in March-to-May in both hemispheres. The upward mass flux is maximized in the boreal winter in the lower stratosphere, while it exhibits semi-annual variation in the upper stratosphere. The boreal winter maximum in the lower stratosphere is attributable to stronger RW activity in both hemispheres than in the austral winter. Plausible deficiencies of current GW parameterizations are discussed by comparing the potential GW contribution and the parameterized GW forcing.

## 1 Introduction

The meridional circulation in the middle atmosphere is an important component of the earth's climate, which globally transports minor constituents and causes adiabatic heating/cooling via the downwelling/upwelling. Part of the middle atmosphere has a thermal structure that is considerably different from the state of radiative equilibrium. The middle atmosphere circulation is mainly wave-driven. While gravity waves (GWs) are a primary driver of the mesospheric summer-to-winter-pole circulation, Rossby waves (RWs), including planetary waves and synoptic-scale waves, are most important for driving

the stratospheric circulation called Brewer-Dobson circulation (BDC). BDC consists of relatively slow residual-mean circulation driven by the wave forcing and rapid isentropic mixing with the turbulence associated with wave breaking and instability (Butchart, 2014). The residual mean circulation is divided into one deep and two shallow branches (e.g., Birner and Bönisch, 2011). The deep branch located in the winter middle and upper stratosphere is essentially driven by planetary waves and two shallow branches in the lower stratosphere of both hemispheres by synoptic-scale waves (e.g., Plumb, 2002). However, these descriptions are a rough sketch of BDC.

Recent advanced research tools, such as reanalysis datasets based on modern data-assimilation systems, have enabled the BDC structure to be examined in detail, and have highlighted the role of GW forcing even in the stratosphere (e.g., Butchart, 2014; Okamoto et al., 2011; Seviour et al., 2012). Iwasaki et al. (2009) made a comparison of the BDC diagnosed from multiple reanalysis using the mass-weighted isentropic zonal mean equations. It was shown that large difference is observed mainly in the low-latitude region. Miyazaki et al. (2016) examined the diffference in the BDC structure and eddy mixing between older reanalysis datasets (NCEP-NCAR, ERA-40, and JRA-25) and newer ones (NCEP-CFSR, ERA-Interim, and JRA-55), showing that the diagnosed BDCs from newer reanalyis datasets have similar structures unlike those from older reanalysis datasets. Such similarlity among the newer reanalysis datasets suggests that the assimilation technique of reanalysis is approaching its mature stage and the reanalysis datasets may withstand more detailed analysis of dynamics as performed in the present study.

Another useful tool for the analysis is the downward control principle derived by Haynes et al. (1991). This principle indicates that the Coriolis torque for the residual mean meridional flow is balanced with the wave forcing in a steady state. The contribution of each wave to the residual mean flow can be evaluated using this principle (McLandress and Shepherd, 2009). Okamoto et al. (2011) applied this method to the ERA-40 data and also to the outputs of a chemistry climate model (CCM). It was shown that the GW forcing significantly contributes to the formation of the summer hemispheric part of the deep branch of the winter circulation where RWs hardly propagate in the mean easterly wind of the summer stratosphere (Charney and Drazin, 1961), and to the formation of the shallow branches where orographic GWs break in the weak wind layer in the lower stratosphere (Lilly and Kennedy, 1973; Sato, 1990; Tanaka, 1986).

The upward mass flux is a quantity describing the strength of BDC. Previous studies showed that the upward mass flux exhibits an annual cycle with a maximum in the boreal winter (e.g., Randel et al., 2008). Seviour et al. (2012) used the ERA-Interim data and estimated the contribution of parameterized orographic GW forcing to the upward mass flux at 70 hPa associated with the residual mean circulation at ~4%, which is much smaller than the difference (~30%) between the total mass flux and the contribution of resolved wave forcing. They suggested the significant contribution of unresolved waves, such as non-orographic GWs whose parameterization is not included in ERA-Interim. Chun et al. (2011) used WACCM climatological simulation data and showed that GWs contribute to the upward mass flux by 17% at 70 hPa with comparable contributions by convective and orographic GWs. They estimated the contribution of GWs by taking the zonal mean zonal wind tendency in the zonal mean zonal momentum equation into consideration following Randel et al. (2008). Abalos et al. (2015) made a comprehensive study on BDC using three reanalysis datasets, ERA-Interim (Dee et al., 2011), NASA Modern Era Reanalysis for Research and Applications (MERRA) (Rienecker et al., 2011), and JRA-55 (Kobayashi et al., 2015), and

discussed tropical upwelling variation linked to the stratospheric quasi-biennial oscillation (QBO), El Niño–Southern Oscillation (ENSO), major sudden stratospheric warmings (SSW) and volcanic eruptions. They estimated upward mass fluxes by three different methods for the three reanalysis datasets and compared the results. The first method is a direct estimation using the definition of residual mean flow. The second is an indirect estimation using the zonal mean zonal momentum equation in which the Eliassen-Palm (EP) flux divergence, parameterized GW forcing, and zonal mean zonal wind tendency are given. The third one is an indirect estimation using the zonal mean thermodynamic equation in which the diabatic heating and zonal mean potential temperature tendency are given. They showed that the difference between the nine (i.e., three times three) estimates is large (about 40%). However, it was also reported that the relatively large discrepancy is mainly due to the difference in the method and not due to the difference in the reanalysis dataset.

Geller et al. (2013) compared absolute GW momentum fluxes expressed by the GW parameterization used in the climate models with those from high-resolution observations and from simulations using GW-resolving general circulation models (GCM) without GW parameterizations. The momentum fluxes expressed by the GW parameterizations are generally larger than satellite observations, and the satellite-derived fluxes fall off much more rapidly with height than the GW parameterizations. Geller et al. (2013) discussed that although the reason of these differences is not very clear, the observational filters of satellite measurements are one of plausible candidates to explain the differences: Satellites can observe only large horizontal-scale GWs, and the propagation characteristics of GWs may depend on the wave scales. Considering this observational filter effect, the GW parameterizations give roughly consistent momentum fluxes with the observations and GW-resolving high-resolution GCMs. However, the momentum fluxes by the parameterizations have significant deficiency in some notable regions which may not be explained only by the observational filters. For example, a significant momentum flux peak around 60°S in Austral winter, which is observed in satellite and GW-resolving GCM data, is missing in the GW parameterizations. This peak may be a key for the improvement of commonly-observed winter stratospheric cold bias in climate models (McLandress et al., 2012). Another notable deficiency of the GW parameterizations is that momentum fluxes are too large in winter and summer polar regions, which may affect the structure of the high latitude part of the zonal wind jets in the middle atmosphere.

Thus, the difference in results between the first and second methods obtained by Abalos et al. (2015) may be attributable to such deficiencies in the GW parameterization. Moreover, these previous studies discussed the structure and strength of BDC only in the lower stratosphere, and its structure in the middle and upper stratosphere has not yet been examined in detail.

The contribution of respective waves was examined for data from future projections by CCMs in a framework of the model intercomparison (Butchart et al., 2010). The results indicate that most CCMs project the acceleration of residual mean circulation in the stratosphere. Although the projected increase in the strength of the circulation did not significantly differ among the models, the ratio of the resolved and unresolved wave contributions largely depended on the model. As a plausible mechanism to explain this puzzling result, Cohen et al. (2013) showed the potential compensation of the parameterized GW forcing due to the barotropic and/or baroclinic instability in the model. Any excess of the parameterized GW forcing can be

adjusted by the instability processes, and hence the contribution of GW forcing in a projected climate is poorly estimated in the model. However, Rossby waves generated through the barotropic and/or baroclinic instability are really present in the middle atmosphere, and significantly contribute to the momentum budget particularly in the mesosphere and lower thermosphere region (Ern et al., 2013; McLandress et al., 2006; Sato and Nomoto, 2015; Sato et al., 2018). Zonal asymmetry of the GW forcing arising from GW sources (e.g., Ern et al., 2004; Hoffmann et al., 2013; Sacha et al., 2015; Sato et al., 2012; Wu et al., 2006) and/or from GW filtering in a large-scale flow modified by RWs (e.g., Smith, 2003) can modulate the RW field and give some impact on BDC (Šácha et al., 2016). In this way, RWs and GWs interplay in the momentum budget in the middle atmosphere. In addition, analysis on BDC in the past and present climate using reanalysis datasets may not considerably be affected by the artificial compensation problem indicated by Cohen et al. (2013, 2014), even if GW parameterizations are not perfect. This is because the analyzed dynamical fields, including resolved waves, tend to be realistic through modern assimilation with a large amount of observation data.

As already mentioned, the downward control principle is useful to estimate respective wave contributions. However, this method is not appropriate for the analysis of tropical regions because it assumes a balance between the Coriolis torque and wave forcing (i.e., the Coriolis parameter is not zero). In addition, differential radiative heating needs to be considered for tropical regions in solstitial seasons. The observed temperature in tropical regions is almost uniform latitudinally even in solstitial seasons where and when the latitudinal gradient of radiative heating by ozone is not negligible. This suggests the presence of thermally-driven circulation called the middle atmosphere Hadley circulation, which was first indicated and examined by Dunkerton (1989) and revisited by Semeniuk and Shepherd (2001). The middle atmosphere Hadley circulation is confined at latitudes lower than 30° and composed of a summer-to-winter hemisphere cell with an upward (downward) blanch in the summer (winter) hemisphere. This cell merges with the deep winter circulation formed by the westward forcing due to the RWs in the middle- and high-latitude regions. As for the wave contribution in the low-latitude region, Kerr-Munslow and Norton (2006) and Norton (2006) indicated that the equatorial RWs generated by strong tropical convection cause significant wave forcing in the off-equatorial region, and suggested that it has a large effect on the upwelling. However, the forcing by equatorial RWs cannot form the equatorward flow in the summer low-latitude region like the middle atmosphere Hadley circulation driven by differential radiative heating, because the forcing by dissipating RWs is westward. In contrast, the forcing associated with GW dissipation and/or breaking can be positive and cause the equatorward flow in the summer subtropical region, as suggested by Okamoto et al. (2011).

Another limitation of the analysis using the downward control principle is the assumption of a steady state. For this reason, the driving force of the residual mean circulation in the equinoctial seasons has not been examined in detail. For example, Seviour et al. (2012) showed the structure of the residual mean circulation in the equinoctial seasons but did not discuss it in details. According to their Fig. 3, even in the equinoctial seasons, the circulation is not symmetric around the equator in the stratosphere. It should be meaningful to elucidate the details on the physics of the circulation with such a structure. Particularly for the equinoctial seasons, the time change (tendency) of zonal mean zonal wind, which is ignored in the downward control principle analysis, needs to be considered in addition to the wave forcing in the zonal mean zonal momentum

equation. A potential method to overcome this issue is that proposed by Randel et al. (2008). They treated the tendency of zonal mean zonal wind as an additional term to the wave forcing for the estimation of the residual mean circulation [i.e., the second method of Abalos et al. (2015) as described above]. The present study will examine the tendency of zonal mean zonal wind with an expression of the stream function. This expression gives an angular momentum transport, which should be prevailing during a seasonal transition from the summer easterly wind to the winter westerly wind and vice versa in the middle atmosphere.

This paper focuses on three new aspects of the residual mean circulation in the stratosphere, which is a main part of BDC. One aspect is the climatological features of the potential GW contribution to the residual mean circulation in the whole stratosphere for the annual mean state and for each season. For this purpose, four modern reanalysis datasets over 30 years are analyzed. The climatological features are discussed in terms of the stream function structure and the upward mass flux. The interplay of RWs and GWs for the residual mean circulation is also highlighted. Particularly, the characteristics of potential GW contributions in equinoctial seasons are first shown by this study. We define them as "potential" because the wave forcing in the zonal momentum equation is not merely balanced with the Coriolis force for the residual mean meridional flow, but it also causes the acceleration of the zonal mean zonal wind.

Another new aspect upon which we focus is the climatological structure of the residual mean circulation in the middle and upper stratosphere, which have not yet been fully examined by previous studies, even for solstitial seasons when the steady assumption is generally valid. The analysis for this region has recently been feasible with the aid of the modern reanalysis datasets using high-top models in the assimilation system, like MERRA and MERRA Version 2 (MERRA-2) (Gelaro et al., 2017). The other new aspect is the mechanism of the asymmetric circulation around the equator observed in the equinoctial seasons.

This study is positioned as a part of the WCRP/SPARC S-RIP project. Thus, a comparison among the four reanalysis datasets itself is important. As the GWs are subgrid-scale phenomena in most models used for the reanalysis, and current GW parameterization schemes are not perfect, the GW contributions can be estimated only indirectly. Different reanalyses use different GW parameterizations, as described later. Thus, comparison between the indirect estimate of GW contribution and the parameterized GW forcing, and comparison of the estimates among the reanalysis datasets give useful insight into the future improvement of GW parameterizations.

An analysis is performed using four reanalysis datasets: MERRA, MERRA-2, ERA-Interim, and JRA-55. Descriptions are mainly made using MERRA-2 data because the model's top level 0.01 hPa of MERRA and MERRA-2 is higher than that of ERA-Interim and JRA55 (0.1 hPa), and because MERRA-2 is newer than MERRA. The analysis method and a brief description of analyzed datasets are given in Sect. 2. The assumption and limitations of the analysis method are also described. The characteristics of the annual mean and seasonal mean stream functions are shown, and the contributions of RWs and GWs are discussed in Sect. 3. The characteristics of seasonal variations in the upward mass flux are described and the contributions of RWs and GWs are discussed in Sect. 4. Section 5 discusses the seasonal variations in the potential GW contribution to the residual mean circulation by comparing the results by previous observational studies of GWs. In Sect. 6,

the indirectly estimated stream function due to real GW forcing and the stream functions due to parameterized GW forcing and due to assimilation increment are compared. Based on the result, plausible deficiencies of the GW parameterization schemes are discussed. Section 7 gives a summary and concluding remarks.

## 2 Method of analysis

We use the zonal mean zonal momentum equation in the transformed-Eulerian mean (TEM) equation system for the spherical coordinates (Andrews et al., 1987),

$$\frac{\partial \overline{u}}{\partial t} - \hat{f}\overline{v}^* + \overline{w}^* \frac{\partial \overline{u}}{\partial z} = \frac{1}{\rho_0 a \cos \phi} \nabla \cdot \boldsymbol{F} + \overline{GWF} + \overline{X}, \tag{1}$$

to evaluate the residual mean flow $(\overline{v}^*, \overline{w}^*)$, where $\boldsymbol{F}$ is the EP flux due to resolved waves, $\overline{GWF}$ is the forcing caused by subgrid-scale waves, and $\overline{X}$ is friction and/or viscosity;

$$\hat{f} \equiv f - \frac{1}{a \cos \phi} \frac{\partial (\overline{u} \cos \phi)}{\partial \phi} = 2\Omega \sin \phi - \frac{1}{a \cos \phi} \frac{\partial (\overline{u} \cos \phi)}{\partial \phi}; \tag{2}$$

$z$ is the log pressure height, and $\phi$ is the latitude. The sum of the first and second terms in the right side of Eq. (1) is referred to as the wave forcing.

The meridional $(\overline{v}^*)$ and vertical $(\overline{w}^*)$ components of the residual mean flow are respectively defined as

$$\overline{v}^* \equiv \overline{v} - \frac{1}{\rho_0} \left( \rho_0 \frac{\overline{v'\theta'}}{\theta_{0z}} \right)_z \quad \text{and} \quad \overline{w}^* \equiv \overline{w} + \frac{1}{a \cos \phi} \left( \cos \phi \frac{\overline{v'\theta'}}{\theta_{0z}} \right)_\phi. \tag{3}$$

See Andrews et al. (1987) for the formulae for $\boldsymbol{F}$ [their equation (3.5.3)]. Other notations throughout in this work except for those defined explicitly are standard, following Andrews et al. (1987).

The residual-mean flow is a good approximate of the Lagrangian mean flow (i.e., the sum of Eulerian mean flow plus the first quadratic term of Stokes drift) according to the small-amplitude theory. From the continuity equation, a stream function $\Psi$ of the residual mean flow is defined as

$$\overline{v}^* \equiv -\frac{1}{\rho_0 \cos \phi} \Psi_z, \quad \text{and} \quad \overline{w}^* \equiv \frac{1}{\rho_0 a \cos \phi} \Psi_\phi. \tag{4}$$

Thus, there are two methods to estimate $\Psi(\phi, z)$ directly from Eq. (4): One is an integration of $\overline{v}^*$ in the vertical with a top boundary condition of $\Psi = 0$. The other is a latitudinal integration of $\overline{w}^*$ with a boundary condition of $\Psi = 0$ at the North Pole or the South Pole. In this study, $\Psi(\phi, z)$ in the Northern Hemisphere (NH) [Southern Hemisphere (SH)] by the latitudinal integration of $\overline{w}^*$ starting $\Psi = 0$ at the North [South] Pole:

$$\Psi(\phi) = -\int_\phi^{\frac{\pi}{2}} \overline{w}^* d\phi' \quad \text{for the NH and} \quad \Psi(\phi) = \int_{-\frac{\pi}{2}}^{\phi} \overline{w}^* d\phi' \quad \text{for the SH.} \tag{5}$$

The comparison of the two methods is discussed in Appendix A. Hereafter, $\Psi$ is called the total stream function to distinguish from the stream functions of wave contributions.

For GW-resolving GCM outputs, the first term in the right side of Eq. (1) includes resolved GWs as well as RWs (e.g., Watanabe et al., 2008). For reanalysis datasets with a relatively coarse grid used in the present study, the first term is primarily due to RWs except for the equatorial region where waves other than RWs, such as Kelvin waves, Rossby-gravity waves, and large-scale inertia-gravity waves are present. Because the calculation for the analysis in the present study is mainly performed for the off-equatorial region, most resolved waves can be regarded as RWs. Assuming that RWs are realistically expressed in the reanalysis datasets and that the grid spacing of the reanalysis datasets is still coarse to express GWs, the resolved (unresolved) waves in the datasets are designated as RWs (GWs), although resolved waves may contain some GWs. Thus, the EP flux divergence, $\nabla \cdot \boldsymbol{F}$, directly calculated using the reanalysis data, is regarded as the RW forcing, and the forcing due to subgrid-scale waves $\overline{GWF}$ is regarded as the GW forcing. In the reanalysis data, a theoretical equation of the momentum conservation Eq. (1) may not be held due to the data assimilation processes. However, advanced data assimilation techniques, such as the four-dimensional variational method (4D-VAR) used for JRA-55 and ERA-Interim, assimilate observation data at the exact time so that the dynamical balance would be maintained (Miyazaki et al., 2016). Theoretically speaking, the term $\overline{GWF}$ in Eq. (1) represents not the parameterized GW forcing but the real GW forcing. The term $\overline{GWF}$ for reanalysis data should be a sum of the parameterized GW forcing and the GW forcing that is not expressed by the GW parameterization. The latter is likely included in the assimilation increment, if the assimilation works to correct for the limitations of GW parameterizations.

The contribution of each term in Eq. (1) to the total stream function is evaluated as follows. First, substitution of Eq. (4) into Eq. (1) yields

$$\frac{\partial(\Psi, \overline{m})}{\partial(\phi, z)} = \left( \frac{1}{\rho_0 a \cos \phi} \nabla \cdot \boldsymbol{F} + \overline{GWF} + \overline{X} - \frac{\partial \overline{u}}{\partial t} \right) \rho_0 a^2 \cos^2 \phi, \tag{5}$$

where $\overline{m} = a \cos \phi \, (\overline{u} + a\Omega \cos \phi)$ is the zonal mean angular momentum per unit mass (Haynes et al., 1991; Randel et al., 2002). Using Eq. (5), $\Psi(y, z)$ is expressed as a sum of three components:

$$\Psi(\phi, z) = \Psi_{\mathrm{RW}}(\phi, z) + \Psi_{\mathrm{GW}}(\phi, z) + \Psi_{\mathrm{X}}(\phi, z) + \Psi_{\mathrm{dU/dt}}(\phi, z) \tag{6}$$

where

$$\Psi_{\mathrm{RW}}(\phi, z) \equiv - \int_z^\infty \left[ \frac{\nabla \cdot \boldsymbol{F}}{a\hat{f}} \right]_{\overline{m}} d\zeta, \tag{7}$$

$$\Psi_{\mathrm{GW}}(\phi, z) \equiv - \cos \phi \int_z^\infty \left[ \frac{\rho_0}{\hat{f}} \overline{GWF} \right]_{\overline{m}} d\zeta, \tag{8}$$

$$\Psi_{\mathrm{X}}(\phi, z) \equiv - \cos \phi \int_z^\infty \left[ \frac{\rho_0}{\hat{f}} \overline{X} \right]_{\overline{m}} d\zeta, \tag{9}$$

$$\Psi_{\mathrm{dU/dt}}(\phi, z) \equiv \cos \phi \int_z^\infty \left[ \frac{\rho_0}{\hat{f}} \frac{\partial \overline{u}}{\partial t} \right]_{\overline{m}} d\zeta, \tag{10}$$

and $\int_z [\quad]_{\overline{m}} d\zeta$ means a vertical integration along a constant $\overline{m}$. With this vertical integration instead of that along a constant $\phi$, the vertical advection of zonal wind $\overline{w}^* \frac{\partial \overline{u}}{\partial z}$ in Eq. (1) is included for the estimation. In this study, $\Psi_{RW}(\phi, z)$ and $\Psi_{GW}(\phi, z)$ are respectively called potential RW and GW contributions to the residual mean flow. We used the "potential" contribution because the wave forcings drive the residual mean flow, but a part of them causes acceleration/deceleration of $\overline{u}$ [i.e., $\partial \overline{u}/\partial t$ in Eq. (1)]. The distribution of the wave forcing to the Coriolis term $-\hat{f}\overline{v}^*$ and the tendency term $\partial \overline{u}/\partial t$ depends on the aspect ratio of the forcing in the meridional cross section soon after the forcing is given (Garcia, 1987; Hayashi and Sato, 2018). The part of the stream function driven by the zonal mean zonal wind tendency is expressed as $\Psi_{dU/dt}(\phi, z)$. The $\Psi_{GW}(\phi, z)$ cannot be directly calculated because of the unknown $\overline{GWF}$: It should be noted that GW parameterizations are not perfect. It was shown by Geller et al. (2013) that parameterized GWs have large discrepancy in the latitudinal profile of their momentum fluxes from those observed and simulated by GW-resolving GCMs. This deficiency may cause cold bias and late final warming in the SH stratosphere (e.g., McLandress et al., 2012), and too weak easterly in the summer middle atmosphere in the GCMs (E. Manzini, private communication). Thus, in our study, $\Psi_{GW}(\phi, z)$ is indirectly estimated using the following formula,

$$\Psi_{GW}(\phi, z) = \Psi(\phi, z) - \Psi_{RW}(\phi, z) - \Psi_{dU/dt}(\phi, z), \tag{11}$$

which is derived from Eq. (1) ignoring the term $\overline{X}$.

The working hypothesis when applying this method to the reanalysis datasets is that three terms in the right hand side of Eq. (11) are accurately estimated owing to the data assimilation. Thus, we do not assume that reanalysis data satisfies the zonal momentum equation. In other words, we assume that most of the assimilation increment is acting to correct for the limitations of GW parameterizations and the reanalysis provides realistic dynamical fields including ageostrophic motions $(\overline{v}, \overline{w})$ appearing in the first term in the right hand side of Eq. (11) [see Eq. (3)]. Under the assumption, the momentum equation described as Eq. (11) can be interpreted as the contribution of the 'actual' GW forcing to the stream function of the residual mean flow. In general, it is quite difficult to validate this hypothesis directly. However, the similarity among $\Psi_{GW}(\phi, z)$ estimated using Eq. (11) from the four reanalysis datasets, if any, may show real dynamics in the atmosphere (i.e., potential GW contribution). The features consistent with observational and/or theoretical knowledge, if any, will give indirect evidence of the validity of the assumption. This study will mainly discuss such observationally and dynamically consistent features which are commonly observed in the reanalysis data.

In this study, the integrations in Eqs. (7) and (10) were performed faithfully along the angular momentum $(\overline{m})$ contour in the vertical because the contribution of GW forcing may be relatively small. Hence, the uncertainty should be reduced as much as possible, although a few previous studies performed an approximated integration at a constant $\phi$ in the vertical (McLandress and Shepherd, 2009; Okamoto et al., 2011). As $\hat{f}$ is quite small near the equator, the stream functions of Eq. (7), Eq. (10), and Eq. (11) are obtained for $|\phi| > 20°$. It is worth noting here that even large horizontal-scale phenomena may not be well represented at low latitudes in the reanalysis data because they are not balanced with well-observed quantities such as temperature due to small $f$. In fact, there is large discrepancy in horizontal winds in the equatorial stratosphere among

reanalysis datasets (Kawatani et al., 2016; Podglajen et al., 2014). It is also discussed by Kim and Chun (2015) that amplitudes of large horizontal-scale equatorial waves may be underestimated because the vertical grid spacing of the model is too coarse to resolve short vertical wavelengths that the equatorial waves may have in the strong vertical shear of the mean zonal wind of the QBO. In contrast, at higher latitudes, for which our calculation was performed, resolved fields are primarily balanced with well-observed quantities and hence they are probably realistic.

Under the steady state assumption, which is valid for the annual mean and approximately valid for the solstitial seasons, Eq. (7) is reduced to the downward control principle by Haynes et al. (1991) (Randel et al., 2002). In this case, $\Psi_{RW}(\phi, z)$ and $\Psi_{GW}(\phi, z)$ estimated using (11) with $\Psi_{dU/dt}(\phi, z) = 0$ are exact contributions by RWs and GWs (McLandress and Shepherd, 2009).

The zonal mean zonal wind tendency $\partial \overline{u}/\partial t$ is large in the equinoctial seasons because of the seasonal change in the radiative heating. As the seasonal time scale is much longer than a typical radiative relaxation time in the stratosphere, the wave forcing hardly causes $\partial \overline{u}/\partial t$ and is almost balanced with a part of $-\widehat{f}\overline{v}^*$ except for the equatorial region where the Coriolis parameter $f$ is quite small. Thus, the zonal mean zonal wind tendency term $\partial \overline{u}/\partial t$ can be considered mainly due to the radiation effect, which should be balanced with the Coriolis force for the residual mean flow similar to the wave forcing. In this study, $\Psi_{RW}(\phi, z)$ and $\Psi_{GW}(\phi, z)$ will mainly be discussed as respective wave contributions to the residual mean circulation, and the potential contributions of RWs and GWs to $\partial \overline{u}/\partial t$ will also be noted.

Next, the method of upward mass flux is described. In the steady state, the amount of upward mass flux $F^{\uparrow}(z)$ should be balanced with the sum of downward mass fluxes in the NH ($F_{NH}^{\downarrow}$ ) and SH ($F_{SH}^{\downarrow}$ ):

$$F^{\uparrow}(z) = -[F_{NH}^{\downarrow}(z) + F_{SH}^{\downarrow}(z)], \qquad (12)$$

$$F_{NH}^{\downarrow}(z) = 2\pi a^2 \rho_0 \int_{\phi_{TL}^{NH}}^{\frac{\pi}{2}} \overline{w}^*(z) \cos\phi \, d\phi = 2\pi a \Psi(\phi_{TL}^{NH}, z), \qquad (13)$$

$$F_{SH}^{\downarrow}(z) = 2\pi a^2 \rho_0 \int_{-\frac{\pi}{2}}^{\phi_{TL}^{SH}} \overline{w}^*(z) \cos\phi \, d\phi = -2\pi a \Psi(\phi_{TL}^{SH}, z), \qquad (14)$$

where $\phi_{TL}^{NH}$ and $\phi_{TL}^{SH}$ are the turn-around latitudes where $\overline{w}^* = 0$ for the NH and SH circulations at each altitude, respectively. Eqs. (12)–(14) indicate that the total upward mass flux and the contributions by the NH and SH are estimated only using stream function values at the turn-around latitudes. Using $\Psi_{RW}(\phi, z)$ and $\Psi_{GW}(\phi, z)$ in place of $\Psi(\phi, z)$, the RW and GW contributions to the upward mass flux are estimated, respectively. In our study, the turn-around latitude used for calculation of each wave contribution is taken the same used for the total upward mass flux. For equinoctial seasons when the steady state assumption does not hold, this method only estimates the potential contributions by the RWs and GWs.

Four reanalysis datasets of MERRA-2, MERRA, JRA-55, and ERA-Interim over 30 years from 1986–2015 are used to examine the climatology of the residual mean circulation in the whole stratosphere as the main part of BDC. Although the horizontal resolutions of the model used for the data assimilation are different (Fujiwara et al., 2017), the output grid intervals are almost the same for the four reanalysis datasets ($1.25° \times 1.25°$ for MERRA, MERRA-2, and JRA-55, and $1.5° \times 1.2°$ for

ERA-Interim). Thus, the horizontal wavenumber range of "resolved waves" examined in the present study is almost the same for all reanalysis datasets. The number of pressure levels of the reanalysis forecast model is 72 for MERRA and MERRA-2 and 60 for ERA-Interim and JRA-55. The top of the model is 0.01 hPa for MERRA and MERRA-2 and 0.1 hPa for ERA-Interim and JRA-55. Features for the annual mean state and four seasons of December to February (DJF), March to May (MAM), June to August (JJA), and September to November (SON) are analyzed.

## 3 Results

Before the details of the circulation for the annual mean state and each season are discussed, the meridional cross sections of the zonal mean zonal wind climatology are shown in Fig. 1, as both RW and GW propagations strongly depend on the mean wind. Since the difference in the stratospheric mean wind is not large among the reanalysis datasets, and the detailed comparison of the mean wind itself is beyond the scope of this study, only the field from MERRA-2, which covers the region up to the highest level, is shown. As is well known, the winter westerly jet is stronger in the SH (JJA) than in the NH (DJF). In spring, the westerly jet is strong and has a peak in the lower stratosphere in the SH (SON), while the westerly jet almost disappears in the NH (MAM). These differences in the westerly jet between the two hemispheres are considered the result of the different activity of RWs generated in the troposphere. Another interesting difference is the strength of the summer easterly jet, which is stronger in the SH (DJF) than in the NH (JJA). This feature is not very well known, but it could be valuable to examine the cause in future studies.

### 3.1 Annual mean structure of the stream functions

Figure 2 shows the latitude-height sections of annual mean values of $\Psi(\phi, z)$, $\Psi_{RW}(\phi, z)$, and $\Psi_{GW}(\phi, z)$ for all the reanalysis datasets. There are many notable, interesting, and important characteristics commonly observed in all datasets. Here and in subsequent sections, first, the characteristics observed in the new reanalysis MERRA-2 covering the wide height region are discussed, and next, similarity and differences among the four datasets are described.

In MERRA-2, two-celled circulation is clearly observed for the annual-mean total stream function $\Psi(\phi, z)$, which is directly estimated using Eqs. (3) and (4), in Fig. 2a. The $\Psi(\phi, z)$ in the NH has slightly larger magnitudes than in the SH in most stratosphere below 2 hPa. This feature is consistent with stronger planetary-scale RW activity in the NH (Fig. 2b). In fact, the two-celled circulation in $\Psi(\phi, z)$ is mainly determined by the RW contribution, $\Psi_{RW}(\phi, z)$. However, the GW contribution, $\Psi_{GW}(\phi, z)$, is also important in some notable regions (Fig. 2c) as described in the following.

The GW contribution is almost symmetric around the equator with a slight hemispheric difference. The GWs contribute largely to the poleward circulation [i.e., clockwise (counter-clockwise) circulation in the NH (SH)] in the middle- and high-latitude regions of the whole stratosphere. This circulation should be caused by the westward forcing due to GWs likely originating from the topography and jet-front system in the troposphere (e.g., Hertzog et al., 2008; Sato et al., 2009). The magnitude of $\Psi_{GW}(\phi, z)$ in the poleward circulation is slightly larger in the SH than in the NH.

In addition, a characteristic equatorward circulation [i.e., counter-clockwise (clockwise) circulation in the NH (SH)] is observed in the low-latitude region in $\Psi_{GW}(\phi, z)$ whose largest latitude extends to 30° at 10 hPa. This equatorward circulation is caused by the eastward forcing due to GWs, which likely originate from vigorous convection in the subtropical region as shown theoretically and numerically by previous studies (e.g., Pfister et al., 1993; Sato et al., 2009). It is also worth noting that the turn-around latitude of the poleward circulation for $\Psi_{GW}(\phi, z)$ is observed at approximately 40–55° depending on the altitude, which is higher than that for $\Psi_{RW}(\phi, z)$. This means that the GW forcing can modify the turn-around latitude of the BDC, as discussed in detail later.

It is important that the characteristics of $\Psi(\phi, z)$, $\Psi_{RW}(\phi, z)$, and $\Psi_{GW}(\phi, z)$ described above are commonly observed in all reanalysis datasets. However it may be worth noting a few slight differences. The equatorward circulation is commonly observed in the low-latitude region for $\Psi_{GW}(\phi, z)$. Note that the equatorward circulation is not very clear for JRA-55 in the displayed latitude range, but it exists at slightly lower latitudes than 20° (not shown). The circulation extends down to 100 hPa for MERRA and MERRA-2. However, the lower end of the circulation is located at 20–30 hPa for ERA-Interim and for JRA-55. Instead, for ERA-Interim and JRA-55, the $\Psi_{GW}(\phi, z)$ exhibits strong poleward circulation below 30 hPa in the low- and middle-latitude regions. Similar strong poleward circulation is observed only in the middle-latitude region for MERRA and MERRA-2. This poleward circulation probably reflects the orographic GW forcing enhanced in the weak wind layer above the subtropical jet (Lilly and Kennedy, 1973; Sato, 1990; Tanaka, 1986).

The feature in $\Psi_{RW}(\phi, z)$ that it is almost symmetric around the equator and slightly stronger in the NH is also commonly observed in all reanalysis datasets. A small but interesting difference in $\Psi_{RW}(\phi, z)$ is the depth of the circulation: it is deeper for MERRA/MERRA-2 than for ERA-Interim/JRA-55. One plausible reason for this is the difference in the top of the model used for the data assimilation (0.01 hPa for MERRA/MERRA-2 and 0.1 hPa for ERA-Interim/JRA-55) and hence the data top (0.1 hPa for MERRA/MERRA-2 and 1 hPa for ERA-Interim/JRA-55). Thus, the top of the vertical integration in Eq. (7) depends on the reanalysis dataset, and the underestimation of $\Psi_{RW}(\phi, z)$ by ignoring the RW forcing above the data top can be greater for ERA-Interim/JRA-55 than that for MERRA/MERRA-2. This inference is supported by the fact that $\Psi_{RW}(\phi, z)$ calculated without using data above 1 hPa for MERRA/MERRA-2 exhibits circulation with similar depth to that for ERA-Interim/JRA-55, although the structure below 10 hPa does not largely depend on the data top (not shown). This result means that the RW forcing in the upper stratosphere and mesosphere is not negligible in the upper stratospheric circulation.

There are other potential elements causing these slight differences in the stream function among the reanalysis datasets. One is the GW parameterizations used the assimilation system: The model for ERA-Interim and JRA-55 use only orographic GW parameterization, while both orographic and non-orographic GW parameterizations are used for MERRA and MERRA-2 (Fujiwara et al., 2017). In addition, Rayleigh friction is included in the upper model part which roughly mimics the forcing by non-orographic GWs for ERA-Interim and JRA-55. Note that the data provided as GW zonal mean acceleration for JRA-55 includes Rayleigh friction as well as GW forcing from orographic GW parameterization. Any difference caused by the parameterized GW forcing should be corrected by the increment given by the data assimilation system. However, the observation data used for the data assimilation are not sufficient, and the correction may not be perfect. The other element is

the assimilation method, which is the 4d-Var for ERA-Interim and JRA-55 and the 3d-Var for MERRA and MERRA-2. A detailed investigation on the reasons for the differences in the stream function among the four analysis datasets is beyond the scope of this paper and left open for future studies.

Next, the annual mean $\Psi(\phi, z)$, $\Psi_{RW}(\phi, z)$, and $\Psi_{GW}(\phi, z)$ are more closely examined as a function of the latitude, focusing on three levels: 70 hPa, 10 hPa, and 3hPa in Figs. 3, 4, and 5, respectively. The positive and negative maxima in $\Psi(\phi)$ (black curves) corresponding to the turn-around latitudes are almost the same for all reanalysis datasets for 70 hPa that are at $\phi_{TL}^{NH} = \sim 35^oN$ and at $\phi_{TL}^{SH} = \sim 30^oS$. In addition, the magnitudes of $\Psi_{RW}(\phi)$ (blue curves) are almost the same for all reanalysis datasets. It is important and interesting that $\Psi_{RW}(\phi)$ is flat and does not have clear peaks near the turn-around latitudes of $\Psi(\phi)$, although RW is considered a primary driver of BDC. Instead, the turn-around latitudes of $\Psi(\phi)$ are mainly determined by the shape of $\Psi_{GW}(\phi)$ (red curves).

The importance of GWs is also the case for 10 hPa (Fig. 4). The turn-around latitudes of $\Psi(\phi)$ at 10 hPa are located at $\phi_{TL}^{NH} = 30^oN$ and at $\phi_{TL}^{SH} = 35^oS$. The $\Psi_{RW}(\phi)$ has the maxima but at lower latitudes (25$^o$N and 25$^o$S) than $\phi_{TL}^{NH}$ and $\phi_{TL}^{SH}$ for all reanalysis datasets, although the magnitude depends on the dataset. The sharp increase with the latitude in $\Psi_{GW}(\phi)$ observed up to 50$^o$ largely contributes to determining the location of the turn-around latitudes. Therefore, it is considered that the determination of the turn-around latitudes is an important role of GWs in the annual mean residual circulation.

These features in the latitudinal profiles of the stream functions at 70 hPa and 10 hPa are commonly observed in all reanalysis datasets with some quantitative differences. However, the difference in the magnitude and shape of $\Psi(\phi, z)$, $\Psi_{RW}(\phi, z)$, and $\Psi_{GW}(\phi, z)$ among the four reanalysis datasets is much larger at 3 hPa (Fig. 5) than at lower levels, although a similar GW contribution to the location of the turn-around latitudes is observed at this level, as well. The difference among the datasets is again likely due to the limitations of the data assimilation because of model performance and/or insufficient observation data. Thus, a further detailed description is not provided for 3 hPa.

**3.2 Stream functions in solstitial seasons**

Figure 6 (Fig. 7) shows the climatology of $\Psi(\phi, z)$, $\Psi_{RW}(\phi, z)$, $\Psi_{GW}(\phi, z)$, and $\Psi_{dU/dt}(\phi, z)$ for DJF (JJA) obtained by each reanalysis dataset. The winter circulation in $\Psi(\phi, z)$ is deep and stronger, and it extends to the summer hemisphere while the summer circulation is strong only in the lower and middle stratosphere, as is well known.

It is seen from the comparison among $\Psi(\phi, z)$, $\Psi_{RW}(\phi, z)$ and $\Psi_{GW}(\phi, z)$ for MERRA-2 (Figs. 6a–6d) that the major part of $\Psi(\phi, z)$ is attributed to the RW forcing. However, the GW contribution is also large: The GWs contribute to the formation of the summer hemispheric part of the winter circulation, as indicated by Okamoto et al. (2011). In particular, the upper stratospheric part in the whole summer hemisphere is mainly determined by the GWs. It is interesting that the GW contribution in the summer upper stratosphere in the NH and that in the SH are comparable. Thus, the GW forcing in the region analyzed in the stratosphere may not be responsible for the significant difference in the mean easterly wind in summer between

the NH (JJA) and SH (DJF) (Figs. 1c and 1a), as indicated earlier. Another notable feature is that the extension of the winter circulation to the high latitudes is largely contributed to by the GW forcing. This feature is clearer for the SH (JJA) where the $\Psi_{RW}(\phi, z)$ values are quite small or almost zero in the middle and upper stratosphere.

The poleward circulation in the summer hemisphere is deeper and stronger in the SH (DJF) than in the NH (JJA). This hemispheric difference is mainly due to larger RW contribution in the SH. This is consistent with the feature observed in the mean wind, in which a relatively strong westerly mean wind remains in the lower stratosphere in the SH (DJF) (Fig. 1a). This westerly mean wind allows RWs from the troposphere to reach the lower stratosphere.

Compared with $\Psi(\phi, z)$, $\Psi_{dU/dt}(\phi, z)$ for the solstitial seasons is quite small except for summer low latitudes. This fact ensures the validity of the steady state assumption for solstitial seasons, which are frequently made for the diagnostics using the downward control principle (e.g., McLandress and Shepherd, 2009). It is interesting that the magnitude of $\Psi_{dU/dt}(\phi, z)$ in the summer low-latitude region is comparable to that of $\Psi_{GW}(\phi, z)$ but confined in the lower stratosphere. The direction and latitudinal location of this circulation are consistent with the middle atmosphere Hadley circulation, although dominant altitude region may be slightly lower than the theoretical expectation (i.e., upper stratosphere) (Semeniuk and Shepherd, 2001). It is also worth noting that there is also a weak equatorward circulation in $\Psi_{dU/dt}(\phi, z)$ in the winter hemisphere located in the middle-latitude region in the NH (DJF) and at relatively low latitudes in the SH (JJA). Equatorward circulation in $\Psi_{dU/dt}(\phi, z)$ means westerly wind weakening. Thus, these equatorward circulations in $\Psi_{dU/dt}(\phi, z)$ can be at least partly due to the strong westward RW forcing in the winter stratosphere and summer lower stratosphere. The difference in the dominant-latitude region of $\Psi_{dU/dt}(\phi, z)$ for the winter season between the two hemispheres is consistent with this inference.

The overall characteristics of the stream functions, including the potential GW contribution in solstitial seasons described above for MERRA-2, are similarly observed in other reanalysis datasets. However, there are some minor differences among the datasets. The poleward circulation in $\Psi(\phi, z)$ in summer is deeper in MERRA and MERRA-2 than in ERA-Interim and JRA-55. Equatorward circulation in the winter low-latitude region is observed in $\Psi_{GW}(\phi, z)$ in MERRA and MERRA-2, while it is not for the other datasets. A similar discussion for the annual mean climatology in Sect. 3.1 would be made for these differences for the solstitial seasons.

## 3.3 Stream functions in equinoctial seasons

The zonal mean zonal wind tendency is large due to a seasonal change in the radiative heating in the equinoctial seasons. Thus, roughly speaking, $\Psi_{dU/dt}(\phi, z)$ is primarily attributable to the radiation in the equinoctial seasons and the acceleration/deceleration by the wave forcing is secondary. Figure 8 (Fig. 9) shows the climatology of $\Psi(\phi, z)$, $\Psi_{RW}(\phi, z)$, $\Psi_{GW}(\phi, z)$, and $\Psi_{dU/dt}(\phi, z)$ for MAM (SON).

The most interesting feature is that $\Psi(\phi, z)$ is not symmetric around the equator (Figs. 8a and 9a) regardless of the equinoctial seasons. The circulation structure rather resembles that in the subsequent solstitial season. The autumn circulation

is stronger and latitudinally wider than the spring circulation. In contrast, $\Psi_{RW}(\phi, z)$ (Figs. 8b and 9b) and $\Psi_{GW}(\phi, z)$ (Figs. 8c and 9c) are almost symmetric around the equator, similar to the annual mean circulations (Figs. 2b and 2c), although the strength is slightly different. The anti-symmetry around the equator observed in $\Psi(\phi, z)$ is attributable to the structure of $\Psi_{dU/dt}(\phi, z)$. The circulation in $\Psi_{dU/dt}(\phi, z)$ is globally southward (northward) in MAM (SON)—in other words, from the

spring pole to the autumn pole. This is consistent with the angular momentum conservation for the easterly (westerly) jet formation in the spring (autumn) hemisphere.

       Except for $\Psi_{GW}(\phi, z)$ in the low-latitude region, most of the $\Psi_{RW}(\phi, z)$, $\Psi_{GW}(\phi, z)$, and $\Psi_{dU/dt}(\phi, z)$ values have the same sign in the autumn hemisphere, while $\Psi_{dU/dt}(\phi, z)$ values have the opposite sign to those of $\Psi_{RW}(\phi, z)$ and $\Psi_{GW}(\phi, z)$ in the spring hemisphere. The difference in the magnitudes of $\Psi_{RW}(\phi, z)$ and $\Psi_{GW}(\phi, z)$ between the spring and

autumn hemispheres is not large compared with that between the two hemispheres in the solstitial seasons, as already mentioned. Therefore, it is inferred that the stronger circulation expanding over a wider latitudinal region in the autumn hemisphere than in the spring one is mainly due to seasonal change in radiative heating.

       Next, detailed contributions by RWs [$\Psi_{RW}(\phi, z)$] and GWs [$\Psi_{GW}(\phi, z)$] to the total circulation [$\Psi(\phi, z)$] are discussed. In the lower stratosphere, the RW contribution [$\Psi_{RW}(\phi, z)$] is large, and its magnitude is comparable to the

contribution by zonal mean zonal wind tendency [$\Psi_{dU/dt}(\phi, z)$] which is mainly due to a radiation effect, whereas the GW contribution $\Psi_{GW}(\phi, z)$ is not negligible in the low-latitude region of the lowermost stratosphere. In the upper stratosphere, the GW contribution [$\Psi_{GW}(\phi, z)$] is rather dominant, particularly at the middle- and high-latitude regions where $\Psi_{RW}(\phi, z)$ is weak. Thus, the GW forcing is likely important to determine the turn-around latitudes and depth of the residual mean circulation in the equinoctial seasons, similar to those for the annual mean circulation.

Another important feature in $\Psi(\phi, z)$ is that the spring circulation is stronger in the SH (SON) than in the NH (MAM), although the autumn circulation does not differ much. This is mainly attributed to the stronger $\Psi_{RW}(\phi, z)$ in the SH (SON). It is interesting to note that in spring, $\Psi_{GW}(\phi, z)$ is also stronger in the SH (SON) than in the NH (MAM), while it is comparable in autumn. The wave forcing is not simply balanced with the Coriolis force but partly accelerates the mean zonal wind in equinoctial seasons when the steady state assumption does not hold (e.g., Garcia, 1987; Hayashi and Sato, 2018). Thus, these

stronger RW and GW forcings are consistent with the more distorted structure of $\Psi_{dU/dt}(\phi, z)$ in the spring hemisphere than in the autumn one. It is inferred that the larger distortion in $\Psi_{dU/dt}(\phi, z)$ in spring in the SH (Fig. 9d) is also a reflection of the stronger RW activity than that in NH.

       These characteristics of the equinoctial seasons—in terms of the structure and contribution by wave forcing and radiative heating—are similarly observed in the other reanalysis datasets, although there are some slight differences as

indicated for the annual mean circulation in Sect. 3.1.

## 4. Seasonal variation of the upward mass flux

The total upward mass flux was estimated using Eqs. (12)–(14) for each month. In addition, contributions by RWs, GWs, and the zonal wind tendency are respectively calculated by replacing $\Psi(\phi, z)$ with $\Psi_{RW}(\phi, z)$, $\Psi_{GW}(\phi, z)$, and $\Psi_{dU/dt}(\phi, z)$ at the same turn-around latitudes determined by $\Psi(\phi, z)$. Figure 10 shows the results for 70 hPa for all reanalysis datasets. Note again that $\Psi_{RW}(\phi, z)$ and $\Psi_{GW}(\phi, z)$ give rough estimates because a part of RW and GW forcings is used to accelerate the zonal wind instead of driving the meridional circulation, and because of the limitations of data assimilation, as discussed in Sect. 2.

The total upward mass flux is maximized in December and January (i.e., boreal winter) and minimized in June and July (i.e., austral winter). The boreal winter maximum is reflected by two features: First, the mass flux associated with the winter circulation is larger in the NH than in the SH, as is consistent with higher activity of planetary-scale RWs in the NH. Second, the mass flux associated with the summer (i.e., boreal winter) circulation in the SH is larger than that in the NH. The latter is attributable to the mean zonal wind, which is westerly up to 30 hPa at the middle- and high-latitude regions of the SH, satisfying the condition of possible upward propagation of planetary waves even in the summer season. These features are commonly seen and quantitatively consistent for all reanalysis datasets.

The boreal winter maximum of the total upward mass flux was examined by Kim et al. (2016) for 100 hPa. Based on the spectral analysis, it was shown that the maximum was attributed to planetary waves with zonal wavenumber 3 originating from NH extratropics and SH tropics. According to their analysis, EP flux divergence due to the s=3 waves is dominant only below 70 hPa. Thus, the DJF maximum observed at 70 hPa shown in Fig. 10 is likely due to RWs with different wavenumbers.

The sum of RWs and zonal wind tendency roughly explains the total mass flux. The contribution of the zonal wind tendency is large in the autumn of each hemisphere, which is consistent with the characteristic structure of $\Psi_{dU/dt}(\phi, z)$, as discussed in Sect. 3.3. While Abalos et al. (2014) discussed that the zonal mean zonal wind tendency as well as meridional circulation caused by wave forcing largely contribute to subseasonal variability of the upward mass flux, Kim et al. (2016) stated that the zonal mean zonal wind tendency contribution is negligible for seasonal time scales. According to our analysis for 70hPa, the zonal mean zonal wind tendency contribution to the total upward mass flux takes its broad maximum of approximately 15% in MAM for all reanalysis datasets. The zonal wind tendency contributes negatively by 10% or less in July and December. Thus, it seems that the zonal wind tendency contribution is not negligible for 70 hPa in several months.

The percentage of the GW contribution to the mass flux largely depends on the reanalysis dataset. The contribution of the GWs to the mass flux is ~20% at 70 hPa for MERRA and MERRA-2 at the most, while it is ~35–40% for ERA-Interim and JRA-55. However, there are common interesting characteristics: The GW contribution is positive in most months and maximized in March (i.e., spring) for the NH and in July (i.e., winter) for the SH, although the estimate in June for JRA-55 could not be made because the turn-around latitude is lower than 20° in the SH.

Figure 11 shows the results for 10 hPa. It is commonly seen in the all reanalyses that the total upward mass flux has a strong peak in December and January and a weak peak in October. The magnitude of the upward mass flux is also similar

among the reanalysis. The former peak reflects strong RW activity in the NH and the latter reflects that in the SH. The SH contribution to the upward mass flux in the boreal winter is almost zero, unlike that at 70 hPa as is consistent with mean easterly wind at this level. Estimates on each contribution at this level could not be made for several months in the SH for ERA-Interim and JRA-55 datasets because the turn-around latitude is lower than 20°. Note that the vertical scale is different from that in Fig. 10. According to the results by MERRA-2 and MERRA datasets, the annual mean contribution is less than 5%, which is significantly smaller than that at 70 hPa. However, this result is not robust. The GW contribution at 10 hPa is not low for ERA-I and JRA-55, even for the months when the estimation was made.

However, it is confirmed from Figs. 4 and 5 that the GW contribution to the upward mass flux is likely small at 10 hPa (and 3 hPa as well) for the annual mean. As discussed in Sect. 3.1, the GW forcing largely modifies the turn-around latitudes at all analyzed levels of 70 hPa, 10 hPa, and 3 hPa. This fact does not contradict the small GW contribution to the upward mass flux as mentioned above. The upward mass flux is determined by the stream function at the turn-around latitude, and the potential GW contribution to the stream function, $\Psi_{GW}(\phi_{TL})$, is small at the turn-around latitudes at 10 hPa and 3 hPa (Figs. 4 and 5). Note that the potential GW contribution is sensitive to the location of the turn-around latitude.

The upward mass flux and contribution by each hemisphere is shown in Fig. 12 as a function of the pressure level. As stated, the annual variation with a maximum in the boreal winter is dominant in the lower stratosphere, while clear semi-annual variation is observed in the upper stratosphere. The second maximum is observed earlier at the higher altitudes in the austral winter and/or spring above the 10 hPa level, which is attributed to the SH circulation.

## 5. Seasonal variation of the potential GW contribution and its relation with the GW activity shown by previous studies

The $\Psi_{GW}(\phi, z)$ is equatorward in the low-latitude region and poleward in the middle-latitude region in most seasons, although the strength and vertical extension slightly differ depending on the reanalysis dataset. In this section, we describe the GW contribution in terms of seasonal dependence and consistency with previous observational studies.

The poleward circulation in $\Psi_{GW}(\phi, z)$ in the middle- and high-latitude regions is strongest in winter (DJF) and second strongest in autumn (SON) in the NH, while it is strong in winter (JJA) and spring (SON) with a slight difference in the strength in the SH. The maximum in winter in both hemispheres is consistent with previous GW studies using radiosondes (Allen and Vincent, 1995; Wang and Geller, 2003) and radars (Sato, 1994). The strong spring circulation in the SH is consistent with the fact that the GW energy is maximized in spring in the high-latitude region (Pfenninger et al., 1999; Yoshiki and Sato, 2000). Note such a spring maximum is also seen at Davis in the Antarctic in Fig. 10 of Allen and Vincent (1995), although its presence was not documented. It is interesting that for equinoctial seasons, the poleward circulation in $\Psi_{GW}(\phi, z)$ is stronger in SON than in MAM for both hemispheres.

The equatorward circulation of $\Psi_{GW}$ in the low-latitude region is strong in summer and weak in winter for both hemispheres (Figs. 6 and 7). This is consistent with radiosonde observations by Allen and Vincent (1995) and rocketsonde observations by Eckermann et al. (1995) for subtropical regions, and a GW-resolving general circulation model (Sato et al.,

2009). Interestingly, in the equinoctial seasons, the equatorward circulation is stronger in SON than in MAM for both hemispheres, similar to the poleward circulation in the middle- and high-latitude regions. This result suggests that GW activities are stronger in SON than in MAM almost globally. This point should be confirmed by observations because the GW characteristics in equinoctial seasons have not been studied in depth thus far.

It is also worth noting that GW activity in the equatorial stratosphere is largely modulated by the quasi-biennial oscillation (Alexander and Vincent, 2000; Sato and Dunkerton, 1997) and does not show clear seasonal variation, although clear seasonal variation is seen at the cloud top level in the troposphere (Sato et al., 2009; Kang et al., 2017). This feature cannot be examined in this study because $f$ (or $\hat{f}$) in the denominator of Eqs. (7) and (10) is used for the estimation.

## 6. Remarks on the GW parameterizations

In this section, stream function due to parameterized GW forcing $\Psi_{\mathrm{pGW}}$ is obtained and compared with the potential GW contribution $\Psi_{\mathrm{GW}}$. Such comparison must give an important insight for future improvement of GW parameterizations. The $\Psi_{\mathrm{pGW}}$ was obtained using

$$\Psi_{\mathrm{pGW}}(\phi, z) \equiv -\cos\phi \int_z^\infty \left[\frac{\rho_0}{\hat{f}} \, \mathrm{p}GWF\right]_{\overline{m}} d\zeta, \tag{15}$$

where p$GWF$ is GW forcing expressed by parameterizations. For ERA-Interim, $\Psi_{\mathrm{pGW}}$ was obtained using total tendency due to physics which should be representative of the GW forcing in the stratosphere (Abalos et al., 2015).

The upper panels in Fig. 13 show annual mean $\Psi_{\mathrm{pGW}}$ for MERRA-2, MERRA, ERA-Interim, and JRA-55. The lower panels show the potential GW contribution $\Psi_{\mathrm{GW}}$, which is the same as in Fig. 2, for comparison. The difference between the upper and lower panels may indicate the deficiency of the GW parameterizations in each reanalysis data. It is encouraging that similarity among the four reanalyses is higher for $\Psi_{\mathrm{GW}}$ than for $\Psi_{\mathrm{pGW}}$. This is also the case for seasonal mean state (e.g., JJA mean as shown in Fig. 14). The data assimilation is originally performed so as to make the model results to be better compared with observations. The similarity in $\Psi_{\mathrm{GW}}$ among the four reanalysis datasets may show that the GW contribution to the total stream function is realistically expressed in the reanalysis data fields as a result of the assimilation with modern methods.

The $\Psi_{\mathrm{pGW}}$ for MERRA including non-orographic GW parameterization and JRA-55 with Rayleigh friction has equatorward circulation at latitudes lower than 30° in the middle and upper stratosphere. This is absent in the $\Psi_{\mathrm{pGW}}$ of ERA-Interim, which does not use non-orographic GW parameterization. Thus, the equatorward circulation is likely due to non-orographic GWs. The $\Psi_{\mathrm{pGW}}$ for MERRA-2 with non-orographic GW parameterizations does not have clear equatorward circulation, either. However, the poleward circulation at the middle and high latitudes is weak in its lower latitude part. The background non-orographic GW forcing and intermittency of orographic GW forcing are different between MERRA and MERRA-2 (Molod et al., 2015). The non-orographic GW forcing was increased at latitudes lower than 20° so as to simulate

the quasi-biennial oscillations in the lower stratosphere for MERRA-2. However, this difference does not directly affect the equatorward circulation at latitudes higher than 20° that we focused on.

On the other hand, the equatorward circulation in $\Psi_{GW}$ in the middle and upper stratosphere is stronger than $\Psi_{pGW}$ for all reanalysis datasets except for JRA-55 using Rayleigh friction. This result may suggest that net non-orographic GW forcing is more strongly eastward in the real atmosphere than given by the parameterizations.

On the other hand, for the middle and high latitudes, $\Psi_{GW}$ is stronger than $\Psi_{pGW}$. Particularly, $\Psi_{pGW}$ of ERA-Interim is almost zero around 60°S where the surface is mostly covered by the ocean, while $\Psi_{GW}$ is rather (negatively) maximized there. This point can be more clearly seen in the JJA-mean stream function shown in Fig. 14. It is seen that winter (SH) circulation is generally stronger for $\Psi_{GW}$ than for $\Psi_{pGW}$ in all reanalyses. A gap in the stream function observed around 60°S for $\Psi_{pGW}$ does not exist for $\Psi_{GW}$, and $\Psi_{GW}$ is rather maximized there for all reanalyses. The maximum around 60°S in $\Psi_{GW}$ is consistent with observations and GW-resolving GCM simulations (Sato et al., 2009; Geller et al, 2013). Unlike MERRA, the intermittency factor for orographic GW parameterization is gradually increased (i.e., the forcing is increased) as the latitude increases until approximately 40°S for MERRA-2 (Molod et al., 2015), which should reflect the difference in $\Psi_{pGW}$ between MERRA and MERRA-2. It seems that weakness of GW forcing around 60° is more enhanced for $\Psi_{pGW}$ in MERRA-2 than in MERRA. However, the strength of $\Psi_{pGW}$ except for the gap around 60° is close to that of $\Psi_{GW}$. This fact suggests that not only are orographic GWs over small islands unresolved in the model (e.g, Alexander et al., 2009), but that other mechanisms are also important to make the $\Psi_{GW}$ maximum around 60°S. Candidate mechanisms are generation of non-orographic GWs from convection and flow imbalance (e.g., Plougonven et al., 2015; Shibuya et al., 2015) and latitudinal propagation of GWs due to refraction and/or advection in the strong westerly jet (e.g., Sato et al., 2009, 2012). Note that Geller et al. (2013) examined GWs with horizontal wavelengths typically shorter than 1000 km. A longer wavelength part of the GWs may be in a resolvable range for the GCMs used for the reanalyses in terms of the horizontal resolution. However, considering relatively coarse vertical grid of the model, the longer horizontal wavelength part may not be fully resolvable because they may be subgrid-scale GWs in the vertical. Such GWs should be parameterized in the GCMs as well.

In JJA, equatorward circulation as a part of winter circulation is observed in the summer hemisphere (i.e., NH) as shown in Fig. 7. The stream function due to parameterized GW forcing $\Psi_{pGW}$ has a clear equatorward circulation for MERRA and JRA-55 (as shown in the upper panels in Fig. 14), but the latitudinal extension is different. The equatorward circulation extends to mid-latitudes for JRA-55, while it is confined in low latitudes for MERRA. The equatorward circulation is not clear for MERRA-2 and ERA-Interim. In contrast, for $\Psi_{GW}$, the equatorward circulation has similar latitudinal extension for all datasets, as shown in the lower panels of Fig. 14. This suggests that assimilation works to show realistic extension of the equatorward circulation in the summer hemisphere.

It is also worth noting that the poleward circulation in $\Psi_{pGW}$ of the summer hemisphere (i.e., NH) is quite different among the four reanalysis datasets. In contrast, a small poleward circulation at summer mid-latitudes is similarly represented for $\Psi_{GW}$ for all datasets, although it is weak for MERRA. Particularly, the summer poleward circulation is quite small for

ERA-Interim without non-orographic parameterization. This difference between $\Psi_{pGW}$ and $\Psi_{GW}$ shown in Fig. 14 indicates that parameterized non-orographic GW forcing is needed in the summer mid-latitude circulation, but it is too strong for MERRA-2 and MERRA in the middle and upper stratosphere.

Figure 15 shows the stream functions due to assimilation increment in the zonal mean zonal wind tendency (INC) $\Psi_{INC}$ for MERRA and MERRA2:

$$\Psi_{INC}(\phi, z) \equiv \cos\phi \int_z^\infty \left[\frac{\rho_0}{\hat{f}}\, INC\right]_{\overline{m}} d\zeta, \qquad (16)$$

Annual mean and JJA mean results are shown. It is clear that equatorward circulation is observed at low latitudes of the both hemispheres in the annual mean $\Psi_{INC}$. The equatorward circulation corresponding to the summer hemispheric part of winter circulation is also observed at low latitudes of the summer hemisphere (i.e., NH) for JJA mean $\Psi_{INC}$. These features are consistent with the difference between $\Psi_{pGW}$ and $\Psi_{GW}$, and suggest a shortage of eastward GW forcing at the low latitudes in the parameterization.

Another interesting feature is observed in the winter hemisphere (i.e., SH) for JJA mean $\Psi_{INC}$. A poleward circulation is significant at mid-latitudes up to the upper stratosphere having a slight poleward tilt in the lower stratosphere, and an equatorward circulation is observed at low latitudes with a slight poleward tilt above 50 hPa. This structure suggests that westward GW forcing is too strong at lower latitudes and too weak at higher latitudes in the middle and upper stratosphere. If this feature reflects the deficiency of GW parameterization, a plausible explanation for this structure in $\Psi_{INC}$ is an insufficient source of eastward (westward) propagating GWs in lower (higher) latitudes relative to the mean wind. Poleward propagation of GWs accompanying westward momentum fluxes through refraction and advection, and their own horizontal group velocity (e.g., Sato et al. 2009; 2012) could also explain this pattern with tilting structure.

## 7. Summary and concluding remarks

The climatology of the residual mean circulation in the whole stratosphere, a main component of BDC, has been examined by using four reanalysis datasets (MERRA-2, MERRA, ERA-Interim, and JRA-55) over 30 years (1986–2015) based on the TEM primitive equation. One purpose of this study is to examine the role of RWs, GWs, and zonal mean zonal wind tendency, which is mainly due to a radiation effect, in the residual mean circulation. Resolved and unresolved waves in the datasets were respectively designated as RWs and GWs, although resolved waves may contain some GWs. The other is to describe the circulation in the middle and upper stratosphere, which is available with the aid of the recent reanalysis covering the upper stratosphere and the lower mesosphere. The residual mean circulation in the equinoctial seasons was also examined. Analysis was focused on the stream function of the residual mean circulation in the whole stratosphere and lowermost mesosphere and the upward mass flux at 70 hPa and 10 hPa evaluated from the stream function.

The stream function of the total residual mean circulation was divided into three components: RW forcing, GW forcing, and the zonal mean zonal wind tendency, according to the zonal mean zonal momentum equation. The former two

components were examined as potential RW and GW contributions, and the latter as a potential radiative contribution. The total residual mean stream function was directly estimated by its definition. The potential GW contribution was estimated as the residual of the contributions by RWs and zonal mean zonal wind tendency from the total residual mean stream function. Vertical advection of the zonal mean zonal wind is also included for the analysis because the GW forcing may be small and comparable to this term in the low-latitude region of the stratosphere.

An important assumption of the method is that the residual mean flow estimated by its definition, EP flux divergence due to resolved waves, and zonal mean zonal wind tendency are accurately estimated using the reanalysis datasets. These three terms are used to estimate the potential GW contribution indirectly. Particularly, $\overline{w}$ (and $\overline{v}$) in the residual mean flow [see Eq. (3)] is not well constrained because it is not usually observed, and not balanced with well-observed quantities, such as temperature. Thus, this analysis is only possible if the dynamics of the model are realistically maintained while assimilating observation data. In general, it is difficult to validate this assumption directly. However, indirect estimates of the potential GW contribution are considered likely to exhibit features in the real atmosphere for two reasons. First, the results from the four reanalysis datasets were qualitatively quite similar with some quantitative difference. Second, the features observed in the indirect estimates of the potential GW contribution were consistent with our knowledge from high resolution observations and GW-resolving GCM simulations. The common results obtained from the four reanalysis datasets are summarized below.

The annual mean total residual circulation is approximately symmetric around the equator. It is composed of an equator-to-pole circulation in each hemisphere. The total residual circulation is determined by the RW forcing. However, the contribution of GWs is also significant. The circulation by GWs is equatorward in the low-latitude region and poleward in the middle- and high-latitude regions, which correspond to eastward and westward forcings, respectively. This GW-induced circulation determines the turn-around latitudes of the total circulation at each height and extends the total circulation to high latitudes in the middle and upper stratosphere. This is one of the new and important findings elucidated by this study. Similar GW contributions are observed in all seasons.

The total circulation in the equinoctial seasons is interesting. The structure is not symmetric around the equator. Rather, it is wider in autumn than in spring. This asymmetry is attributable to the radiative-driven circulation from the spring pole to the autumn pole corresponding to the zonal mean zonal wind tendency, which is understood by the angular momentum conservation. In contrast, the RW and GW contributions are almost symmetric around the equator. The direction of the radiative circulation is the same as that of potential RW and GW contributions in autumn but opposite in spring, except for the GW contribution in the low-latitude region.

The potential GW contribution exhibits interesting seasonal variation, which is maximized in slightly different seasons between the NH and SH. The maximum is observed in winter in both hemispheres, but the second maximum is observed in autumn in NH and in spring in SH. This means that the GW contribution is stronger in SON than in MAM globally. It is interesting to confirm this feature by analyzing GWs using high-resolution satellite observations.

The upward mass flux exhibits annual variation with a maximum in the boreal winter in the lower stratosphere, while it is maximized twice a year in the middle and upper stratospheres. The boreal winter maximum in the lower stratosphere is

explained not only by strong RW activity in winter NH, but also by strong RW activity in summer SH. The annual mean GW contribution to the upward mass flux is not very large—approximately 10–40% at 70 hPa depending on the reanalysis. It is interesting that the GW contribution is smaller at 10 hPa and 3 hPa. This is because the GW contribution is relatively small at the turn-around latitude at 10 hPa and 3 hPa, although the turn-around latitude itself is largely affected by GWs.

It is again emphasized that the features of the potential GW contributions to the residual mean circulation described above are commonly observed for all reanalysis datasets, suggesting that they are robust results. Comparison between the stream function due to parameterized GW forcing and the indirectly estimated potential GW contribution suggests inadequacy of the current GW parameterizations—that is, shortage of eastward GW forcing at low latitudes and that of westward GW forcing at winter high latitudes. This suggests that the GW source description in the parameterizations is not sufficiently

realistic. Another possibility particularly for the shortage of westward GW forcing in the winter high latitude region is the lack of horizontal propagation, which is consistent with the features observed in the assimilation increment.

       BDC affects the global climate by modifying the tropopause structure, such as static stability and westerly jet latitudes (e.g., Kidston et al., 2015; Kohma and Sato, 2014; Li and Thompson, 2013). The significant potential contribution of GWs shown by the present study indicates the necessity of further constraint to the GW parameterization by high-resolution

observations. The use of GW permitting general circulation models is also promising.

**Acknowledgements**

This study was an extensive study initiated by a part of Kota Okamoto's PhD thesis, for which KS supervised. The authors thank Yoshihiro Tomikawa, Takenari Kinoshita, and Masashi Kohma for their fruitful discussion. Thanks are also for Peter Haynes, Marta Abalos, Petr Šácha, and two anonymous reviewers for their constructive comments. The JRA-55 dataset was

20 downloaded from the JRA project site (http://jra.kishou.go.jp/JRA-55/index_en.html), MERRA and MERRA2 from the NASA GES DISC site (https://disc.gsfc.nasa.gov/), and ERA-Interim from the ECMWF Data Server (http://apps.ecmwf.int/datasets/). This study was supported partly by the Sumitomo Foundation, by JSPS Kakenhi Grant Number 25247075, and by JST CREST Grant Number JPMJCR1663.

**Appendix A: Difference in the residual mean stream function between the vertical integration of $\overline{v}^*$ and the latitudinal**

**integration of $\overline{w}^*$**

As described in Sect. 2, there are two methods to estimate $\Psi(\phi, z)$ from the residual mean flow: One is a vertical integration of $\overline{v}^*$ from the top, and the other is a latitudinal integration of $\overline{w}^*$ from the North Pole or South Pole. The former scheme has an advantage in which a relatively large, and hence (probably) a reliable, quantity of $\overline{v}$ can be used, but also a disadvantage in which $\overline{v}^*$ above the top of the data needs to be ignored. In contrast, the latter method requires the use of quite a small quantity

$\overline{w}$, but an exact boundary condition, $\Psi = 0$, at the pole can be used.

Figures Aa, Ad, and Ag show the stream functions obtained from the vertical integration for the annual mean state, for DJF, and for MAM using MERRA-2 data. Figures Ab, Ae, and Ah (Ac, Af, and Ai) show those obtained using the latitudinal integration from the North (South) Pole. Note that the results of two latitudinal integrations from the North Pole and from the South Pole accord at least in the low-latitude region. The difference seen in the high-latitude region of the opposite hemisphere to the initial location for the integration is likely due to the accumulation of error in $\overline{w}^*$ through the integration. The stream functions of total circulation shown in Figs. 2–12 of this paper were made by joining the NH and SH stream functions at the equator, which were obtained by the latitudinal integration from the North and South Poles, respectively.

Next, the results between the vertical and latitudinal integrations are compared. The annual mean $\Psi(\phi, z)$ obtained from the vertical (Fig. Aa) and latitudinal (Fig. 2a) integrations accord well for the main part of the stratosphere, although the $\Psi(\phi, z)$ values from the vertical integration for the upper stratosphere and lower mesosphere above 5 hPa are smaller in both the NH and SH than those from the latitudinal integration. This suggests that the residual mean circulation in the lower and middle stratosphere is mainly determined by the large wave forcing in the stratosphere, but the effect of the wave forcing in the mesosphere is not completely negligible for the circulation in the upper stratosphere above the levels with the large stratospheric forcing. These features are similarly observed for the equinoctial seasons (i.e., Fig. Ag and 8a for MAM).

The difference for the solstitial seasons is more complex. The $\Psi(\phi, z)$ from the vertical integration has a deeper summer circulation and a slightly weaker winter circulation than that from the latitudinal integration, while the summer-to-winter pole circulation caused by the GW forcing that is dominant in the mesosphere is clearer in the lowermost mesosphere in the $\Psi(\phi, z)$ (above 1 hPa) from the latitudinal integration (Fig. 6a). This result is consistent with the existence of the GW forcing in the mesosphere that is westward in the summer hemisphere and westward in the winter hemisphere, which is ignored for the estimation from the vertical integration with a top boundary condition of $\Psi(\phi, z) = 0$.

In conclusion, the stream function of the residual mean circulation is better calculated from $\overline{w}^*$ by the latitudinal integration using recent modern reanalysis datasets. However, it should be noted that both $\overline{v}$ and $\overline{w}$ in Eq. (3) are ageostrophic components, and hence not well constrained by the data assimilation. Thus, it is necessary to further examine the cause of the difference between the two methods using $\overline{v}^*$ and $\overline{w}^*$. A possible way to accomplish this is utilizing outputs of free runs by GW-resolving GCMs with high top, which is left for future research.

**Appendix B: Effects of the vertical shear of mean zonal wind on the residual mean stream function**

As described by Haynes et al. (1991), the vertical integration should be made along a contour of the angular momentum ($m$) when the vertical advection by the residual mean flow $\overline{w}^* \frac{d\overline{u}}{dz}$ is not negligible. This may be the case for a low-latitude region where the latitudinal gradient of the angular momentum is small (i.e., $f$ is small) [see Fig. 1 of Haynes et al. (1991), for example]. However, it is not easy to calculate the integration along the $m$ contour. Thus, several previous studies used a simple integration in the vertical at a latitude ignoring the term $\overline{w}^* \frac{d\overline{u}}{dz}$ instead of the integration along the $m$ contour. It is therefore

useful to compare the results from the two methods and discuss the limitation of the simple vertical integration. It will be useful to discuss the limitation of this simplified method using this comparison.

As seen in Fig. 1 of Haynes et al. (1991), the $m$ contours are greatly distorted at latitudes lower than 30°, and even closed contours are observed near the equator while they are almost vertical at higher latitudes. Figure B-1 (B-2) shows the meridional cross sections of $\Psi_{RW}$, $\Psi_{dU/dt}$, and $\Psi_{GW}$ from the top obtained by the integration in the vertical at each latitude (left) and by that along the $m$ contours (right) in DJF (JJA). As expected, a slight difference is observed in latitudes lower than 30°. A notable difference is observed in $\Psi_{RW}$ for the low-latitude region of the SH in DJF, in which the positive stream function contours are extended more poleward for the results from the along-$m$ integration. As a result, $\Psi_{GW}$ is slightly weaker there. Such difference is not distinct for the NH in JJA. Another difference is observed in $\Psi_{GW}$ in the low-latitude region of the winter hemisphere around 20 hPa particularly in the SH in JJA, where a small counter circulation (i.e., equatorward) is present. This equatorward circulation is more evident for the along-$m$ integration. Similarly, a slight difference was observed for the equinoctial seasons (not shown). However one of the important findings of the present paper, that is stronger equatorward circulation by GWs in the low-latitude region in SON than in MAM, is robust for the different vertical integration. Therefore, it is concluded that although the vertical advection term, $\overline{w}^* \frac{d\overline{u}}{dz}$, is not negligible in the low-latitude region, overall features in the residual circulation, including potential contributions by GWs, can be estimated by a simple vertical integration of the wave forcing.

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

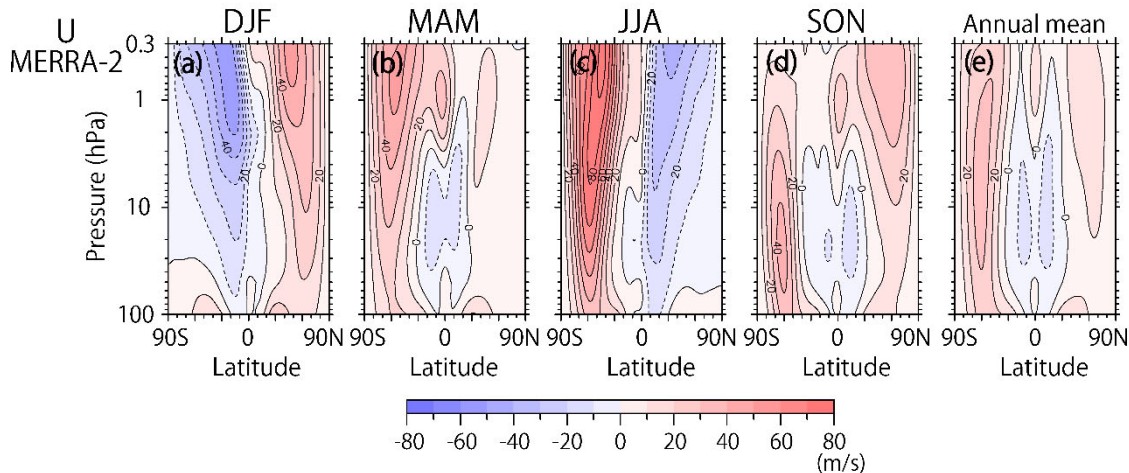

**Figure 1: Meridional cross sections for the climatology of seasonal mean zonal mean zonal wind for (a) DJF, (b) MAM, (c) JJA, and (d) SON, and (e) for the annual mean.**

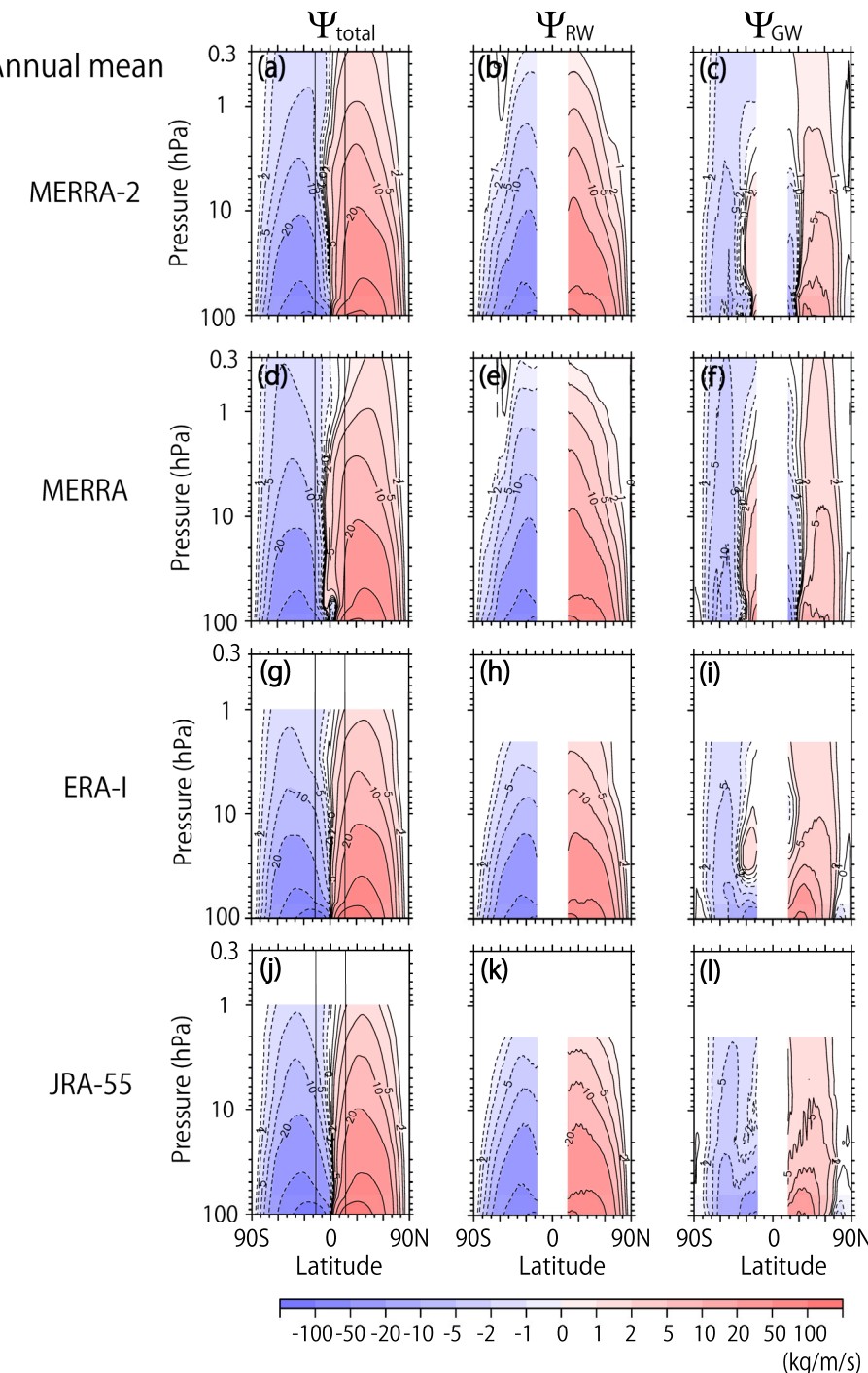

**Figure 2: Meridional cross sections of the climatology of the annual mean stream function of the residual mean flow (a), contributions of RWs (resolved waves) (b) and GWs (unresolved waves) (c) for MERRA-2, for MERRA [(d), (e), and (f)], for ERA-Interim [(g), (h), and (i)], and for JRA-55 [(j), (k), and (l)].**

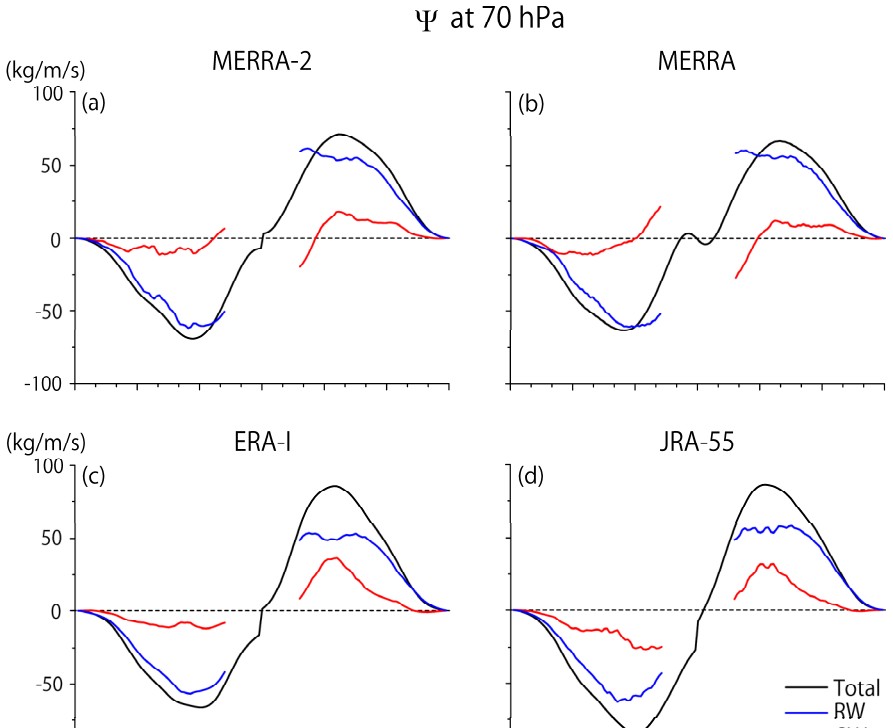

**Figure 3: Latitudinal profiles of the climatology of the annual mean stream function of the residual mean flow (black), contributions of RWs (blue) and GWs (red) at 70 hPa for (a) MERRA-2, (b) MERRA, (c) ERA-Interim, and (d) JRA-55.**

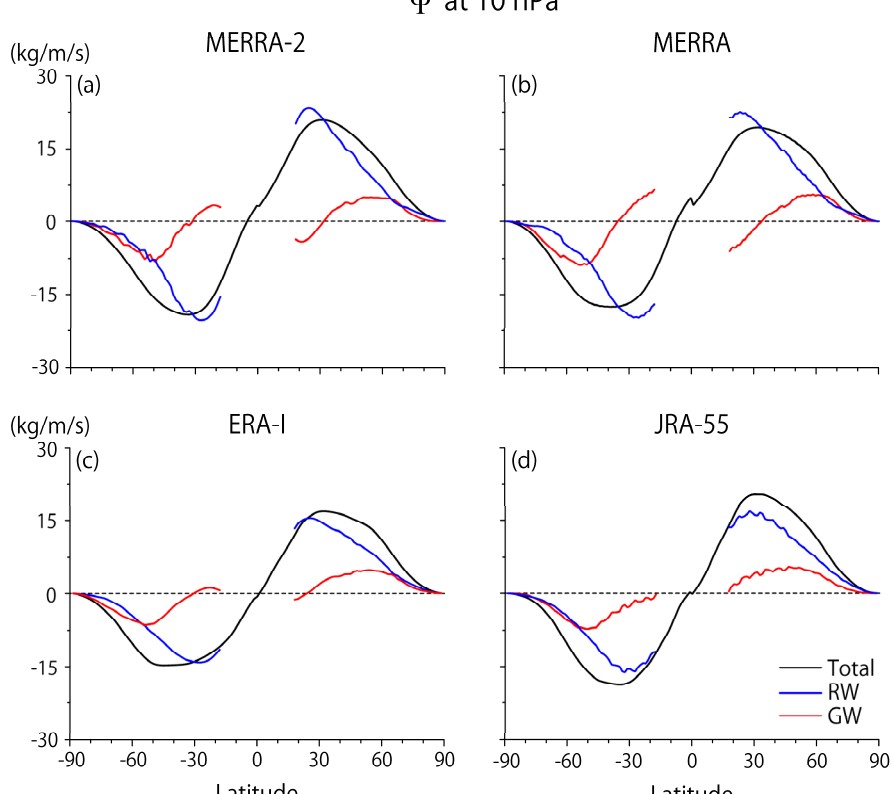

**Figure 4: The same as Figure 3 but for 10 hPa.**

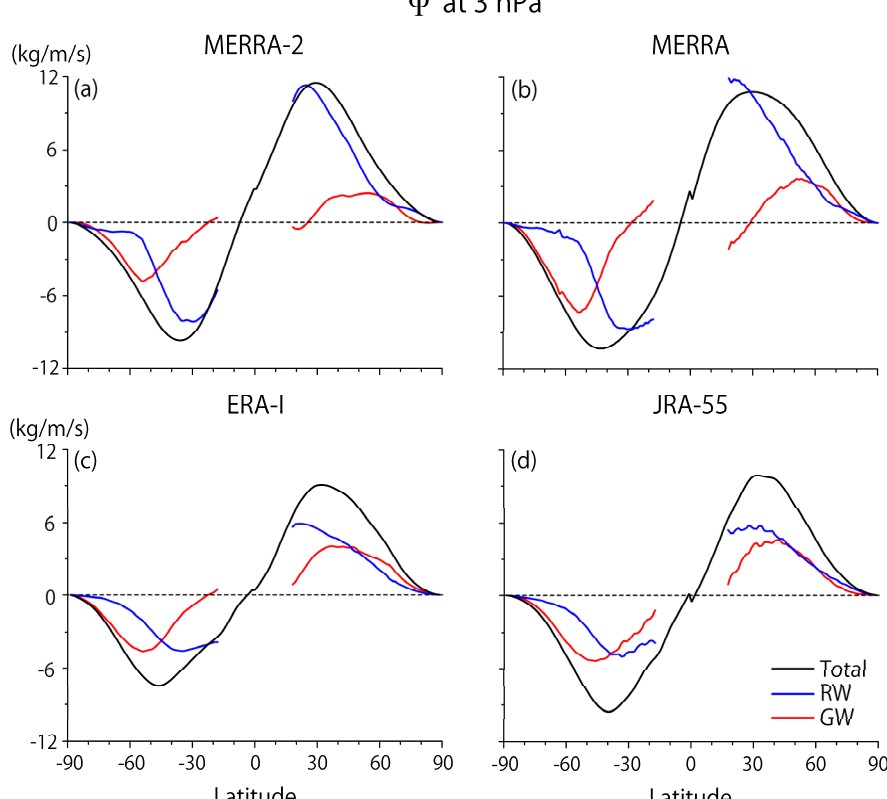

**Figure 5: The same as Figure 3 but for 3 hPa.**

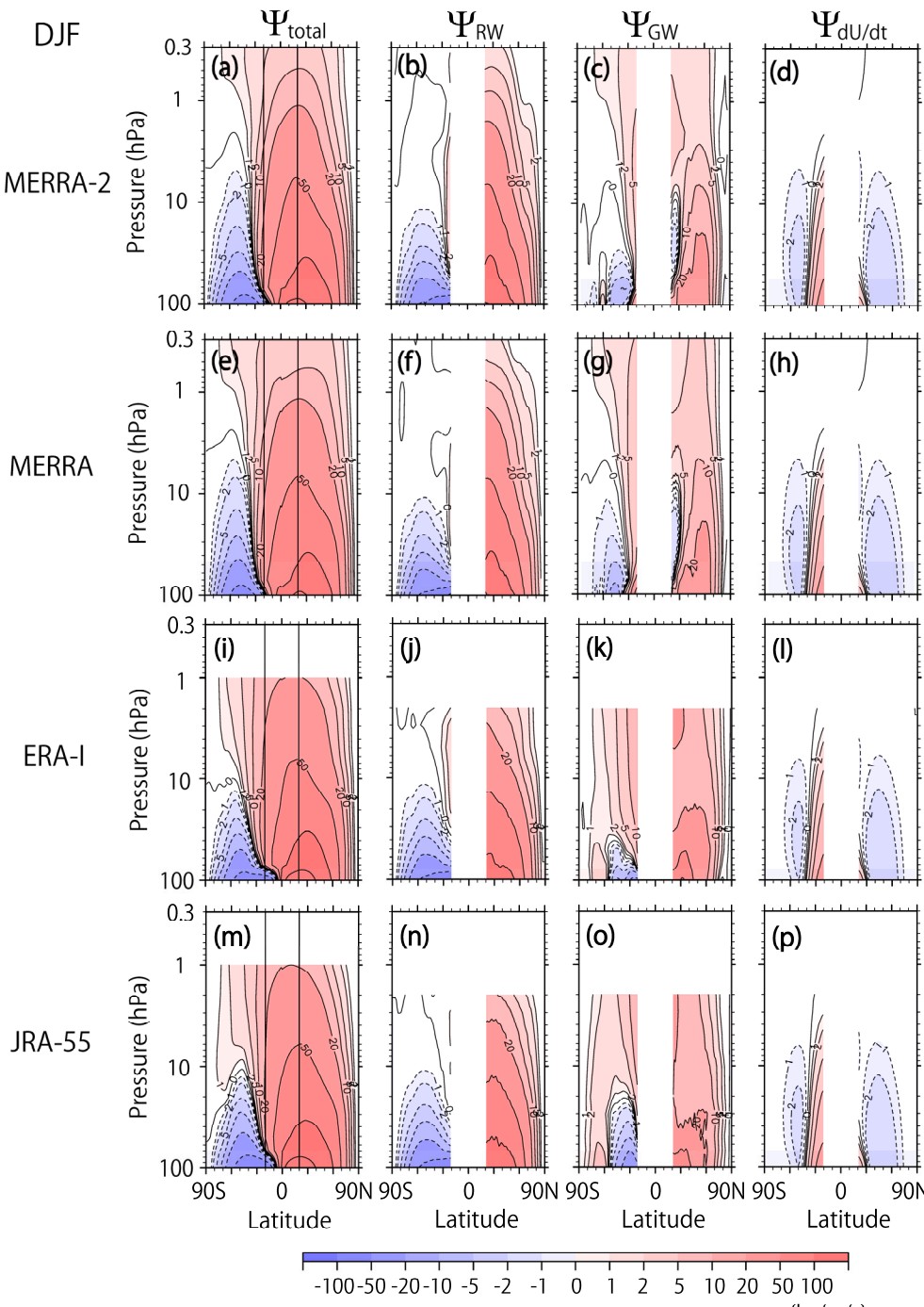

**Figure 6: Meridional cross sections of the climatology of the seasonal mean stream function of the residual mean flow and potential contributions of RWs (resolved waves), GWs (unresolved waves), and the tendency of zonal mean zonal wind in DJF for MERRA-2 [from the left, (a), (b), (c), and (d)], MERRA [(e), (f), (g), and (h)], ERA-Interim [(i), (j), (k), and (l)], and JRA-55 [(i), (j), (k), and (l)].**

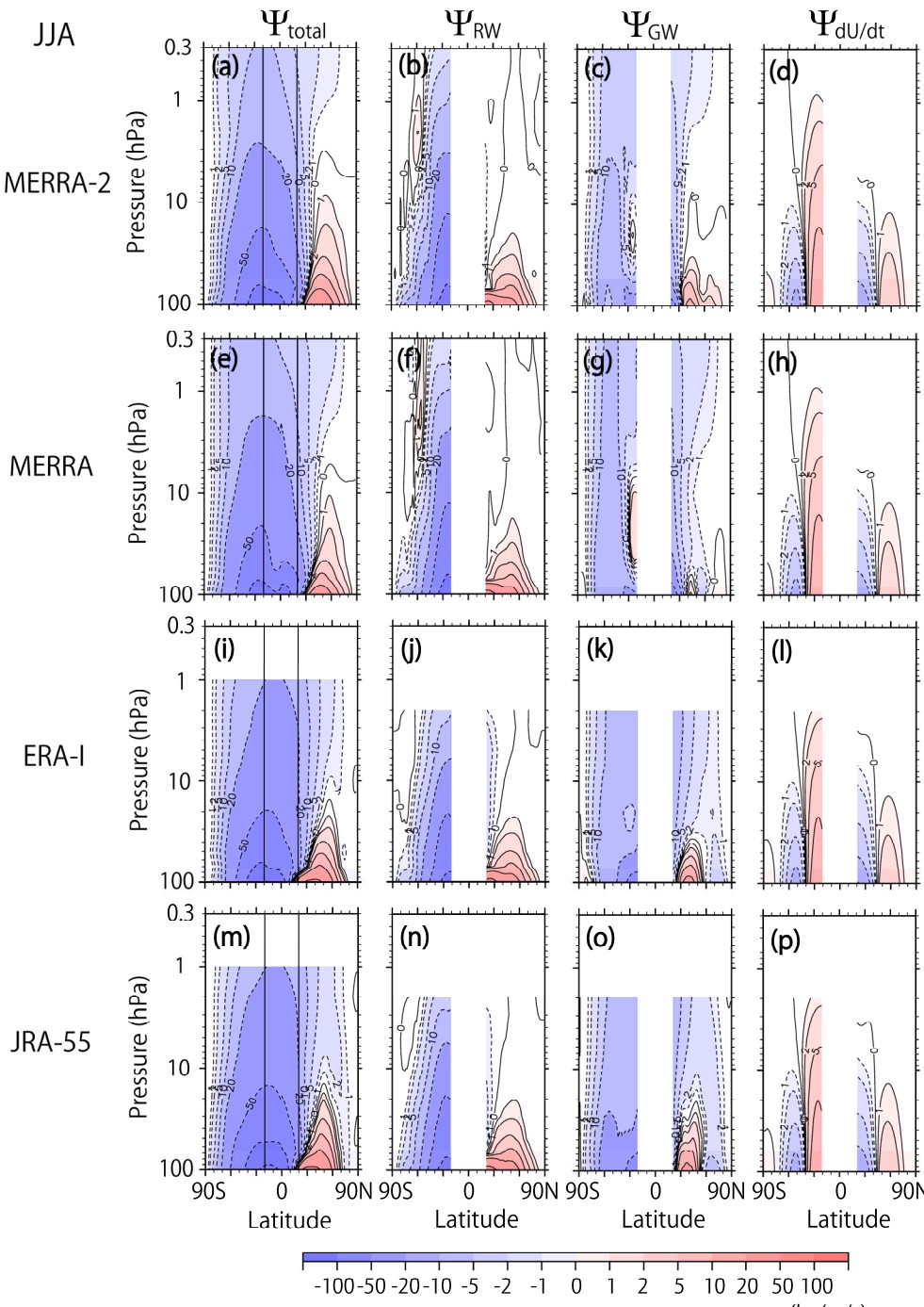

**Figure 7: The same as Figure 6 but for JJA.**

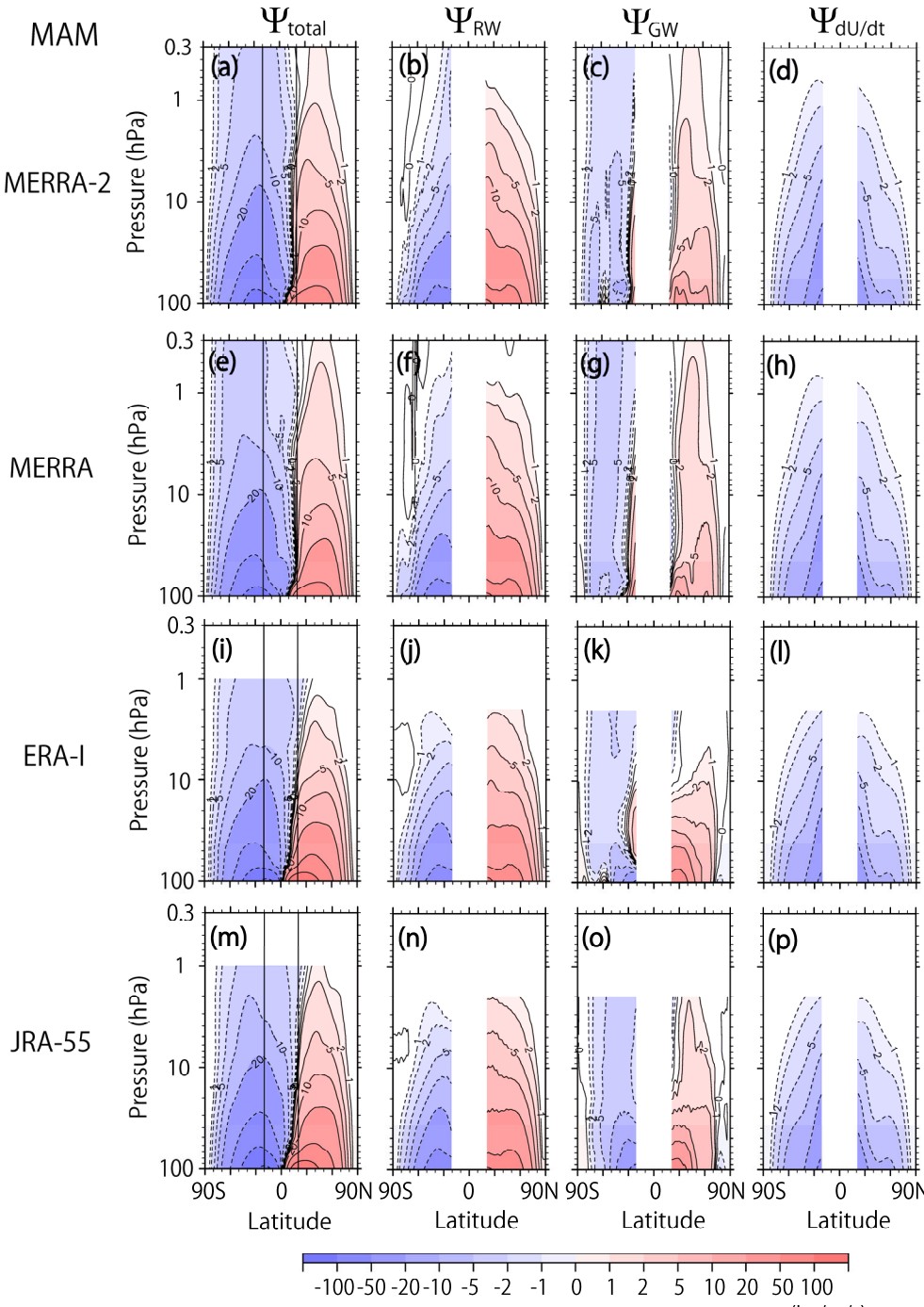

**Figure 8: The same as Figure 6 but for MAM.**

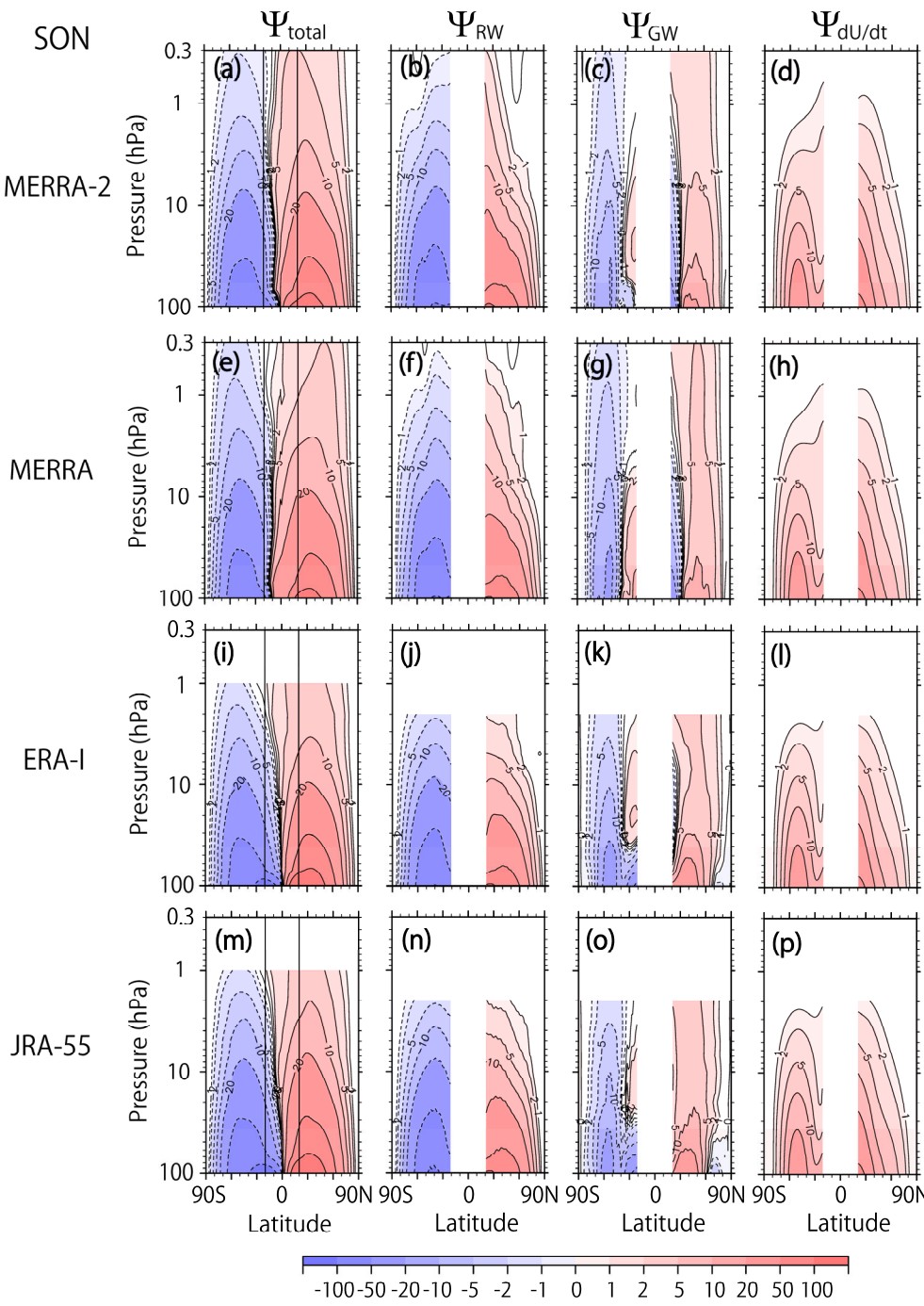

**Figure 9: The same as Figure 6 but for SON.**

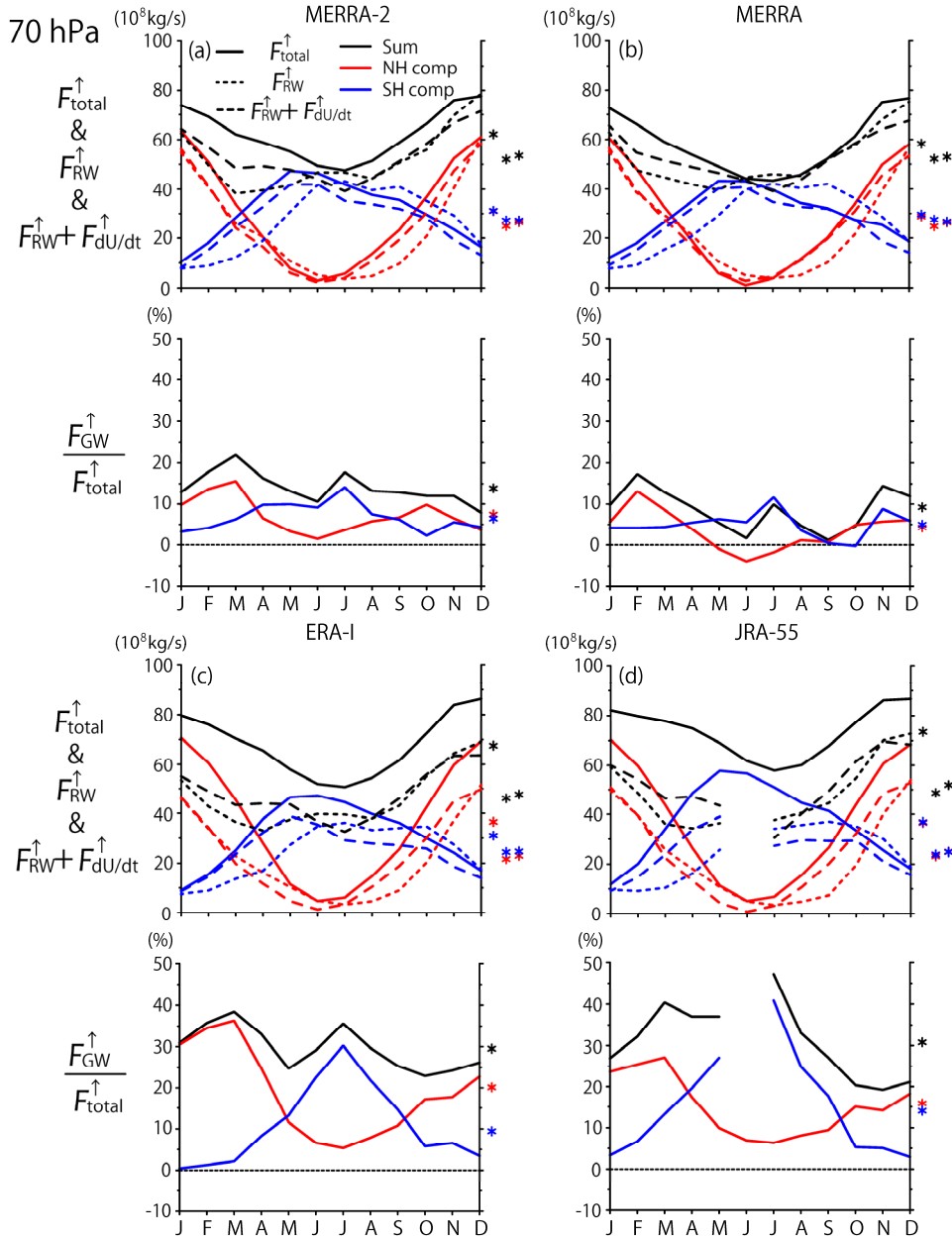

**Figure 10: Upward mass flux at 70 hPa as a function of the month for (a) MERRA-2, (b) MERRA, (c) ERA-Interim, and (d) JRA-55. Upper panel: Black solid curves show the net upward mass flux and red (blue) solid curves show contributions of the NH and SH. Solid curves show the total mass flux. Dashed curves show potential contributions of RWs plus the tendency of the zonal mean zonal wind. Asterisks on the right show the annual mean of total mass flux, potential RW contribution, potential RW contribution plus contribution of the zonal mean zonal wind tendency from the left. Lower panel: Percentage of the potential contribution of GWs to the total mass flux. The asterisks on the right show their annual mean. Contributions by each wave and zonal wind tendency could not be calculated in JJ in the SH for JRA-55 because the turn-around latitude was lower than 20°S.**

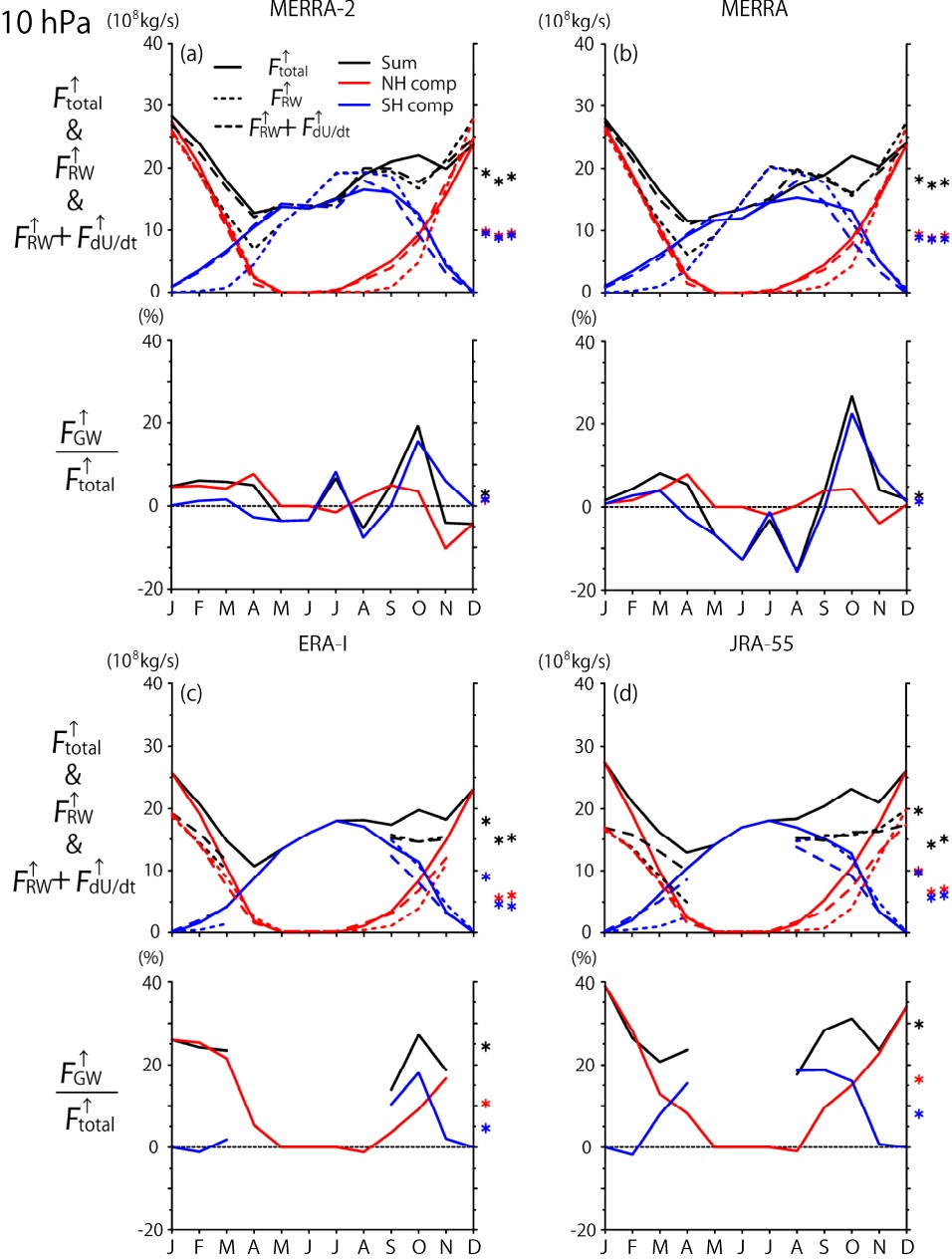

Figure 11: The same as Figure 10 but for 10 hPa.

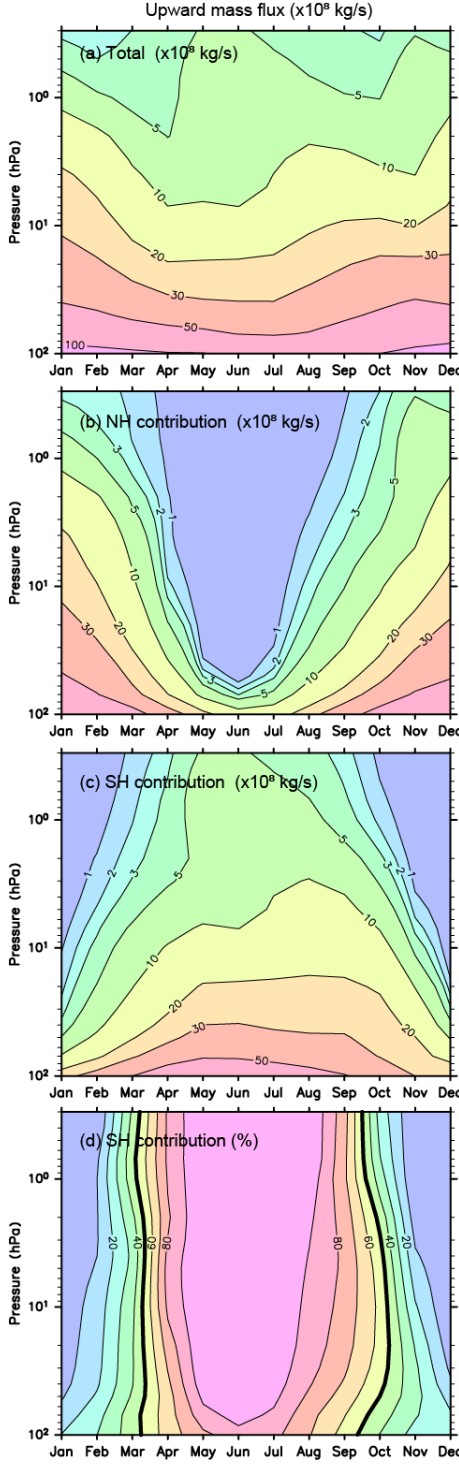

**Figure 12. Upward mass flux as a function of the pressure level. (a) Total and contributions of the (b) NH and (c) SH. (d) The percentage of the SH contribution to the total upward mass flux.**

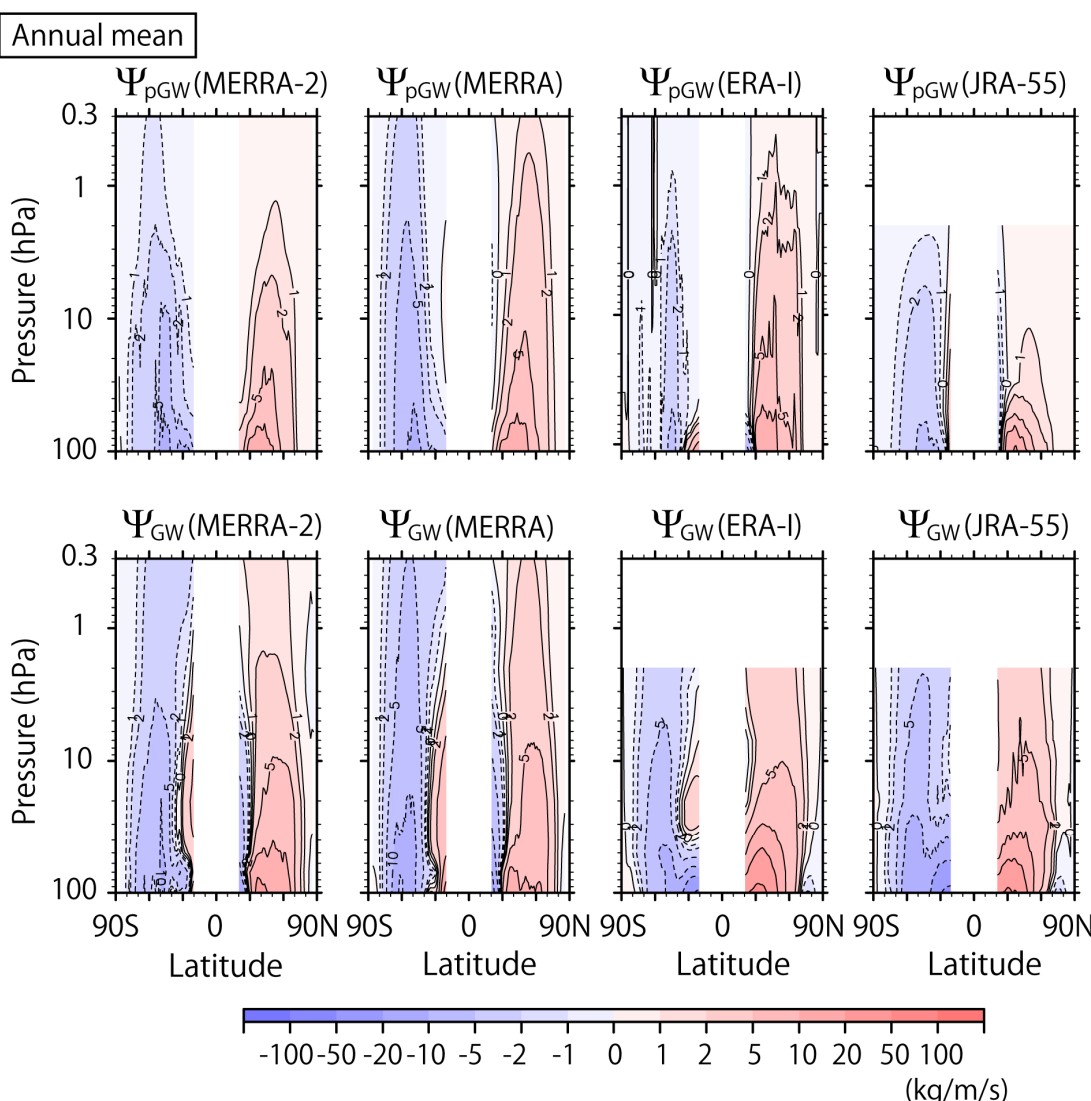

**Figure 13. Meridional cross sections of the climatology of the annual mean stream function due to parameterized GWs (upper panels) and potential GW contributions (lower panels) for MERRA-2, MERRA, ERA-Interim and JRA-55 (from the left).**

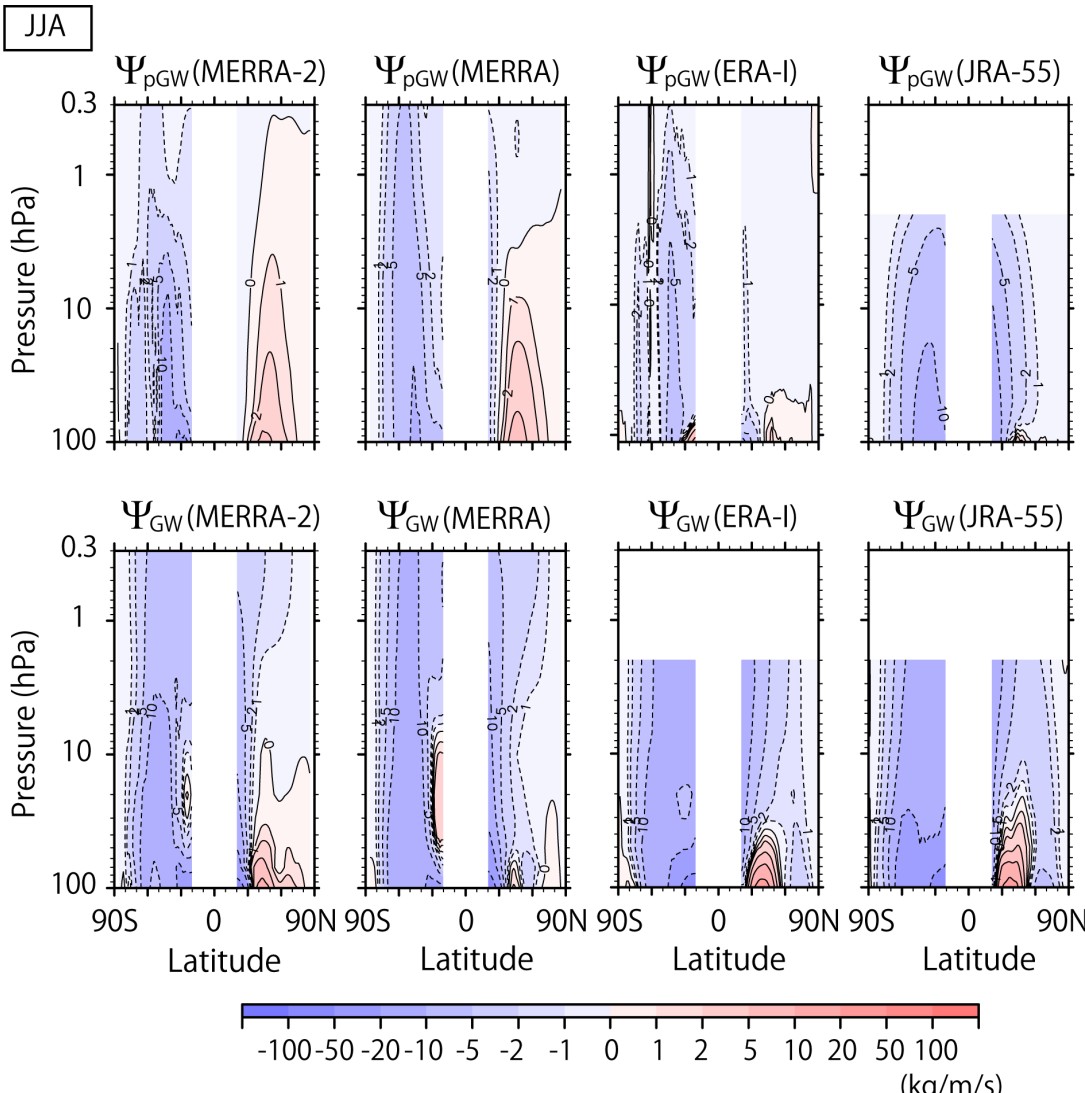

**Figure 14. The same as Figure 13 but for JJA.**

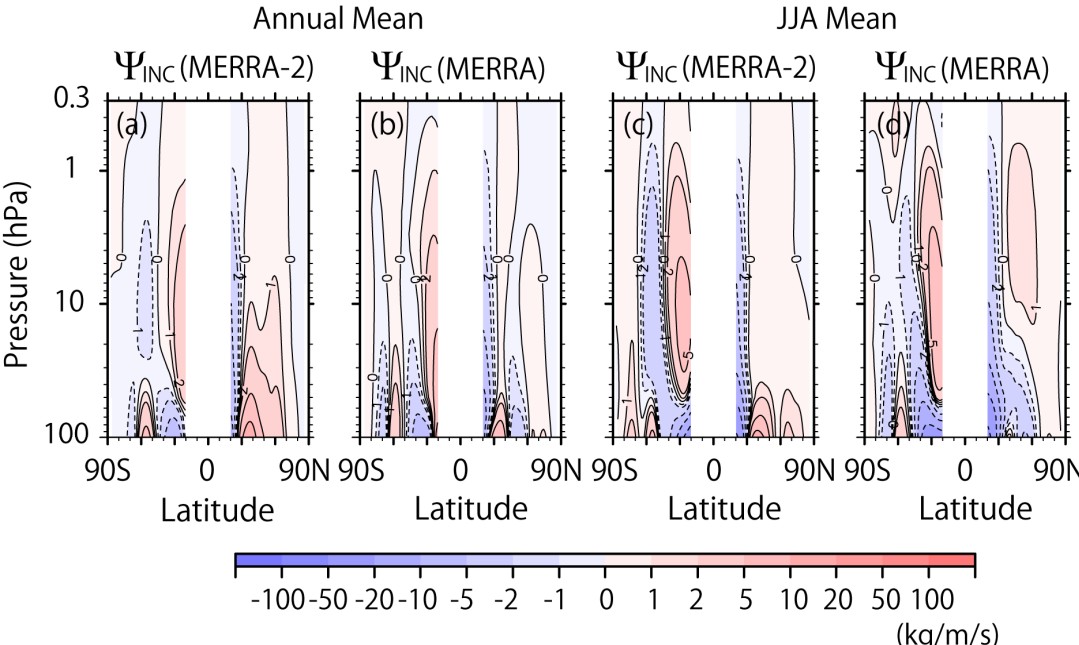

**Figure 15. Meridional cross sections of the climatology of the annual mean stream function due to assimilation increment for zonal mean zonal wind tendency for MERRA-2 and MERRA (left two panels) and of the JJA mean (right two panels)**

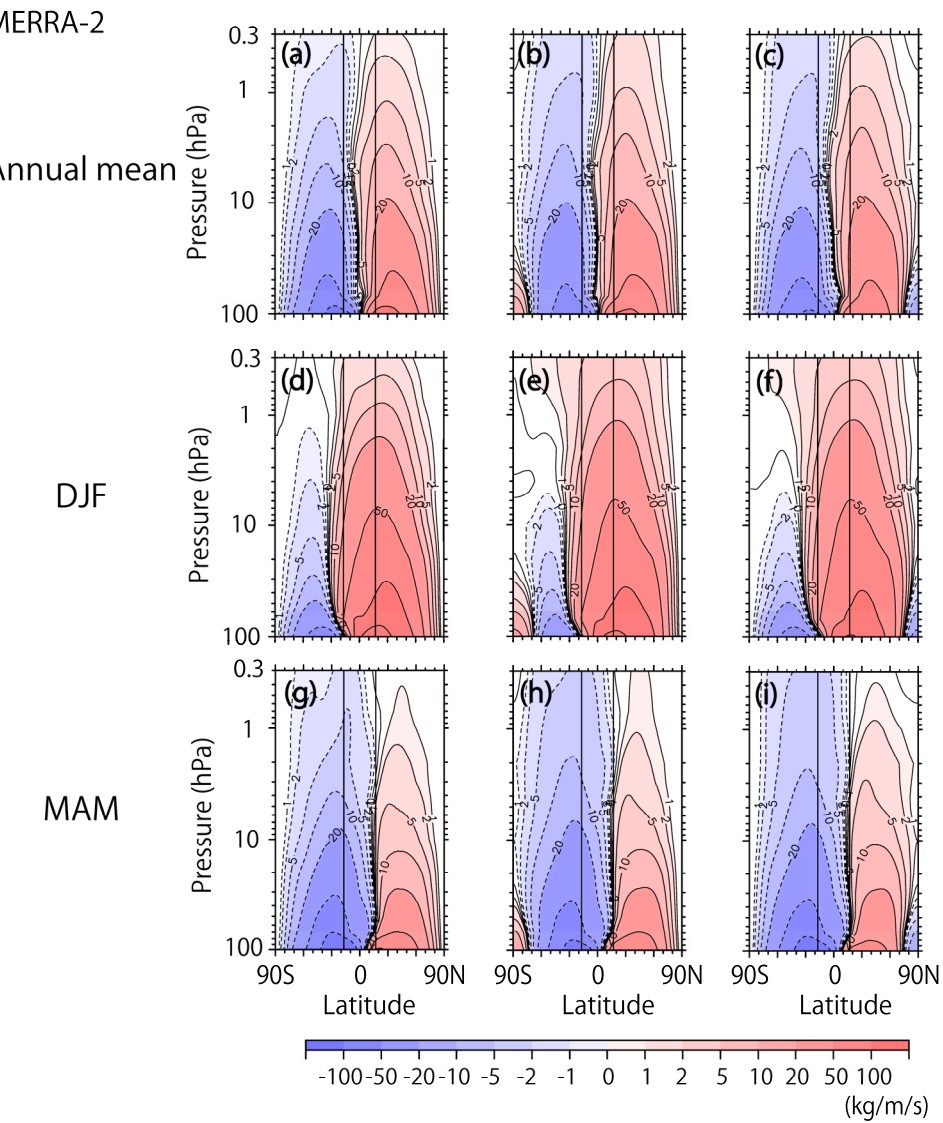

**Figure A. Meridional cross sections of the climatology of the annual mean, DJF, and MAM stream function of the residual mean flow from the top. (a), (d), and (g): Estimates from the vertical integration of $\overline{v}^*$. (b), (e) and (h) [(c), (f) and (i)]: Estimates from the latitudinal integration of $\overline{w}^*$ starting from the north [south] pole.**

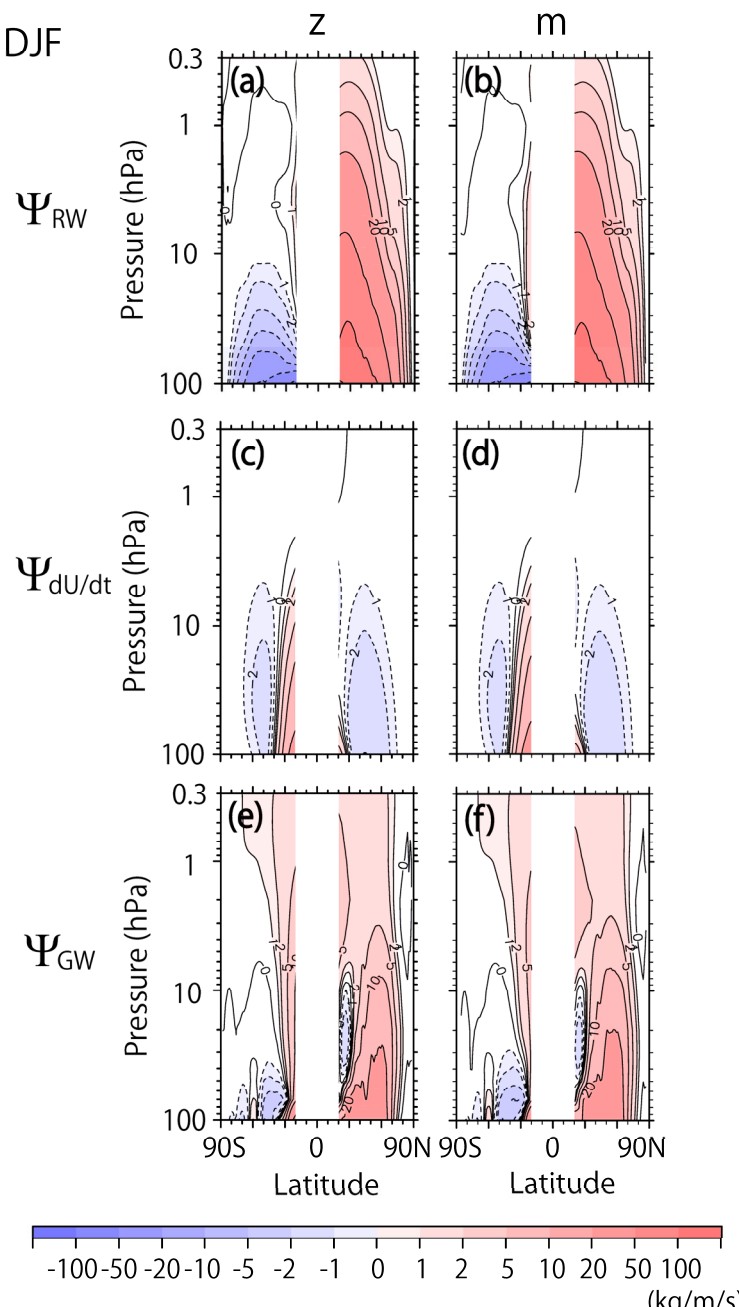

**Figure B-1. Meridional cross sections of the DJF climatology of potential contributions by (a) (b) the RWs, (c) (d) the tendency of zonal mean zonal wind, and (e) (f) the GWs in DJF estimated from MERRA-2. Estimates from (a), (c), (e) a vertical integral at a constant latitude (i.e., ignoring vertical advection of momentum) and from (b), (d), (f) a vertical integral along a constant angular momentum ($m$).**

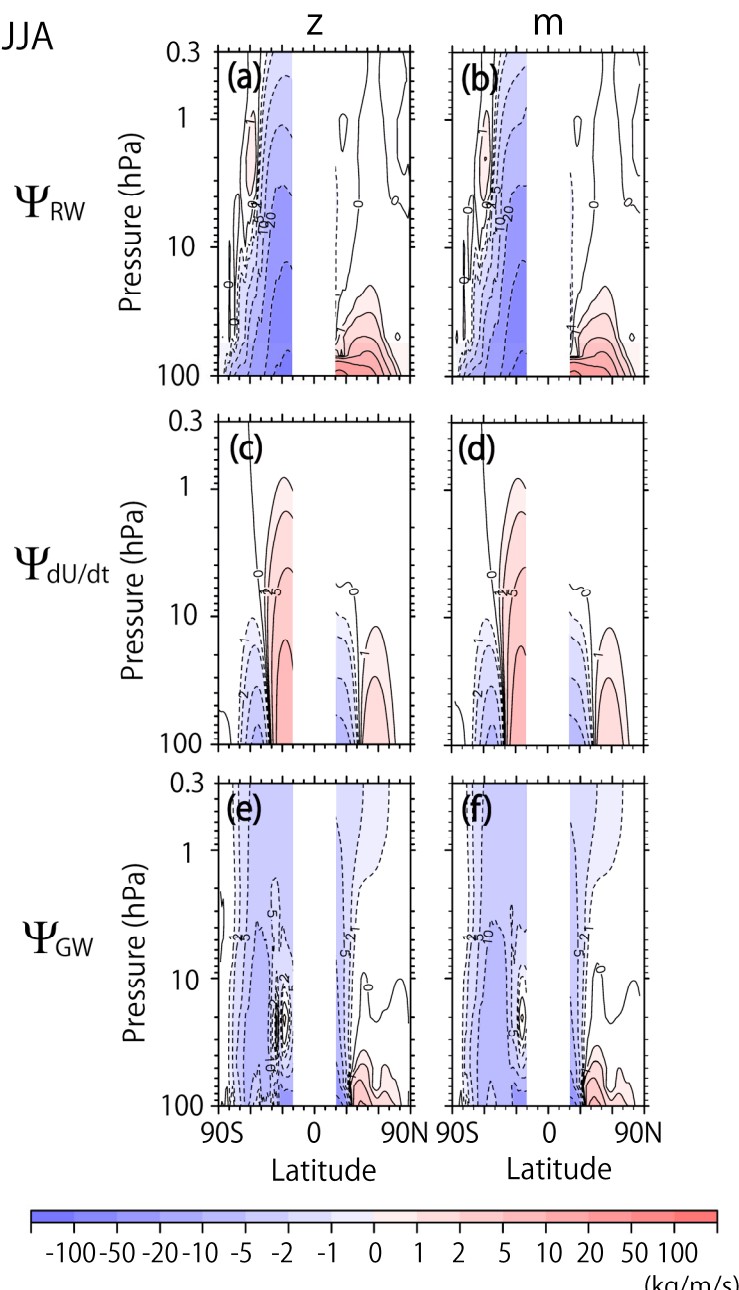

**Figure B-2. The same as Figure B-2 but for JJA.**