# Peer review of "The climatology of Brewer-Dobson circulation and the contribution of gravity waves"

_Atmospheric Chemistry and Physics, 2018_

## Short Comment (SC1) · 9 Apr 2018

Dear authors,

Thank you very much for an interesting study. This comment is mainly motivated by the relationship of your analysis with the compensation mechanism (not only Cohen et al. 2013, but also Cohen et al., 2014 and Sigmond and Shepherd, 2014).

Your argument against the compensation mechanism in reanalyses (P2L30) appears to be speculative and it does not take into account various processes that can stand behind the compensation. For example, Haynes et al. (1991) noted that the DC prin-

ciple applies to the zonally symmetric forcing, as the longitude-dependent force could set-up a Rossby wave field. This was demonstrated in a modeling study by Šácha et al. (2016) together with the effect of different zonal distribution of forcing on the residual circulation. There is an inherent zonal asymmetry in the gravity wave drag distribution (concentration into hotspots -e.g. Hoffmann et al., 2013; Šácha et al., 2015), which is reflected also in the parameterizations (at least orographic GW parameterizations, Šácha et al., 2018).

The compensation hardly makes it possible to clearly separate the effects of resolved and unresolved waves. This is an important point and we think that it has to be properly discussed in your paper.

We also report on a typo (P15L16), where you probably wanted to relate the boundary condition (wstar = 0) to the turn-around latitudes.

Besides that, we have general doubts regarding the conclusions of Appendix A, as we can see similar inequality between the wstar and vstar based method for residual mean streamfunction computation in a model reaching up to 150 km (i.e. including the wave forcing in the mesosphere, not published yet). We would recommend checking if the net tropical upwelling across a particular level inferred from wstar integration nears zero.

Best regards,

Petr Šácha (sacha@uvigo.es) and Petr Pišoft.

References

Cohen, N. Y., Gerber, E. P., and Bühler, O.: Compensation between resolved and unresolved wave driving in the stratosphere: Im- plications for downward control, J. Atmos. Sci, 70, 3780–3798, doi:10.1175/JAS-D-12-0346.1, 2013.

Cohen, N. Y., Gerber, E. P., and Bühler, O.: What drives the Brewer- Dobson circulation?, J. Atmos. Sci., 71, 3837–3855, doi:10.1175/JAS-D-14-0021.1, 2014.

[Figure]

Haynes, P.H., M.E. McIntyre, T.G. Shepherd, C.J. Marks, and K.P. Shine, 1991: On the "Downward Control" of Extratropical Diabatic Circulations by Eddy-Induced Mean Zonal Forces. J. Atmos. Sci., 48, 651–678, https://doi.org/10.1175/1520-0469(1991)048<0651:OTCOED>2.0.CO;2

Hoffmann, L., X. Xue, and M. J. Alexander (2013), A global view of stratospheric gravity wave hotspots located with Atmospheric Infrared Sounder observations, J. Geophys. Res. Atmos., 118, 416–434, doi:10.1029/2012JD018658.

Sigmond, M. and T.G. Shepherd, 2014: Compensation between Resolved Wave Driving and Parameterized Orographic Gravity Wave Driving of the Brewer–Dobson Circulation and Its Response to Climate Change. J. Climate, 27, 5601–5610, doi: 10.1175/JCLI-D-13-00644.1.

P. Sacha, A. Kuchar, C. Jacobi, and P. Pisoft. Enhanced internal gravity wave activity and breaking over the northeastern pacific–eastern asian region. Atmospheric Chemistry and Physics, 15(22):13097–13112, https://doi.org/10.5194/acp-15-13097-2015, 2015.

Šácha, P., Lilienthal, F., Jacobi, C., and Pišoft, P.: Influence of the spatial distribution of gravity wave activity on the middle atmospheric dynamics, Atmos. Chem. Phys., 16, 15755-15775, https://doi.org/10.5194/acp-16-15755-2016, 2016.

Sacha, P., Miksovsky, J., and Pisoft, P.: Interannual variability of the gravity wave drag – vertical coupling and possible climate links, Earth Syst. Dynam. Discuss., https://doi.org/10.5194/esd-2018-1, in review, 2018.
* * *

---

## Referee Comment (RC1) · Anonymous Referee #1 · 16 Apr 2018

This paper compared the Brewer-Dobson circulation diagnosed from four reanalysis datasets, with emphases on the contribution from unresolved gravity waves as well as the seasonal cycle. This is a useful comparison and fits well into the scope of the S-RIP project. However, I found the methodology of the analysis problematic, which may lead to most of the conclusions from the analysis incorrect. Therefore, I cannot recommend publishcation of the paper at this time.

Major comment:

My main concern is on how the contribution from Rossby waves and gravity waves to the Brewer-Dobson circulation is calculated. The decomposition was based on the following equation: Psi=Psi_RW + Psi_GW + Psi_dUdt (their equation 6), where Psi_RW and Psi_dUdt were estimated by integrating the momentum equation, Psi was estimated by integrating the TEM velocity, and Psi_GW was then estimated as a residual. This is valid in theory, but in practice, because reanalysis data is not fully consistent, there should be an additional residual error term on the right-hand side of the equation. As a result, the gravity wave contribution estimated in the paper includes both true gravity wave contribution and the residual errors as well. Furthermore, based on the study by Abalos et al. (2015), a highly relevant study the authors seem to have missed, the residual errors dominate over the true gravity wave contribution.

The authors claimed that "the gravity wave contributions can be estimated only indirectly", which is incorrect. Most modern reanalysis products do explicitly provide the parameterized gravity wave drag employed in their model. Therefore, one can directly estimate the gravity wave contribution to the circulation, which is done in Abalos et al. (2015). According to Abalos et al. (2015), the gravity wave contribution is substantially smaller than the resolved waves in all three reanalysis datasets they analyzed, which are also included in this study. In addition, Abalos et al. (2015) compared the different estimations of the Brewer-Dobson circulation: one based on integration of TEM velocity (equivalent to Psi in this paper), and one based on integration of the momentum equation (equivalent to Psi_RW+Psi_GW+Psi_dUdt here). They reported a larger difference between the two estimations than the contribution from parameterized gravity waves (Fig. 3 and Fig. 4 in Abalos et al.), and larger difference among estimation methods than among datasets. Comparing result shown in this paper with those in Abalos et al. (2015), it is clear that the "gravity wave contribution" estimated here is consistent with the difference between the two estimation methods in Abalos et al., indicating that most of the "gravity wave contribution" here is actually the residual errors.

Other comments:

1. The author used the term "Rossby wave", but what they actually referred to is the resolved waves. It is true that in the extratropics, most resolved waves are indeed

[Figure]

Rossby waves. But in the tropics, there are also Kelvin waves and other gravity waves that are large enough to be resolved.

2. Page 8 Line 28-29: The claim about the usage of gravity wave parameterization in the reanalysis is incorrect. According to Seviour et al. (2011), ERA interim does not include non-orographic gravity wave drag. According to Gelaro et al. (2017), orographic gravity wave drag in included in MERRA2.

Reference: Abalos, M., B. Legras, F. Ploeger and W.J. Randel, 2015: Evaluating the advective Brewer-Dobson circulation in three reanalyses for the period 1979-2012. J. Geophys. Res., 120, doi:10.1002/2015JD023182

Gelaro, R., and coauthors, 2017: the Modern-Era Retrospective Analysis for Reseaerch and Applications, Version 2 (MERRA-2), J. Clim., 30, 5419-5454.

Seviour, W. J., Butchart, N. and Hardiman, S. C., 2012: The Brewer–Dobson circulation inferred from ERA‐Interim. Q.J.R. Meteorol. Soc., 138: 878-888. doi:10.1002/qj.966

---

## Referee Comment (RC2) · M. Abalos (Referee) · 21 Apr 2018

The paper examines the climatology and the Rossby and gravity wave forcing of the residual circulation in four modern reanalyses. The topic is certainly of interest for ACP and the analysis of the forcing of the residual circulation (RC) would make a noteworthy contribution to the S-RIP special issue. Unfortunately, in my opinion the current version of the paper does not provide an accurate description of the RC forcing because it lacks important considerations regarding the balances in the reanalyses, which lead to misinterpretation of the results. While the paper presents some interesting analyses of the RC forcing and seasonality, I consider that it will only be suitable for publication

in ACP after the following major issues are addressed. I included suggestions on how the raised issues could be addressed.

Major issues

- The paper methodology is at present based on the assumption that the difference between the RC computed from the TEM definition (Eqs. 3 and 4) and that estimated from momentum balance (i.e. downward control plus du/dt term) is attributed exclusively to the gravity wave (GW) drag parameterized in the reanalysis. However this assumption is not necessarily valid because it does not take into account that assimilation increments can play a key role in the momentum balance of the reanalyses.

- In contrast with what is stated in the paper, GW drag is provided for all the reanalyses considered here and thus the GW contribution to the RC can be directly computed using Eq. 8. Also, all the reanalyses considered include orographic gravity wave parameterizations, and only ERA-Interim does not include non-orographic gravity waves. This information is found in the reanalysis description papers cited but is wrongly stated in the paper.

Suggestions

In my opinion the paper would notably improve and make a useful contribution to S-RIP if the authors include an analysis of the GW drag (and perhaps the zonal wind assimilation increment) provided by the reanalyses.

This would allow direct evaluation of the contribution of the parameterized GW drag in the reanalysis models to the RC using Eq. 8, without need of the assumption pointed out in the first comment. This calculation was already done in Abalos et al. (2015 JGR) for ERA-Interim, MERRA and JRA-55. However in that paper the GW contribution is not examined in detail for the different seasons and their analysis extends only to 10 hPa, so it will be interesting to present extended results here.

Moreover, analysis of the difference between the total RC computed by explicitly including the forcing by resolved and parameterized GW (Eq. 6) versus the RC computed from the TEM definition (Eqs. 3 and 4) will provide useful information on the momentum budget in the reanalyses. In particular, based on the results of Abalos et al. (2015 JGR) the two estimates of the RC are significantly different (even including the parameterized GW term). This could imply that the GW parameterizations in the reanalyses are insufficiently capturing the role of GW on the RC. In that sense it could be argued that most of the difference is attributed to the GW drag in the real atmosphere but absent in the parameterizations. This important point is not discussed in the paper, and as a result there is a confusion between the GW drag that is parameterized in the reanalyses and the real GW drag assumed to equal the residual of the momentum balance.

Explicitly computing the GW contribution and clearly explaining these issues would substantially strengthen the current discussion in the paper on the role of the different waves on the RC in reanalyses, and on the limitations of current reanalyses GW parameterizations. In addition, consideration of the assimilation increments provided by the reanalyses can help interpret the momentum balance and further clarify these issues.

Other general comments

- Literature citation: The previous studies Iwasaki et al. (2010 R. Met. Soc. Japan), Abalos et al. (2015 JGR) and Miyazaki et al. (2016 ACP) have already examined and compared the residual circulation in modern reanalyses and should be cited accordingly.

- Acknowledgement of the reanalysis centers for providing the data should be included.

- I recommend carefully reading the draft before submitting the new version to improve the wording in several parts.

- I find interesting the analysis of the du/dt term contribution to the RC seasonality. This

term is key for the subseasonal tropical upwelling variability (Abalos et al. 2014 JAS), consistent with the downward control principle (Haynes et al. 1991 JAS), but its role for the seasonal cycle is not fully understood. For instance Kim et al. (2016 JAS) argue that it is negligible for the seasonality in tropical upwelling.
* * *

---

## Author Comment (AC1) · 5 May 2018

We greatly appreciate Dr. Šácha's invaluable comments and suggestion to confirm that $\overline{w}^*$ integration nears zero.

We calculated a latitudinal average of $\overline{w}^*$ at each pressure level and confirmed that it is generally less than a few percent of the $\overline{w}^*$ maximum at each level (Figure 1). So, we do not think that the uncertainty in $\overline{w}^*$ explains much the difference in the stream functions between the two methods using $\overline{w}^*$ and using $\overline{v}^*$. We appreciate the important information that the inequality still remains when using the model reaching up to 150 km. It is naturally expected that the mesospheric gravity wave forcing is

responsible to the structural difference but it is interesting that it is not all.

We will add descriptions and discussion regarding the compensation in the revised manuscript, if we have a chance of revision. Thanks are also for indicating the typo regarding the boundary condition. The correct one is $\overline{\Psi}(\phi, z) = 0$ at the pole.

———————————————

[Figure]

[Figure]

Ratio of the latitudinal mean wstar/the maximum of wstar at each pressure level
Climatology for 1986-2015                                    MERRA-2

**Fig. 1.**

---

## Author Comment (AC2) · 6 May 2018

The authors are extremely grateful to the reviewers' valuable and constructive comments on our manuscript. We are very sorry that we were not aware of the important reference, Abalos et al. (2015, hereafter referred to as A2015). Unfortunately, however, our methodology seems to be completely misunderstood by both reviewers (Anonymous referee #1 and Dr. Marta Abalos). We think that it is necessary to carefully revise the methodology part so as not to confuse the readers. However, here we will explain this method in detail as a quick response to comments from the reviewers. The motivation of our research is partly similar to A2015, but the handling of the gravity

wave drag is quite different.

We used the zonal momentum equation in the TEM framework, but we did not use the gravity wave drag calculated from gravity wave parameterizations. This is because we do "not" think that the gravity wave parameterizations accurately represent real gravity waves. In a group work regarding gravity waves published as Geller et al. (J. Climate, 2013), we showed a significant difference between observations and parameterized gravity waves, and concluded that further improvement through constraints by observations would be necessary for the gravity wave parameterizations. Candidates that may cause this significant difference from observations are that horizontal propagation of gravity waves is not included in most parameterizations (e.g., Sato et al., 2009; 2012), that small islands as a source of gravity waves are not included (e.g., Alexander et al., 2009), and that there is a large uncertainty in non-orographic gravity wave sources and distribution. Thus, we examined and adopted an estimation method without using the parameterized gravity wave drag.

A2015 reported that the residual mean flow $\overline{v}^*$ estimated from the zonal momentum equation using $\frac{\partial \overline{u}}{\partial t}$, $\frac{1}{\rho_0} \nabla \cdot F$, and parameterized gravity wave drag did not match what was obtained directly from its definition (Eq. 3 in the present ACPD manuscript). This is possible evidence of deficiency of the gravity wave parameterizations.

In our study, we used the same zonal momentum equation, but not for the purpose of estimating the residual mean flow $\overline{v}^*$, but for the purpose of estimating gravity wave drag in the real atmosphere (not parameterized gravity wave drag). The residual mean flow $\overline{v}^*$ was directly estimated from its definition in our study. We will explain the difference between the A2015's method and our method by using the following simplified zonal momentum equation:

$$\begin{array}{ccccc} \frac{\partial \overline{u}}{\partial t} & -f\overline{v}^* & = & \frac{1}{\rho_0}\nabla \cdot F & +GWD & +X \\ (1) & (2) & & (3) & (4) & (5) \end{array}$$

Here, *GWD* [Term (4)] is "not" parameterized gravity wave drag but real gravity wave drag, and Term (5) is a friction and/or viscosity term.

A2015 used directly-calculated Term (1) and Term (3) and the parameterized gravity wave drag as Term (4), ignored Term (5), and estimated Term (2). Thus, difference of the parameterized gravity wave drag from the real gravity wave drag can cause large estimation error in the residual mean flow [i.e., Term (2)]. The difference between the parameterized gravity wave drag and the real gravity wave drag would be expressed as an increment in the assimilation system.

In contrast, we used directly-calculated Term (1), Term (2), and Term (3), ignored Term (5), and estimated Term (4). It is important that we calculated Term (2) by its definition (Eq. 3 in the present ACPD manuscript). As is well known, the residual mean flow $\overline{v}^*$ estimated by its definition is a good approximate of Lagrangian mean flow (Eulerian mean flow + a quadratic term of Stokes drift) and is frequently used to estimate upward mass flux etc. quantitatively.

We showed in Okamoto et al. (2011) that this method is effective to estimate Term (4) [in other words, Term (5) is negligible compared with Term (4)] using data from a chemistry climate model (CCM). Note that gravity waves are parameterized components only in this CCM where no assimilation module is implemented.

We hope that our methodology will be clarified by this explanation.

We will also revise as much as possible following all other comments by the reviewers, if there is opportunity for revision.

References

Alexander, M. J., S. D. Eckermann, D. Broutman, and J. Ma (2009), Momentum flux estimates for South Georgia Island mountain waves in the stratosphere observed

via satellite, Geophys. Res. Lett., 36, L12816, doi: 10.1029/2009GL038587.

Geller, M. A., M. J. Alexander, P. T. Love, J. Bacmeister, M. Ern, A. Hertzog, E. Manzini. P. Preusse, K. Sato, A. A. Scaife and T. Zhou (2013), A Comparison Between Gravity Wave Momentum Fluxes in Observations and Climate Models, J. Climate, 26, 6383-6405, doi:10.1175/JCLI-D-12-00545.1

Okamoto, K., K. Sato, and H. Akiyoshi (2011), A study on the formation and trend of the Brewer-Dobson circulation, J. Geophys. Res., 116, D10117, doi:10.1029/2010JD014953

Sato, K., S. Watanabe, Y. Kawatani, Y. Tomikawa, K. Miyazaki, and M. Takahashi (2009), On the origins of mesospheric gravity waves, Geophys. Res. Lett., 36, L19801, doi:10.1029/2009GL039908.

Sato, K., S. Tateno S. Watanabe, and Y. Kawatani (2012), Gravity wave characteristics in the Southern Hemisphere revealed by a high-resolution middle-atmosphere general circulation model, J. Atmos. Sci., 69, 1378–1396, doi:10.1175/JAS-D-11-0101.1.

---

## Referee Comment (RC3) · M. Abalos (Referee) · 7 May 2018

Dear Dr. Kaoru Sato,

I very much appreciate your clear response to my comment. I would like to clarify the key points of my review. I fully understand the methodology of the paper and I consider it valid, but I find it necessary to include a discussion regarding the role of assimilation increments in reanalyses, and I recommend including a comparison to the parameterized GWD from reanalyses, as explained below.

Indeed, Okamoto et al. (2011) showed that in a climate model the GWD equals the

[Figure]

residual of the momentum balance using Terms 1, 2 and 3. However this is not the case in reanalyses because data assimilation produces an assimilation increment, i.e. an additional term in the momentum equation. The working hypothesis of the paper is that most of this assimilation increment is acting to correct for the limitations of GW parameterizations, and thus the residual of the momentum equation can be interpreted as the 'actual' GWD. While I consider this a valid hypothesis, I argue that it should be explicitly stated as such in the paper, because it is not self-evident. There could be model biases having little to do with 'actual' GWD (e.g. in radiative heating) that need to be offset by the data assimilation. Having this discussion in the paper would notably help the reader understand the reasoning behind the methodology.

In addition, in my review I suggest the authors to include an analysis of the parameterized GWD provided by the reanalysis centers. A comparison between the parameterized GWD and that 'estimated' from the balance Terms 1, 2, 3 would be very useful to highlight the limitations of parameterized GWD in reanalyses pointed out in Dr. Sato's comment, especially in the context of an S-RIP paper (e.g. are the differences larger for ERA-Interim which does not have non-orographic GW parameterizations?). It seems to me that such comparison would be useful for the S-RIP community and that the present paper is an adequate place to discuss these issues.

---

## Referee Comment (RC4) · Anonymous Referee #1 · 7 May 2018

I agree with the authors that the parameterized gravity wave drag are not the true gravity wave drag, and the difference between the parameterized and the true gravity wave drag contributes to the increment error. However, I still do not agree with the authors that their method can serve as an estimation for the true gravity wave drag. Basically, the authors are assuming all the terms except the gravity wave drag in the momentum equation can be perfectly accurately calculated from reanalysis data or small enough to be negligible, and therefore the residual term from the equation would be the gravity wave drag. This may be true in a model simulation where all variables are dynamically consistent with each other, which explains the results from Okamoto et al. (2011) the authors are referring to. But the assumption that all the other terms

can be accurately calculated does NOT hold in the reanalysis products. Due to the assimilation process in the reanalysis data, nonphysical increments are introduced in all the variables. As also pointed out by Dr. Abalos, the other reviewer, this increment error does not only arise from gravity wave but also from many other processes as well. The residual term of the momentum equation therefore does not only consist of the true gravity wave drag, but also differences between the calculated and the true value in all the other terms. More importantly, the true gravity wave drag may not dominate in this residual term, so the residual term does not even give a bulk approximation for the gravity wave drag. Take the term (2) for example, if it can be accurately calculated from reanalysis, then one would expected that v* calculated from its definition would be the same among different reanalysis products since they are representing the same real world. But as shown in this paper as well as in Abalos et al. (2015), this directly calculated TEM velocity does vary among reanalysis products, and the difference is NOT small compared to the residual term or the so-called "true gravity wave drag".

I also agree with Dr. Abalos that it would be helpful to compare the increment error explicitly, since it represents the accumulative errors from not only gravity waves but also resolved waves as well as mean circulation. While the parameterized gravity waves may not represent the true gravity wave effects in the real world, I think a comparison among reanalysis is still meaningful.

---

## Referee Comment (RC5) · Anonymous Referee #3 · 10 May 2018

This study presents seasonal variations of Brewer-Dobson circulation in terms of stream function and upward mass flux for 30 years (1985-2015), using four recent reanalysis data sets (MERRA, MERRA2, ERA-Interim, and JRA55). Special emphasis is given to the contribution of gravity waves on the stream function and upward mass flux. Although this is an interesting subject that is likely to extend some previous works related to the same subject, using recent reanalysis data sets that include more recent years and, in particular, more useful variables such as gravity-wave drag (GWD), the methodology used in this paper is highly problematic, and conclusions based on the current method may lead for readers misleading. Therefore, the reviewer could not accept the current manuscript for publication in ACP.

[Figure]

Major Comments:

In the present study, two assumptions were made: (1) stream function calculated using Eqs. (3)-(4), so-called direct stream function (Psi), and using Eq. (5) based on downward-control principle (Psi_DC) is exactly the same. (2) stream function of Psi_DC induced by the residual term X_bar in the TEM equation represents GW contribution. Based on these two assumptions, Eq. (10) is derived. Followings are comments on the two assumptions.

1)Although the stream function Psi and Psi_DC should be equal theoretically, it is not exactly the same, likely because the governing equations used and physical processes in the GCM of each reanalysis data set are somehow different from rather simple TEM equation. Accordingly, the mass flux calculated from the two stream functions are somewhat different from each other as shown in some previous studies. Note that this is different from the case of recent work by Abalos et al. (2015) where Psi_DC is calculated using GWD rather than X_bar, which is not in momentum balance of TEM equation, and their comparison between Pai and Psi_DC stems mostly from difference between X_bar and GWD. It is curious for the reviewer why authors use Eq. (10) in calculation of stream function of GW rather than Eq. (8).

2)The major benefit of Psi_DC is to calculate the contribution of resolved planetary waves (EPD), du_bar/dt, and non-conservative term (represented by X_bar ) separately. The term X_bar can be calculated from any reanalysis data set as a residual of the TEM equation. The reviewer cannot understand why the authors state " Psi_GW cannot be directly calculated because of the unknown X_bar " (Page 6, line 6). The term represents implicitly the parameterized GWD, numerical diffusion, and assimilation increment. In most recent reanalysis data sets that provide GWD variables, the magnitude of GWD is much smaller than that of X_bar. Therefore, even when GWD variables are provided from reanalysis data sets, quite large value of the residual term, say X'_bar, after excluding GWD, is required for momentum balance in the TEM equation. Therefore, the stream function calculated using Eq. (10) of the current study is not

from GWD but from X_bar, which include several sources other than GWs, in particular, assimilation increment.

3)Note that the GWD variables provided from reanalysis data are purely model output, without data assimilation, and thus high degree of uncertainties may exist. In addition, large values of X'_bar from assimilation increment may also include some parts of un-parameterized GWD, if there are. However, assimilation increment stems from various, probably all, processes in the model, including underestimation of resolved wave forcing (EDP), not exclusively from GWD. Therefore, it is not acceptable that stream function calculated using Eq. (10) of the current manuscript represent the stream function from GWs.

Minor comments: 1)GWD variables are available from all reanalysis data set used in the present study, although non-orographic GWD output is not available from ERA-Interim. This is not correctly mentioned in the manuscript.

---

## Author Comment (AC4) · 10 May 2018

Dear Dr. Abalos,

Thank you very much for reading our response soon, understanding our methodology and posting your second constructive comments. Following your comment, we will clearly describe the working hypothesis and its limitation for the estimation of gravity wave contribution to the residual mean circulation using reanalysis data.

Motivated by your comments posted as RC2, we were also thinking that it would be interesting to obtain the stream function using parameterized gravity wave drag provided

by each reanalysis and compare it with the results obtained by our method. We would be pleased to add the figures and discussion for the S-RIP community.

The description of our manuscript on the gravity wave parameterization used in each reanalysis was not correct as you indicated. ERA-Interim and JRA-55 use orographic gravity wave parameterization only, while MERRA and MERRA-2 use both orographic and non-orographic gravity wave parameterizations [Table 3 of Fujiwara et al. (ACP, 2017, doi:10.5194/acp-17-1417-2017]. Note that JRA-55 does not use non-orographic gravity wave parameterization, similar to ERA-Interim.

---

## Author Comment (AC5) · 10 May 2018

We greatly appreciate Referee #3's critical reading and valuable comments. We are afraid that Referee #3, like the other two reviewers, may misunderstand the methodology of our analysis. This misunderstanding simply comes from the insufficient description of our ACPD manuscript. We really apologize about it. We will seriously improve the methodology part of the manuscript. We would very much appreciate it if Referee #3 kindly reads our quick response to the other two reviewers (#AC1, the same as #AC2) in which we tried to clarify our methodology.

The reason why we used X_bar was described in #AC1 (and #AC2). We know that

[Figure]

X_bar in (1) of our manuscript includes several sources other than GWs for the re-analysis data. However, we have obtained very similar structure in the stream function corresponding to X_bar in all the reanalysis data. Thus, we think that the stream function corresponding to X_bar comes from real dynamics in the atmosphere. In addition, as discussed in Section 5, many of the characteristics found in the stream function corresponding to X_bar are consistent with the characteristics of gravity waves known from previous studies using high-resolution observations and gravity-wave permitting GCM simulations. We will revise our manuscript and clarify the working hypothesis and its limitation of our analysis, if we have the opportunity.

We are also sorry that the description of gravity wave parameterization in our manuscript was not correct. ERA-Interim and JRA-55 use orographic gravity wave parameterization only, while MERRA and MERRA-2 use both orographic and non-orographic gravity wave parameterizations [(Table 3 of Fujiwara et al. (ACP, 2017, doi:10.5194/acp-17-1417-2017)]. This point was also indicated by Dr. Abalos (Referee #2) and we plan to correct that description.

---

## Referee Comment (RC6) · M. Abalos (Referee) · 16 May 2018

Dear Dr. Kaoru Sato,

thank you again for your response, I am looking forward to reading the revised version of the paper.

I am somewhat confused by the fact that JRA-55 uses only an orographic GW parameterization. The variable 'Gravity wave zonal acceleration' provided for JRA-55 has a large (eastward) drag in the summer hemisphere (see Fig. 1 showing GWD in color and zonal wind in contours, easterlies dashed). This is not seen in ERA-Interim, con-

sistent with orographic gravity waves with near-zero phase speeds being filtered out by the summer stratospheric easterlies. Indeed the JRA-55 drag looks more consistent with that in MERRA, which includes non-orographic GWD. Perhaps I am missing something, but it might be helpful to shed light on this issue in the paper.

**GWD (m/s/day) 1979-2012 EI month 7**

**GWD (m/s/day) 1979-2012 JRA-55 month 7**

**GWD (m/s/day) 1979-2012 MERRA month 7**

**Fig. 1.**

---

## Author Comment (AC6) · 30 May 2018

We are sorry that we could not submit this response soon because the first author did not find time (due to conferences and teaching).

We describe the reasons why we consider that our method can serve as an estimation for the true gravity wave drag. First, the stream functions attributed to GW contribution that were obtained from the four reanalyses have very similar structure. Thus, we think that the stream function obtained by our method reflects at least real dynamics in the atmosphere. Second, as discussed in Section 5, many of the characteristics found in the stream function are consistent with the characteristics of gravity waves known from

previous studies using high-resolution observations and gravity-wave permitting GCM simulations. Third, the features observed in the "GW" stream function are consistent with the model study by Okamoto et al. (2011) referred to in AC1. As the reviewer said, it is true that increments include several sources other than GWs in the reanalysis, and we also understood it. Thus, we will carefully revise our manuscript and clarify the working hypothesis and its limitation of our analysis.

Abalos et al. (2015) showed that the spread among the nine estimates (three methods times three reanalysis data) of upward mass flux is large and about 40% for 70-10hPa (their Figure 6). However, they also noted that in general the differences are larger among the three estimates in each reanalysis than among the three reanalyses for each estimates. This is also clear in their Figure 6 above 70hPa. We consider that the difference between the direct estimate and the indirect estimate from the zonal momentum equation seen in Abalos et al's analysis is mainly due to a part of the GW drag that is not accurately expressed by the GW parameterizations. Note also that even in our analysis, as shown in Figures 10 and 11, the total upward mass fluxes obtained from directly estimated residual mean circulation using four reanalyses do not differ much. These facts suggest that the direct method using its definition gives a good estimation of w* (or v*).

In contrast, the contribution of GWs to the upward mass flux shown also in Figures 11 and 12 largely depends on the reanalysis data. However, it should be noted that this quantity is quite sensitive to a slight difference in the stream function structure because the upward mass flux is calculated only using stream function values at two particular latitudes (i.e., turn-around latitudes). This point will be clearly discussed in the revised manuscript so as to avoid unnecessary confusion, if we are given the opportunity.
* * *

---

## Author Comment (AC7) · 30 May 2018

Dear Dr. Abalos,

Thank you very much for your information on gravity wave drag in JRA-55.

I asked the JRA-55 project members about the gravity wave parameterization used in JRA-55 during the spring meeting of the Meteorological Society of Japan held last week. According to them, JRA-55 used an operational model for the weather prediction of Japan Meteorological Agency (JMA). Non-orographic gravity wave parameterizations were first implemented in the JMA operational model in March 2014, which was

after the completion of JRA-55 calculation. Thus, it is certain that the model used for JRA-55 does not include non-orographic gravity wave parameterizations.

I agree that the feature of gravity wave drag in the summer middle atmosphere in JRA-55 of your figure is inscrutable as orographic gravity wave drag. I will further examine this issue with JMA people, and describe it in the revised manuscript, if I have the opportunity.

Kaoru Sato

---

## Author Response (AR1)

**Response to Reviewer 1's comments**

The authors greatly thank Reviewer 1 for his/her careful reading of our manuscript and constructive comments and suggestions. Details of our response to respective comments are described below. The comments by the reviewer are printed in italic and our responses are in roman.

**Response to RC1**

Although we already replied to RC1 partly as AC2, we describe here in detail how the manuscript has been revised.

*Major comment:*

*My main concern is on how the contribution from Rossby waves and gravity waves to the Brewer-Dobson circulation is calculated. The decomposition was based on the following equation: Psi=Psi_RW + Psi_GW + Psi_dUdt (their equation 6), where Psi_RW and Psi_dUdt were estimated by integrating the momentum equation, Psi was estimated by integrating the TEM velocity, and Psi_GW was then estimated as a residual. This is valid in theory, but in practice, because reanalysis data is not fully consistent, there should be an additional residual error term on the right-hand side of the equation. As a result, the gravity wave contribution estimated in the paper includes both true gravity wave contribution and the residual errors as well.*

As the reviewer indicated, reanalysis data is not fully consistent because of the assimilation process. The method proposed in this study, however, does not assume that the reanalysis data satisfies the zonal momentum equation. We only use the theoretical fact that the residual flow can be decomposed into contributions by Rossby waves (RWs), gravity waves (GWs), and du/dt ignoring viscosity/friction. The assumption in this method is that the residual mean flow using its definition (a good approximate of the Lagrangian flow), EP flux divergence (i.e., RW forcing), and du/dt can be accurately obtained by the reanalysis data. A critical assumption among these is that ¥overline{w} is accurately obtained. We have revised in the 3rd and 4th paragraphs of Section 2 which explain the method of this study. The 5th paragraph of Section 2 and the 3rd paragraph of Section 7 have been added so as to clearly specify the assumptions.

The 2nd paragraph of Section 1 has been added to describe the recent development of the recent reanalysis data.

*Furthermore, based on the study by Abalos et al. (2015), a highly relevant study the authors seem to have missed, the residual errors dominate over the true gravity wave contribution. The authors claimed that "the gravity wave contributions can be estimated only indirectly", which is incorrect. Most modern reanalysis products do explicitly provide the parameterized gravity wave drag employed in their model. Therefore, one can directly estimate the gravity wave contribution to the circulation, which is done in Abalos et al. (2015). According to Abalos et al. (2015), the gravity wave contribution is substantially smaller than the resolved waves in all three reanalysis datasets they analyzed, which are also included in this study. In addition, Abalos et al. (2015) compared the different estimations of the Brewer-Dobson circulation: one based on integration of TEM velocity (equivalent to Psi in this paper), and one based on integration of the momentum equation (equivalent to Psi_RW+Psi_GW+Psi_dUdt here). They reported a larger difference between the two estimations than the contribution from parameterized gravity waves (Fig. 3 and Fig. 4 in Abalos et al.), and larger difference among estimation methods than among datasets. Comparing result shown in this paper with those in Abalos et al. (2015), it is clear that the "gravity wave contribution" estimated here is consistent with the difference between the two estimation methods in Abalos et al., indicating that most of the "gravity wave contribution" here is actually the residual errors.*

We thank the reviewer for letting us know our oversight of an important paper by Abalos et al. (2015). This paper has been introduced in the 4th paragraph of Section 1. As indicated by Geller et al. (2013), current GW parameterizations do not completely represent the GW forcing in the real atmosphere, and there are several serious discrepancies with high-resolution observations and numerical modellings. Therefore, the difference in the residual mean flow among the three methods shown by Abalos et al. (2015) can be due to such inadequacy of GW parameterizations. As a purpose of the present study is to examine the role of GW forcing in the real atmosphere, we

decided not to use the value of gravity wave parameterization, but we used an indirect method. This point has been clarified by adding the last three sentences in the 4$^{th}$ paragraph of Section 2.

*Other comments:*
*1. The author used the term "Rossby wave", but what they actually referred to is the resolved waves. It is true that in the extratropics, most resolved waves are indeed Rossby waves. But in the tropics, there are also Kelvin waves and other gravity waves that are large enough to be resolved.*

Indeed, there are planetary scale Kelvin waves and GWs in the equatorial region. However, since the present study treats the latitudes higher than 20 degrees for examination regarding the contribution of each type of waves to the residual mean circulation, most analyzed resolved waves can be regarded as RWs. The 2nd and 3rd sentences in Section 2 have been added.

*2. Page 8 Line 28-29: The claim about the usage of gravity wave parameterization in the reanalysis is incorrect. According to Seviour et al. (2011), ERA interim does not include non-orographic gravity wave drag. According to Gelaro et al. (2017), orographic gravity wave drag in included in MERRA2.*

We have correctly rewritten the 7th paragraph of Section 3.1.

**Response to RC4**
Although we already replied to RC4 partly as AC6, we concretely describe how the manuscript has been revised.

*I agree with the authors that the parameterized gravity wave drag are not the true gravity wave drag, and the difference between the parameterized and the true gravity wave drag contributes to the increment error. However, I still do not agree with the authors that their method can serve as an estimation for the true gravity wave drag. Basically, the authors are assuming all the terms except the gravity wave drag in the momentum equation can be perfectly accurately calculated from reanalysis data or small enough to be negligible, and therefore the residual term from the equation would be the gravity wave drag. This may be true in a model simulation where all variables are dynamically consistent with each other, which explains the results from Okamoto et al. (2011) the authors are referring to. But the assumption that all the other terms can be accurately calculated does NOT hold in the reanalysis products. Due to the assimilation process in the reanalysis data, nonphysical increments are introduced in all the variables. As also pointed out by Dr. Abalos, the other reviewer, this increment error does not only arise from gravity wave but also from many other processes as well. The residual term of the momentum equation therefore does not only consist of the true gravity wave drag, but also differences between the calculated and the true value in all the other terms. More importantly, the true gravity wave drag may not dominate in this residual term, so the residual term does not even give a bulk approximation for the gravity wave drag. Take the term (2) for example, if it can be accurately calculated from reanalysis, then one would expected that v\* calculated from its definition would be the same among different reanalysis products since they are representing the same real world. But as shown in this paper as well as in Abalos et al. (2015), this directly calculated TEM velocity does vary among reanalysis products, and the difference is    NOT small compared to the residual term or the so-called "true gravity wave drag".*

We agree that the three terms, namely, du/dt, EP flux divergence, and residual mean flow v\* (by its definition), would not be perfectly correctly estimated by reanalysis data. However, it is quite difficult to show how accurate the estimates are. In this study, we compare the results from the four reanalysis data and consider that if there are common features, they should show real physics. With regard to GWs, we consider that if there are consistent characteristics with the facts revealed by previous observations and GW-resolving numerical model simulations, the characteristics can be real. With such a way of thinking, we described the common features and consistent characteristics obtained from the reanalysis data. This point has been described in the 5th paragraph of Section 2, and the 3rd paragraph of Section 7. The last sentence of the 8th paragraph of Section 1 has also been added to mention that a potential error

coming from data assimilation is one more reason to use the term "potential" for the GW contribution.

*I also agree with Dr. Abalos that it would be helpful to compare the increment error explicitly, since it represents the accumulative errors from not only gravity waves but also resolved waves as well as mean circulation. While the parameterized gravity waves may not represent the true gravity wave effects in the real world, I think a comparison among reanalysis is still meaningful.*

Following the reviewer's comment, we have performed the comparison between the streamline function corresponding to the GW forcing expressed by GW parameterizations of reanalysis data and the potential GW contribution estimated by our method. Comparison with the stream function corresponding to the assimilation increment to the zonal mean zonal wind has also been performed for MERRA and MERRA 2. The results have been described in newly added Section 6. It is seen that the stream functions corresponding to the GW forcing described by GW parameterizations are largely different among the four reanalysis data, but the difference in the potential GW contribution estimated by our method among the reanalysis data is small. This result is quite encouraging, and suggests that data assimilation improves the residual mean flow field including that induced by GW forcing in the reanalysis data. The difference between the stream function corresponding to the parameterized GW forcing and the potential GW contribution suggests that in the current GW parameterizations, eastward GW forcing in the low latitude region and westward GW forcing in the winter high latitude region are too weak. These results suggest incomplete description of the GW sources in the GW parameterizations. Another possibility for the shortage of westward GW forcing in the winter high latitude region is the lack of horizontal propagation, which is important as suggested by recent studies, but not expressed in the GW parameterizations in the model used for reanalysis data. This point has also been described in the 8th paragraph of Section 7.

Regarding upward mass flux for 10 hPa (shown in Fig. 11), a robust result for the potential GW contribution was not obtained. This point has been clearly described and a more robust discussion on 10 hPa and 3 hPa based on the annual mean stream function shown in Fig. 4 and Fig. 5 has been added in the 6th, and 7th paragraphs of Section 4.

**Response to Reviewer 2 (Dr. M. Abalos) 's comments**

The authors greatly thank Dr. M. Abalos for her careful reading of our manuscript and constructive comments and suggestions. Details of our response to respective comments are described below. The comments by Dr. Abalos are printed in italic and our responses are in roman.

**Response to RC2**

Although we already replied to RC2 partly as AC3, we concretely describe how the manuscript has been revised.

*Major issues*

*- The paper methodology is at present based on the assumption that the difference between the RC computed from the TEM definition (Eqs. 3 and 4) and that estimated from momentum balance (i.e. downward control plus du/dt term) is attributed exclusively to the gravity wave (GW) drag parameterized in the reanalysis. However this assumption is not necessarily valid because it does not take into account that assimilation increments can play a key role in the momentum balance of the reanalyses.*

As the reviewer indicated, reanalysis data is not fully consistent because of the assimilation process. The method proposed in this study, however, does not assume that the reanalysis data satisfies the zonal momentum equation. We only use the theoretical fact that the residual flow can be decomposed into contributions by Rossby waves (RWs), gravity waves (GWs), and du/dt ignoring viscosity/friction. The assumption in this method is that the residual mean flow using its definition (a good approximate of the Lagrangian flow), EP flux divergence (i.e., RW forcing) and du/dt can be accurately obtained by the reanalysis data. A critical assumption among these is that ¥overline{w} is accurately obtained. We have revised in the 3rd and 4th paragraphs of Section 2 which explain the method of this study. The 5th paragraph of Section 2 and the 3rd paragraph of Section 7 have been added so as to clearly specify the assumptions.

*- In contrast with what is stated in the paper, GW drag is provided for all the reanalyses considered here and thus the GW contribution to the RC can be directly computed using Eq. 8.*

As indicated by Geller et al. (2013), current GW parameterizations do not completely represent the GW forcing in the real atmosphere, and there are several serious discrepancies with high-resolution observations and numerical modellings. As a purpose of the present study is to examine the role of GW forcing in the real atmosphere, we decided not to use the value of gravity wave parameterization, but we used an indirect method. This point has been clarified by adding the last three sentences in the 4[th] paragraph of Section 2.

*Also, all the reanalyses considered include orographic gravity wave parameterizations, and only ERA-Interim does not include non-orographic gravity waves. This information is found in the reanalysis description papers cited but is wrongly stated in the paper.*

We are sorry that the description was not accurate. We have correctly rewritten the 7th paragraph of Section 3.1. JRA-55 and ERA-Interim do not include non-orographic GW parameterizations, but instead they use Rayleigh friction which mimics non-orographic GW forcing. This fact has also been described.

*Suggestions*

*In my opinion the paper would notably improve and make a useful contribution to SRIP if the authors include an analysis of the GW drag (and perhaps the zonal wind assimilation increment) provided by the reanalyses.*
*This would allow direct evaluation of the contribution of the parameterized GW drag in the reanalysis models to the RC using Eq. 8, without need of the assumption pointed out in the first comment. This calculation was already done*

*in Abalos et al. (2015 JGR) for ERA-Interim, MERRA and JRA-55. However in that paper the GW contribution is not examined in detail for the different seasons and their analysis extends only to 10 hPa, so it will be interesting to present extended results here.*

We really acknowledge the reviewer's recognizing the advantage of this research. Following the reviewer's comment, we have performed the comparison between the streamline function corresponding to the GW forcing expressed by GW parameterizations of reanalysis data and the potential GW contribution estimated by our method. Comparison with the stream function corresponding to the assimilation increment to the zonal mean zonal wind has also been performed for MERRA and MERRA 2. The results have been described in newly added Section 6. It is seen that the stream functions corresponding to the GW forcing described by GW parameterizations are largely different among the four reanalysis data, but the difference in the potential GW contribution estimated by our method among the reanalysis data is small. This result is quite encouraging, and suggests that data assimilation improves the residual mean flow field including that induced by GW forcing in the reanalysis data. The difference between the stream function corresponding to the parameterized GW forcing and the potential GW contribution suggests that in the current GW parameterizations, eastward GW forcing in the low latitude region and westward GW forcing in the winter high latitude region are too weak. These results suggest incomplete description of the GW sources in the GW parameterizations. Another possibility for the shortage of westward GW forcing in the winter high latitude region is the lack of horizontal propagation, which is important as suggested by recent studies, but not expressed in the GW parameterizations in the model used for reanalysis data. This point has also been described in the 8th paragraph of Section 7.

*Moreover, analysis of the difference between the total RC computed by explicitly including the forcing by resolved and parameterized GW (Eq. 6) versus the RC computed from the TEM definition (Eqs. 3 and 4) will provide useful information on the momentum budget in the reanalyses. In particular, based on the results of Abalos et al. (2015 JGR) the two estimates of the RC are significantly different (even including the parameterized GW term). This could imply that the GW parameterizations in the reanalyses are insufficiently capturing the role of GW on the RC. In that sense it could be argued that most of the difference is attributed to the GW drag in the real atmosphere but absent in the parameterizations. This important point is not discussed in the paper, and as a result there is a confusion between the GW drag that is parameterized in the reanalyses and the real GW drag assumed to equal the residual of the momentum balance.*

We are very sorry that we do not notice an important paper by Abalos et al. (2015). This paper has been introduced in the 4th paragraph of Section 1. As already mentioned, Geller et al. (2013) indicated that current GW parameterizations do not completely represent the GW forcing in the real atmosphere, and there are several serious discrepancies with high-resolution observations and numerical modellings. Therefore, the difference in the residual mean flow among the three methods shown by Abalos et al. (2015) can be due to such inadequacy of GW parameterizations, as also mentioned by the reviewer. As a purpose of the present study is to examine the role of GW forcing in the real atmosphere, we decided not to use the value of gravity wave parameterization, but we used an indirect method. Following the reviewer's suggestion, this point has been clarified by adding the last three sentences in the 4th paragraph of Section 2.

*Explicitly computing the GW contribution and clearly explaining these issues would substantially strengthen the current discussion in the paper on the role of the different waves on the RC in reanalyses, and on the limitations of current reanalyses GW parameterizations. In addition, consideration of the assimilation increments provided by the reanalyses can help interpret the momentum balance and further clarify these issues.*

As already mentioned, additional analyses were made using parameterized GW forcing and increment for the zonal mean zonal wind. The results have been described and discussed in Section 6 following the reviewer's suggestion.

*Other general comments*

*- Literature citation: The previous studies Iwasaki et al. (2010 R. Met. Soc. Japan), Abalos et al. (2015 JGR) and Miyazaki et al. (2016 ACP) have already examined and compared the residual circulation in modern reanalyses and should be cited accordingly.*

The study by Abalos et al. (2015) has been described in the last half of the 4$^{th}$ paragraph of Section 1. The descriptions of Iwasaki et al. (2010) and Miyazaki et al. (2016) have been added in the 2nd paragraph of Section 1.

*- Acknowledgement of the reanalysis centers for providing the data should be included.*

The description for centers for providing the data has been added following the reviewer's comment.

*- I recommend carefully reading the draft before submitting the new version to improve the wording in several parts.*

The manuscript has been proofread by a specialized company.

*- I find interesting the analysis of the du/dt term contribution to the RC seasonality. This term is key for the subseasonal tropical upwelling variability (Abalos et al. 2014 JAS), consistent with the downward control principle (Haynes et al. 1991 JAS), but its role for the seasonal cycle is not fully understood. For instance Kim et al. (2016 JAS) argue that it is negligible for the seasonality in tropical upwelling.*

Abalos et al. (2014) and Kim et al. (2016) have been cited and a description on the du/dt term contribution has been added in 4th paragraph of Section 4. Discussion on Kim et al. (2016) has also been added as the 3rd paragraph of Section 4.

**Response to RC3**

Although we already replied to RC3 as AC4, we concretely describe how the manuscript has been revised here.

*I very much appreciate your clear response to my comment. I would like to clarify the key points of my review. I fully understand the methodology of the paper and I consider it valid, but I find it necessary to include a discussion regarding the role of assimilation increments in reanalyses, and I recommend including a comparison to the parameterized GWD from reanalyses, as explained below.*

*Indeed, Okamoto et al. (2011) showed that in a climate model the GWD equals the residual of the momentum balance using Terms 1, 2 and 3. However this is not the case in reanalyses because data assimilation produces an assimilation increment, i.e. an additional term in the momentum equation. The working hypothesis of the paper is that most of this assimilation increment is acting to correct for the limitations of GW parameterizations, and thus the residual of the momentum equation can be interpreted as the 'actual' GWD. While I consider this a valid hypothesis, I argue that it should be explicitly stated as such in the paper, because it is not self-evident. There could be model biases having little to do with 'actual' GWD (e.g. in radiative heating) that need to be offset by the data assimilation. Having this discussion in the paper would notably help the reader understand the reasoning behind the methodology.*

As already described in detail in our response to the reviewer's comments RC2, we have significantly revised Section 2. The 4th paragraph has been revised and the 3rd and 5th paragraphs have been added.

*In addition, in my review I suggest the authors to include an analysis of the parameterized GWD provided by the reanalysis centers. A comparison between the parameterized GWD and that 'estimated' from the balance Terms 1, 2, 3 would be very useful to highlight the limitations of parameterized GWD in reanalyses pointed out in Dr. Sato's*

*comment, especially in the context of an S-RIP paper (e.g. are the differences larger for ERA-Interim which does not have non-orographic GW parameterizations?). It seems to me that such comparison would be useful for the S-RIP community and that the present paper is an adequate place to discuss these issues.*

As already described in detail in our response to the reviewer's comments RC2, additional analyses were made using parameterized GW forcing and increment for the zonal mean zonal wind. The results have been described and discussed in Section 6 following the reviewer's suggestion. The results have also been discussed in the 8th paragraph of Section 7.

**Response to RC6**

*I am somewhat confused by the fact that JRA-55 uses only an orographic GW parameterization. The variable 'Gravity wave zonal acceleration' provided for JRA-55 has a large (eastward) drag in the summer hemisphere (see Fig. 1 showing GWD in color and zonal wind in contours, easterlies dashed). This is not seen in ERA-Interim, consistent with orographic gravity waves with near-zero phase speeds being filtered out by the summer stratospheric easterlies. Indeed the JRA-55 drag looks more consistent with that in MERRA, which includes non-orographic GWD. Perhaps I am missing something, but it might be helpful to shed light on this issue in the paper.*

We thank Dr. Abalos very much for her providing the figure showing the GW forcing represented by the GW parameterization for each reanalysis data. We have inquired the JRA-55 group and found that the "Gravity wave zonal acceleration" includes both parameterized GW forcing and Rayleigh friction. We can understand the features seen in Fig.1 if the Rayleigh friction is included. This issue has been described in the 7th paragraph of Section 3.1.

**Response to Reviewer 3's comments**

The authors greatly thank Reviewer 3 for his/her careful reading of our manuscript and constructive comments and suggestions. Details of our response to respective comments are described below. The comments by the reviewer are printed in italic and our responses are in roman.

**Response to RC5**

Although we already replied to RC5 as AC5, we concretely describe how the manuscript has been revised here.

*Major Comments:*

*In the present study, two assumptions were made: (1) stream function calculated using Eqs. (3)-(4), so-called direct stream function (Psi), and using Eq. (5) based on downward-control principle (Psi_DC) is exactly the same. (2) stream function of Psi_DC induced by the residual term X_bar in the TEM equation represents GW contribution. Based on these two assumptions, Eq. (10) is derived. Followings are comments on the two assumptions.*

*1)Although the stream function Psi and Psi_DC should be equal theoretically, it is not exactly the same, likely because the governing equations used and physical processes in the GCM of each reanalysis data set are somehow different from rather simple TEM equation. Accordingly, the mass flux calculated from the two stream functions are somewhat different from each other as shown in some previous studies. Note that this is different from the case of recent work by Abalos et al. (2015) where Psi_DC is calculated using GWD rather than X_bar, which is not in momentum balance of TEM equation, and their comparison between Pai and Psi_DC stems mostly from difference between X_bar and GWD. It is curious for the reviewer why authors use Eq. (10) in calculation of stream function of GW rather than Eq. (8).*

Current GW parameterizations do not completely represent the GW forcing in the real atmosphere. According to Geller et al. (2013), at least spatial distribution of the parameterized GWs has serious discrepancies with the high-resolution observations and numerical modellings. As a purpose of the present study is to examine the role of GW forcing in the real atmosphere, we decided not to use the value of gravity wave parameterization, but we used an indirect method. This point has been clarified by adding the last three sentences in the 4th paragraph of Section 2. The difference in the residual mean flow among the three methods shown by Abalos et al. (2015) can be due to such inadequacy of GW parameterizations. The work by Abalos et al. (2016) has been cited and discussed in the 4th paragraph of Section 1.

*2)The major benefit of Psi_DC is to calculate the contribution of resolved planetary waves (EPD), du_bar/dt, and non-conservative term (represented by X_bar ) separately. The term X_bar can be calculated from any reanalysis data set as a residual of the TEM equation. The reviewer cannot understand why the authors state " Psi_GW cannot be directly calculated because of the unknown X_bar " (Page 6, line 6). The term represents implicitly the parameterized GWD, numerical diffusion, and assimilation increment. In most recent reanalysis data sets that provide GWD variables, the magnitude of GWD is much smaller than that of X_bar. Therefore, even when GWD variables are provided from reanalysis data sets, quite large value of the residual term, say X'_bar, after excluding GWD, is required for momentum balance in the TEM equation. Therefore, the stream function calculated using Eq. (10) of the current study is not from GWD but from X_bar, which include several sources other than GWs, in particular, assimilation increment.*

As the reviewer indicated, reanalysis data is not fully consistent because of the assimilation process. The method proposed in this study, however, does not assume that the reanalysis data satisfies the zonal momentum equation. We only use the theoretical fact that the residual flow can be decomposed into contributions by Rossby waves (RWs), gravity waves (GWs), and du/dt ignoring viscosity/friction. The assumption in this method is that the residual mean flow using its definition (a good approximate of the Lagrangian flow), EP flux divergence (i.e., RW forcing) and

du/dt can be accurately obtained by the reanalysis data. A critical assumption among these is that ¥overline{w} is accurately obtained. We hypothesize that the assimilation increment acts to correct the limitations of the GW parameterization. We have revised in the 3rd and 4th paragraphs of Section 2 which explain the method of this study. The 5th paragraph of Section 2 and the 3rd paragraph of Section 7 have been added so as to clearly specify the assumptions. In addition, so as to make the description of methodology more clearly, the 3rd and 4th paragraphs of Section 2 have been revised significantly.

*3)Note that the GWD variables provided from reanalysis data are purely model output, without data assimilation, and thus high degree of uncertainties may exist. In addition, large values of X'_bar from assimilation increment may also include some parts of un-parameterized GWD, if there are. However, assimilation increment stems from various, probably all, processes in the model, including underestimation of resolved wave forcing (EDP), not exclusively from GWD. Therefore, it is not acceptable that stream function calculated using Eq. (10) of the current manuscript represent the stream function from GWs.*

As already mentioned, it is likely that parameterized GW forcing does not perfectly show the real GW forcing (Geller et al., 2013). Thus the potential GW contribution to the stream function of the residual mean circulation cannot be obtained by only using parameterized GW forcing. Thus we used an indirect method using Eq. (11) in the revised manuscript under the working hypothesis as already described. In general, it is quite difficult to show how accurate the estimates are. In this study, we compare the results from the four reanalysis data and consider that if there are common features, they should show real physics. With regard to GWs, we consider that if there are consistent characteristics with the facts revealed by previous observations and GW-resolving numerical model simulations, the characteristics can be real. With such a way of thinking, we described the common features and consistent characteristics obtained from the reanalysis data. This point has been described in the 5th paragraph of Section 2, and the 3rd paragraph of Section 7. The last sentence of the 8th paragraph of Section 1 has also been added to mention that a potential error coming from data assimilation is one more reason to use the term "potential" for the GW contribution.

We have made an analysis of parameterized GW forcing for the four reanalysis data and assimilation increment for MERRA and MERRA2, following the other reviewers' comments. The indirectly estimated potential GW contribution has been compared with the stream functions corresponding to the parameterized GW forcing and increment. In newly added Section 6, we have described this analysis and discussed the limitations of the current GW parameterizations based on the result. It is seen that the stream functions corresponding to the GW forcing described by GW parameterizations are largely different among the four reanalysis data, but the difference in the potential GW contribution estimated by our method among the reanalysis data is small. This result is quite encouraging, and suggests that data assimilation improves the residual mean flow field including that induced by GW forcing in the reanalysis data. The difference between the stream function corresponding to the parameterized GW forcing and the potential GW contribution suggests that in the current GW parameterizations, eastward GW forcing in the low latitude region and westward GW forcing in the winter high latitude region are too weak. These results suggest incomplete description of the GW sources in the GW parameterizations. Another possibility for the shortage of westward GW forcing in the winter high latitude region is the lack of horizontal propagation, which is important as suggested by recent studies, but not expressed in the GW parameterizations in the model used for reanalysis data. This point has also been described in the 8th paragraph of Section 7.

*Minor comments: 1)GWD variables are available from all reanalysis data set used in the present study, although non-orographic GWD output is not available from ERAInterim. This is not correctly mentioned in the manuscript.*

We appreciate the reviewer's indication. The 7th paragraph of Section 3.1 has been revised.

**Response to Drs. Šácha's and Pišoft comments**

The authors greatly thank Drs. P. Šácha and P. Pišoft for their careful reading of our manuscript and constructive comments and suggestions. Details of our response to respective comments are described below. The comments by Drs. Šácha and Pišoft are printed in italic and our responses are in roman.

**Response to RC1**

Although we already replied to RC1 as AC1, we concretely describe how the manuscript has been revised here.

*Your argument against the compensation mechanism in reanalyses (P2L30) appears to be speculative and it does not take into account various processes that can stand behind the compensation. For example, Haynes et al. (1991) noted that the DC principle applies to the zonally symmetric forcing, as the longitude-dependent force could set-up a Rossby wave field. This was demonstrated in a modeling study by Šácha et al. (2016) together with the effect of different zonal distribution of forcing on the residual circulation. There is an inherent zonal asymmetry in the gravity wave drag distribution (concentration into hotspots -e.g. Hoffmann et al., 2013; Šácha et al., 2015), which is reflected also in the parameterizations (at least orographic GW parameterizations, Šácha et al., 2018).*

I agree with the indication by Drs. Šácha and Pišoft. Many previous papers indicate the zonal asymmetry of GW distribution originating from the GW source and GW filtering in the large-scale flow modified by RWs. Such asymmetry generates and/or modulates the RW fields and subsequently has an impact on the BDC. I added the 4th sentence from the bottom of the 5th paragraph of Section 1.

*The compensation hardly makes it possible to clearly separate the effects of resolved and unresolved waves. This is an important point and we think that it has to be properly discussed in your paper.*

Since large scale waves such as RWs can be captured by observations, RWs are likely expressed realistically in the reanalysis data through assimilation of the observation data. This is different from the case of the model projection discussed by Cohen et al. (2013) and subsequent studies. The last sentence of the 5th paragraph of Section 1 has been revised.

*We also report on a typo (P15L16), where you probably wanted to relate the boundary condition (wstar = 0) to the turn-around latitudes.*

We appreciate the indication of our typo. The boundary condition has been properly described in the last sentence of the 1st paragraph of Appendix A.

*Besides that, we have general doubts regarding the conclusions of Appendix A, as we can see similar inequality between the wstar and vstar based method for residual mean streamfunction computation in a model reaching up to 150 km (i.e. including the wave forcing in the mesosphere, not published yet). We would recommend checking if the net tropical upwelling across a particular level inferred from wstar integration nears zero.*

This is interesting information and needs to be examined in future studies in detail. The last three sentences have been added in the last paragraph of Appendix A. The analysis on $\overline{w}^*$ was made and we confirmed the global mean of $\overline{w}^*$ is almost zero as was already shown in AC1.

**A list of all relevant changes made in the manuscript**

**Section 1**

· Previous studies on Brewer Dobson circulation have been cited in the 2nd paragraph of Section 1 (Responding to Reviewer #2's comment).

· An important paper by Abalos et al. (2015) has been cited and discussed, and the deficiency of current GW parameterizations indicated by Geller et al. (2013) has been stated in the 4th paragraph of Section 1 (Reviewer #1, #2, and #3).

· Additional mechanism of the RW and BDC modulation by GWs has been described in the 5th paragraph of Section 1 (Drs. Šácha's and Pišoft).

· Since large-scale waves such as RWs can be captured by observations, RWs are likely expressed realistically in the reanalysis data through assimilation of the observation data. The last sentence of the 5th paragraph of Section 1 has been revised to clarify this point (Drs. Šácha's and Pišoft).

· Additional reason to use the term "potential" has been added as the last sentence of the 8th paragraph of Section 1 (Reviewer #1 and #3).

· The reason why we used an indirect method to estimate the potential GW contribution to BDC has been described in the 10th paragraph of Section 1 (Reviewer #1, #2, and #3).

· As Section 6, in which comparison between the indirectly estimated GW contribution and the stream functions due to parameterized GW forcing and assimilation increment is made, has been added following the reviewers' comments, the last paragraph of Section 1 describing the structure of this manuscript has been rewritten.

**Section 2**

· The reason why all resolved waves can be regarded as RWs in the analysis of this study has been described as the 2nd and 3rd sentences in the 1st paragraph of Section 2 (Reviewer #1).

· The 3rd paragraph of section 2 has been added to explain how GW forcing in the zonal mean zonal momentum equation is included in the reanalysis data (Reviewer #1, #2, and #3).

· The 4th paragraph of Section 2 has been significantly revised so as to clarify the method used in this study (Reviewer #1, #2, and #3).

· The 5th paragraph of Section 2 has been added to clarify the assumption used in the method (Reviewer #1, #2, and #3).

**Section 3**

· The 7th paragraph of Section 3.1 has been corrected (Reviewer #1, #2, and #3)

**Section 4**

· A phrase "and because of the limitations of data assimilation," has been added to the last sentence of the 1st paragraph of Section 4 to make consistent discussion through the manuscript revised following the reviewers.

· Discussion on the seasonality of the upward mass flux has been added by referring to previous related studies in the 3rd and 4th paragraphs of Section 4 (Reviewer #2).

· To clarify the similarity and difference of the results among the four reanalysis data, the 2nd and last sentences of the 6th paragraph, and the 7th paragraph of Section 6 have been added. This revision is important to make consistent discussion through the manuscript revised following the reviewers.

**Section 6**

· Section 6 has been newly included in the revised manuscript (Reviewer #1, #2 and #3). In this section, results of the analysis for the stream function corresponding to the parameterized GW forcing and that corresponding to assimilation increment has been added. Comparing the results with the indirectly estimated potential GW contribution, potential deficiency of the current GW parameterizations has been discussed.

**Section 7**

- We have added the 3rd paragraph of Section 7 and described the assumption of the method used in this study and how we indirectly confirmed the validity of the assumption.
- The summary of results from Section 6 has been given in the 8th paragraph of Section 7.

**Appendix A**
- Responding to Drs. Šácha's and Pišoft's comment, the last 3 sentences have been added.

**Figures**
- Figures 13–15 needed for Section 6 have been added.

[revised manuscript text omitted]

---

## Referee Report (RR1)

**Comments on "The climatology of Brewer-Dobson circulation and the contribution of gravity waves" by Sato and Hirano**

This study presents seasonal variations of Brewer-Dobson circulation in terms of stream function and upward mass flux for 30 years (1986-2015), using four recent reanalysis data sets (MERRA, MERRA2, ERA-Interim, and JRA55). Special emphasis is given to the contribution of gravity waves on the stream function and upward mass flux in the whole stratosphere, and their seasonal and height dependency. Compared with the original manuscript, the current version is improved by including additional discussions on the logistics of the authors' approach and several new figures. Nevertheless, some major questions raised from the reviewer still remain in the current manuscript and some statements, especially related to the limitations in GWD parameterization in GCMs and their conjunction to the assimilation increment, are highly problematic, which could be misleading readers. Therefore, the review would like for the authors to address following comments properly before the current manuscript to be accepted to ACP.

**Major Comments:**

**1. Logistics in $\Psi_{GW}$ calculation**

In the present study, $\Psi_{GW}$ is calculated using Eq. (11). As the reviewer understand, first the stream function $\Psi_{dir}$ is calculated using Eqs. (3)-(5), which is conventionally called as a direct stream function. Then, based on the downward-control principle, stream function by planetary waves ($\Psi_{DC\_RW}$) and zonal-mean zonal wind tendency ($\Psi_{DC\_du/dt}$) is calculated. Then, $\Psi_{GW}$ is estimated as a difference between $\Psi_{dir}$ and sum of $\Psi_{DC\_RW}$ and $\Psi_{DC\_du/dt}$. The reviewer still cannot understand why this rather odd approach is needed, which may include some additional uncertainties.

1) Although the stream function $\Psi_{dir}$ and $\Psi_{DC}$ should be equal theoretically, it is not exactly the same, as shown from many previous studies, likely because the governing equations used and physical processes in the GCM of each reanalysis data set are somehow different from rather simple TEM equation.

2) If we agree with that $\Psi_{dir}$ and $\Psi_{DC}$ are exactly the same, as the authors assumed, then $\Psi_{GW}$ estimated based on Eq. (11) is the same as $\Psi_{DC\_GWF}$ using Eq. (8), if $\overline{GWF} + \overline{X}$ is considered as GW forcing. However, in Line 10-11 of Page 7 of the manuscript, the authors mention that "$\Psi_{GW}$ *cannot be directly calculated because of unknown* $\overline{GWF}$." This statement makes the reviewer be confused. Note that when

$\Psi_{GW}$ is estimated using Eq. (11), TEM equation is no need for being used. This is not a matter of whether GW forcing is represented by either parameterized GWF provided from the reanalysis data or the residual of the first four terms in the TEM equation ($\overline{GWF} + \overline{X}$). The $\Psi_{GW}$ is better to be calculated using the TEM equation to assure the momentum balance.

**2. Sources of $\overline{X}$**

1) One of main assumptions of the current study is that the grid-resolved planetary waves and zonal-mean zonal wind tendency are accurate by assimilation process (Abstract: Line 11-12), and $\overline{X}$ represents assimilation increment due to the limitation in GW parameterization. This is too optimistic, given that there are many factors for the resolved meteorological variables to be biased from the observed variables at each time step, which is represented by assimilation increment from the reanalysis data. As shown in the current results, there are quite significant differences in the planetary wave forcing and resultant stream function among the reanalyses used. It may be almost impossible to directly separate out the resolved part and parameterization part of the assimilation increment.

2) Accordingly, some statements regarding this issue should be modified.
  A. Line 12-13, Page 16: "*The difference between the upper and lower panels suggests the deficiency of the GW parameterizations*"
  B. Line 13-15, Page 16: "*It is encouraging that similarity among the four reanalyses is higher for $\Psi_{GW}$ than for $\Psi_{pGW}$. ... This suggests that current assimilation schemes act to make the GW contribution to the stream function realistic*". The relatively similar $\Psi_{GW}$ among the reanalyses than $\Psi_{pGW}$ is due to the assimilation increment in general, to make the model results to be better compared with the observation, not necessarily for fix the parameterized GWD.
  C. Line 26-27, Page 16: "*This result suggests that net non-orographic GW forcing is more strongly eastward in the real atmosphere than given by parameterizations*"

3) There is one paper that may need to be included in the current manuscript, which is similar objective, although using a single reanalysis data set (MERRA), which calculated stream function by EPD, $\overline{GWF}$, and $\overline{X}$ separately (Kim et al. 2014).

Interestingly, the stream function by $\overline{X}$ is larger than that by $\overline{GWF}$, but the mass flux (Fig. 6 of Kim et al. 2014) that is calculated using the stream function at turn-around latitude is smaller than that by $\overline{GWF}$.

**3. Contribution of GWs to the mass flux**

In the present study, the contribution of GWs to the mass flux is up to 40%, although it is different from each reanalysis data set, as mentioned at Line 22-24, Page 14: "*the GWs to the mass flux is ~20% at 70 hPa for MERRA and MERRA-2 at the most, while it is ~35–40% for ERA-Interim and JRA-55*". It should be noted that most GCMs already overestimate parameterized GWF through the tuning process, in order to compensate extremely underestimated planetary wave forcing in the model (Geller et al. 2013; Kim and Chun 2015; Kang et al. 2018). Therefore, contribution of GWs to the mass flux, which is calculated using $\overline{GWF} + \overline{X}$ in the present study might be strongly overestimated.

**4. Limitation in GW parameterization**

The authors estimate GW forcing by $\overline{GWF} + \overline{X}$ in the TEM equation, although their calculation method is not directly from TEM, as mentioned in the comment #1. As the reviewer understand, the major reason for not using $\overline{GWF}$, which could be provided from reanalysis data sets, as the GW forcing is likely due to that the authors consider that there is a significant limitation in GW parameterization used in GCMs. The limitation of GW parameterization is likely based on some previous studies using satellite data, such as Geller et al. (2013), where GW momentum flux estimated from HIRDLS is compared with GW parameterization from GCMs (and some resolved from relatively high-resolution GCMs). This comparison, however, is not very meaningful, because satellite observed GWs with horizontal wavelengths about 500-1000 km (Kalish et al. 2016) and parameterized subgrid-scale GWs with horizontal wavelength mostly less than 100 km are in significantly different scales. This has been discussed in depth in recent work by Kang et al. (2018). It is true that some non-orographic GW parameterizations, especially for those not physically formulated and source-dependent, have high degree of uncertainties in the tuning process, mostly with too strong source magnitude in order to compensate highly underestimated planetary wave forcing (Kim and Chun 2015). The real problem in the GW parameterization is in fact that it is strongly overestimated, rather than underestimated. Therefore, the stream function and resultant mass flux in the present study that are based on $\overline{GWF} + \overline{X}$ are even more overestimated compared with real GW contribution.

**5. Sensitivity of mass flux to the turn-around latitude**

In the mass flux calculation, the results should be very sensitive to the choice of turn-around latitude, which is different for the stream function derived from different wave forcings (Figs. 3-5), although the turn-around latitude from $\Psi_{dir}$ is used for the calculation shown in Figs. (10-12). The sensitivity of the mass flux calculation to the turn-around altitude should be included.

**Minor comments:**

1) Line 1-2, Page 16: GWs in the convective region has a clear annual variation, which is shown recently by Kang et al. (2017, JAS). Accordingly, the statement should be modified.

2) Line 26-27, Page 17 (Fig. 15): The way to calculate $\Psi_{INC}$ needs to be explained.

3) Line 30-31, Page 17: "*These features are consistent with the difference between* $\Psi_{pGW}$ *and* $\Psi_{GW}$, *and likely indicate a shortage of eastward GW forcing at the low latitudes*". The first part of the statement is correct, because your $\Psi_{GW}$ is simply the sum of $\Psi_{pGW}$ and $\Psi_{INC}$ (by $\overline{GWF} + \overline{X}$), but the last part of the statement is not correct, again, given that the assimilation increment is not solely by GW parameterization but include all part of model deficiency.

4) Line 3-4, Page 17: "*The maximum around 60°S in* $\Psi_{GW}$ *is consistent with observations and GW-resolving GCM simulations (Sato et al., 2009; Geller et al, 2013).*" The GWs considered in the TEM equation is the small-scale GWs that are not resolved from GCMs but are parameterized. The GWs estimated from satellite observations have horizontal wavelengths longer than 500 km, and those calculated from high resolution GCMs with horizontal grid spacing of 0.25º x 0.25º are longer than 200 km. If these relatively long GWs observed from satellite and resolved from GCMs are matched with the $\Psi_{GW}$, this is more likely related to $\Psi_{INC}$ causes by grid-scale GWs rather than small-scale parameterized GWs.

5) Figure 10: The reason not to calculate JJ in JRA-55 is better to be included in the figure caption.

**References**

Kalisch, S., H.-Y. Chun, M. Ern, P. Preusse, Q. T. Trinh, S. D. Eckermann, and M. Riese, 2016: Comparison of simulated and observed convective gravity waves. *J. Geophys. Res. Atmos*., 121, 13 474–13 492.

Kang, M.-J., H.-Y. Chun, and Y.-H. Kim, 2017: Momentum flux of convective gravity waves derived from an offline gravity wave parameterization. Part I: Spatiotemporal variations at source level. *J. Atmos. Sci*., 74, 3167–3189.

Kang, M.-J., H.-Y. Chun, Y.-H. Kim, P. Preusse, and M. Ern, 2018: Momentum flux of convective gravity waves derived from an offline gravity wave parameterization. Part II: Impacts on the Quasi-Biennial Oscillation (QBO). *J. Atmos. Sci*. (in press)

Kim, J.-Y., H.-Y. Chun, and M.-J. Kang, 2014: Changes in the Brewer-Dobson Circulation for 1980-2009 revealed in MERRA reanalysis data. *Asia-Pac. J. Atmos. Sci*., 50(S), 73-92.

Kim, Y.-H., and H.-Y. Chun, 2015: Momentum forcing of the quasi-biennial oscillation by equatorial waves in recent reanalyses. *Atmos. Chem. Phys*., 15, 6577–6587.

---

## Author Response (AR2)

**Response to Reviewer 1's comments**

The authors greatly thank Reviewer 1 for his/her careful reading of our manuscript and constructive comments and suggestions. Details of our response to respective comments are described below. The comments by the reviewer are printed by blue and our responses by black.

The authors argued that the gravity wave contribution can be inferred from the residual for two reasons: similarity among reanalysis data and consistency with high resolution observations and simulations. In this regard, the equatorward circulation driven by gravity waves in the low latitudes is a less robust result, as not all reanalyses data show this feature, which is only seen in the upper stratosphere in ERA-i, and not obvious in JRA-55.
The previous revised manuscript may have emphasized the difference too much. So the 1st and 2nd sentences of the 5th paragraph of section 3.1 have been revised. The equatorward circulation is not clear in JRA-55 but it exists at slightly lower latitudes than 20°. See the right figure which is the same as Fig. 2l but for the latitude range extending to 17°. The 4th sentence of the same paragraph has been added.

[Figure]

Also, is there any theoretical arguments or observational evidence to support that the gravity waves lead to an eastward forcing in the low latitudes?
We had already cited a few papers. To make this clearer, we added a phrase "as shown theoretically and numerically by previous studies" in the 2nd sentence of the 4th paragraph of section 3.1.

Editorial comments:
P1L8: modern four -> four modern
Revised. (The 1st sentence of abstract)

P3L28: I am not sure what you mean by "real middle atmosphere"
The word "real" has been removed. (The 3rd sentence of the 5th paragraph of section 1)

P5L25: "inertia-gravity waves" The resolved non-Rossby wave in the tropics are mainly Kelvin waves and mixed Rossby-gravity waves.
Planetary-scale inertia-gravity waves are observed in the equatorial region, too (e.g., Wada et al., 1999). The 2nd sentence of the 4th paragraph of section 2 has been revised.
· Wada, K., et al., (1999), Equatorial inertia-gravity waves in the lower stratosphere revealed by TOGA-COARE IOP data, *J. Meteor. Soc. Japan,* **77,** 721-736.

P8L5: Eqs 7 and 9 -> Eqs. 7 and 10
Revised. (The 1st sentence of the 5th last paragraph of section 2)

P9L3: "grid interval" -> output grid
Revised. (The 2nd sentence of the last paragraph of section 2)

P9L14: spring -> boreal spring
We have not revised because "spring" here indicates "spring" in the both hemispheres. (The 4th sentence of the 1st paragraph of section 3)

[Figure]

MERRA−2

P10 L18-24: The authors argue the higher top leads to the difference between MERRA/MERRA-2 and ERAi/JRA-55. One can easily verify this point by calculate the fraction of contribution by EP flux divergence above the ERA-i/JRA-55 top in MERRA/MERRA-2.

We appreciate this comment. We had already made this analysis and confirmed this before submission of the original manuscript. The right figure shows $\Psi_{RW}$ calculated without using data above 1 hPa for MERRA-2. The depth of the circulation is similar to those for ERA-Interim and JRA-55 shown in Figure 2. We have added the 2nd last sentence to the 6th paragraph of section 3.1

P13L21: I don't understand why stronger wave forcings is related to the more distorted structure in Psi_{dU/dt}

To explain the reason, the 3rd sentence of the 2nd last paragraph of section 3.3 "The wave forcing is not simply balanced with the Coriolis force but partly accelerates the mean zonal wind in equinoctial seasons when the steady state assumption does not hold (e.g., Garcia, 1987: Hayashi and Sato, 2018)." has been added.

Revised. (The 2nd sentence of the 3rd paragraph of section 4)

We have not revised because the expression is correct (i.e., it is not "on the seasonal time scales"). (The last sentence of the 4th paragraph of section 4)

**Response to Reviewer 3's comments**

The authors greatly thank Reviewer 3 for his/her careful reading of our manuscript and constructive comments and suggestions. Details of our response to respective comments are described below. The comments by the reviewer are printed by blue and our responses by black.

**Major Comments:**
**1. Logistics in $\Psi_{GW}$ calculation**
In the present study, $\Psi_{GW}$ is calculated using Eq. (11). As the reviewer understand, first the stream function $\Psi_{DIR}$ is calculated using Eqs. (3)-(5), which is conventionally called as a direct stream function. Then, based on the downward-control principle, stream function by planetary waves ($\Psi_{DC\_RW}$) and zonal-mean zonal wind tendency ($\Psi_{DC\_du/dt}$) is calculated. Then, $\Psi_{GW}$ is estimated as a difference between $\Psi_{DIR}$ and sum of $\Psi_{DC\_RW}$ and $\Psi_{DC\_du/dt}$. The reviewer still cannot understand why this rather odd approach is needed, which may include some additional uncertainties.
We simply integrated each term of the zonal mean zonal momentum equation in the vertical (e.g., (7) to (10)). We are afraid that the reviewer misread our manuscript. In our understanding, the downward control principle indicates that the wave forcing tends to control the circulation downward. This is exactly the case for steady state. So, it may not be correct that the contribution by the du/dt term to the stream function is described using the terminology of the downward control principle as the reviewer made.

1) Although the stream function $\Psi_{DIR}$ and $\Psi_{DC}$ should be equal theoretically, it is not exactly the same, as shown from many previous studies, likely because the governing equations used and physical processes in the GCM of each reanalysis data set are somehow different from rather simple TEM equation.
Of course, the governing equations used in GCMs are not the TEM equation. However, as many previous studies discussed, the dynamics in the middle atmosphere can be well described by primitive equations and the momentum budget can be well examined and interpreted using the TEM equations. Thus, we used the TEM equation based on the primitive equations for our analysis. To make this clearer, the order of paragraphs in section 2 has been changed with some needed modification: First, a theoretical description has been made (i.e., the 1st to 3rd paragraphs). Second, a description of the analysis performed in this study has been made (i.e., the 4th and later paragraphs).

2) If we agree with that $\Psi_{DIR}$ and $\Psi_{DC}$ are exactly the same, as the authors assumed, then $\Psi_{GW}$ estimated based on Eq. (11) is the same as $\Psi_{DC\_GWF}$ using Eq. (8), if *GWF+X* is considered as GW forcing. However, in Line 10-11 of Page 7 of the manuscript, the authors mention that "$\Psi_{GW}$ *cannot be directly calculated because of unknown GWF.*" This statement makes the reviewer be confused. Note that when $\Psi_{GW}$ is estimated using Eq. (11), TEM equation is no need for being used. This is not a matter of whether GW forcing is represented by either parameterized GWF provided from the reanalysis data or the residual of the first four terms in the TEM equation (*GWF+X*). The $\Psi_{GW}$ is better to be calculated using the TEM equation to assure the momentum balance.
The GW forcing is not *directly* calculated because the parameterized GW forcing is not accurate. However, the GW forcing can be estimated *indirectly* using Eq. (11). Equation (11) is exactly the vertical integration of the zonal mean zonal momentum TEM equation ignoring the term $\overline{X}$. To clarify this, a phrase "which is derived from Eq. (1)" has been added in the last sentence of the 5th paragraph of section 2.

**2. Sources of $X$**
1) One of main assumptions of the current study is that the grid-resolved planetary waves and zonal-mean zonal wind tendency are accurate by assimilation process (Abstract: Line 11-12), and $X$ represents assimilation increment due to the limitation in GW parameterization. This is too optimistic, given that there are many factors for the resolved meteorological variables to be biased from the observed variables at each time step, which is represented by assimilation increment from the reanalysis data. As shown in the current results, there are quite significant differences in the planetary wave forcing and resultant stream function among the reanalyses used. It may be almost impossible to directly separate out the resolved part and parameterization part of the assimilation increment.
The reviewer is misreading our manuscript. Eq. (1) is a theoretical equation, and the term $\overline{X}$ in Eq. (1) is not assimilation increment but friction and/or viscosity. To make it clearer, the order of first several paragraphs in section 2 has been changed as already mentioned. We hope that this revision reduces the possibility of misreading. In addition, we wanted to show similarities of the stream function due to resolved waves, which are not only the planetary waves, among the reanalysis data, and next we tried to discuss slight but interesting differences. To clarify this, the 1st and 2nd sentences of the 5th paragraph of section 3.1 and the 1st and 2nd sentences of the 6th paragraph of section 3.1

have been revised.

We do not conclude that the difference between $\Psi_{pGW}$ and $\Psi_{GW}$ is only due to the deficiency of the GW parameterizations. What we can obtain from the comparison between $\Psi_{pGW}$ and $\Psi_{GW}$ is suggestion regarding deficiency of the parameterizations. This is a reason why we used "suggests" in this sentence. But taking into account the impression of the reviewer that this expression is too strong, we have used the words "may suggest" instead of "suggests" in the 3rd sentence of the 2nd paragraph of section 6.

It is true that the intention of data assimilation is not to fix the parameterized GWD but to make the model results to be better compared with the observation. But, the similarity among the four reanalysis data suggests that the current assimilation acts to make the GW contribution to the stream function realistic, as a result. The 2nd last sentence of the 2nd paragraph of section 6 have been added and the last sentence of the same paragraph has been revised.

The expression of "suggests" has been revised to "may suggest". (The last sentence of the 4th paragraph of section 6)

The *GWF* and *X* analyzed by Kim et al. (2014) are parameterized GWF and the residual of the momentum balance among the directly-estimated residual mean flow, EP flux divergence, and parameterized GW forcing. Thus, while the notation is similar, the terms examined in their study are different from those in our study. Therefore, direct comparison is not possible. Quoting this paper may cause unnecessary misleading to readers. So, we did not refer to this paper.

**3. Contribution of GWs to the mass flux**

We do not agree with the reviewer's comment that most GCMs already overestimate parameterized GWs through the tuning process, in order to compensate extremely underestimated planetary wave forcing in the model. Geller et al (2013) did not discuss about this (Note that KS, one of the authors of the current manuscript, is an author of Geller et al.). Kim and Chun (2015) and Kang et al. (2018) examine the QBO forcing in the equatorial region, in which dominant dynamics is different from that in the middle and high latitudes. Our response to this comment is included in that for the next comment.

**4. Limitation in GW parameterization**

HIRDLS is compared with GW parameterization from GCMs (and some resolved from relatively high-resolution GCMs). This comparison, however, is not very meaningful, because satellite observed GWs with horizontal wavelengths about 500-1000 km (Kalish et al. 2016) and parameterized subgrid-scale GWs with horizontal wavelength mostly less than 100 km are in significantly different scales. This has been discussed in depth in recent work by Kang et al. (2018). It is true that some non-orographic GW parameterizations, especially for those not physically formulated and source-dependent, have high degree of uncertainties in the tuning process, mostly with too strong source magnitude in order to compensate highly underestimated planetary wave forcing (Kim and Chun 2015). The real problem in the GW parameterization is in fact that it is strongly overestimated, rather than underestimated. Therefore, the stream function and resultant mass flux in the present study that are based on $GWF+X$ are even more overestimated compared with real GW contribution.

As already mentioned, the reviewer misunderstood the meaning of $GWF$ and $X$ in Eq. (1). The reviewer considers that $GWF$ is parameterized wave forcing and $X$ is increment. However, in Eq. (1), $GWF$ is subgrid-scale phenomena and $X$ is viscosity and/or friction. In order to avoid such unnecessary misunderstanding, the order of the first several paragraphs of section 2 has been revised as is already mentioned. The border of resolved scales and subgrid scales depends on the grid spacing of the data. In addition, the reviewer considers subgrid scales only in terms of the horizontal wavelength. However, even large horizontal-scale GWs can be subgrid scales if they have small vertical wavelengths that are not resolvable in the coarse vertical grid. Four sentences have been added in the last part of the 5th paragraph of section 6.

Kang et al (2018) and Kim and Chun (2015) examine wave forcings driving the QBO in the equatorial region, which is different from the latitude region examined by the present study. Kim and Chun (2015) discussed that smaller amplitudes of planetary-scale equatorial waves such as Kelvin waves in the reanalysis data than satellite observations, which was indicated by Ern et al. (2008), are (at least partly) attributable to coarse vertical resolution of the GCM of the reanalysis. This is because the coarse vertical grid cannot resolve smaller wavelengths of the waves in the strong vertical shear of the QBO. We do not think that this result for the equatorial region indicates unrealistically weak amplitudes of planetary waves in middle and high latitudes in the reanalysis data. In the equatorial region, the Coriolis force is quite weak, and hence winds are not well balanced by well-observed quantities such as temperature. We suppose that this is another possible reason of underestimation of Kelvin wave amplitudes in the reanalysis data. In contrast, resolved waves in the middle and high latitudes, for which our analysis was performed are mainly (well balanced) Rossby waves and hence it is expected that they are well constrained by satellite observations. In fact, Kawatani et al. (2016) showed large spread of the zonal wind in the recent reanalyses but mainly in the equatorial region. The reviewer seems confusing at this point. The last four sentences of the 7th paragraph of section 2 has been added for this discussion.

- Ern, M., Preusse, P., Krebsbach, M., Mlynczak, M. G., and Russell III, J. M.: Equatorial wave analysis from SABER and ECMWF temperatures, Atmos. Chem. Phys., 8, 845-869, https://doi.org/10.5194/acp-8-845-2008, 2008.
- Kawatani, Y., Hamilton, K., Miyazaki, K., Fujiwara, M., and Anstey, J. A.: Representation of the tropical stratospheric zonal wind in global atmospheric reanalyses, Atmos. Chem. Phys., 16, 6681-6699, https://doi.org/10.5194/acp-16-6681-2016, 2016.

**5. Sensitivity of mass flux to the turn-around latitude**
In the mass flux calculation, the results should be very sensitive to the choice of turn-around latitude, which is different for the stream function derived from different wave forcings (Figs. 3-5), although the turn-around latitude from $\Psi_{DIR}$ is used for the calculation shown in Figs. (10-12). The sensitivity of the mass flux calculation to the turn-around altitude should be included.

The stream function due to the GW contribution does not show a simple one-celled circulation for each hemisphere. The mass flux is not easily estimated from the stream function due to GW contributions because the stream function is missing in the low latitude region. Thus, we defined the GW and RW contributions as those obtained taking the turn-around latitude of directly-estimated total stream function in Eqs. (12)-(14). The 2nd last sentence of 2nd last paragraph of section 2 "In our study, the turn-around latitude used for calculation of each wave contribution is taken the same used for the total upward mass flux." has been added. The last sentence of 2nd last paragraph of section 4 "Note also that the potential GW contribution is sensitive to the location of the turn-around latitude." has been added following the reviewer's suggestion.

**Minor comments:**
1) Line 1-2, Page 16: GWs in the convective region has a clear annual variation, which is shown recently by Kang et al. (2017, JAS). Accordingly, the statement should be modified.
A phrase "although clear seasonal variation is seen at the cloud top level in the troposphere (Sato et al., 2009; Kang

et al., 2017)" has been added in the 1st sentence of the last paragraph of section 5 following the reviewer's suggestion.

2) Line 26-27, Page 17 (Fig. 15): The way to calculate $\Psi_{INC}$ needs to be explained.
Eq. (16) has been added in the 2nd last paragraph of section 6.

3) Line 30-31, Page 17: "*These features are consistent with the difference between $\Psi_{pGW}$ and $\Psi_{GW}$, and likely indicate a shortage of eastward GW forcing at the low latitudes*". The first part of the statement is correct, because your $\Psi_{GW}$ is simply the sum of $\Psi_{pGW}$ and $\Psi_{INC}$ (by *GWF+X*), but the last part of the statement is not correct, again, given that the assimilation increment is not solely by GW parameterization but include all part of model deficiency.
As the reanalysis data does not satisfy the momentum equations through the assimilation processes as other reviewers indicated. So, our $\Psi_{GW}$ is not necessary the same as the sum of $\Psi_{pGW}$ and $\Psi_{INC}$. However, the last part of the statement (the last sentence of the 2nd last paragraph of section 6) may be too strong, we have weakened the expression by replacing "likely indicate" with "suggest".

4) Line 3-4, Page 17: "*The maximum around 60°S in $\Psi_{GW}$ is consistent with observations and GW-resolving GCM simulations (Sato et al., 2009; Geller et al, 2013).*" The GWs considered in the TEM equation is the small-scale GWs that are not resolved from GCMs but are parameterized. The GWs estimated from satellite observations have horizontal wavelengths longer than 500 km, and those calculated from high resolution GCMs with horizontal grid spacing of 0.25° x 0.25° are longer than 200 km. If these relatively long GWs observed from satellite and resolved from GCMs are matched with the $\Psi_{GW}$, this is more likely related to $\Psi_{INC}$ causes by grid-scale GWs rather than small-scale parameterized GWs.
The reviewer's comment "The GWs considered in the TEM equation is the small-scale GWs that are not resolved from GCMs but are parameterized." is not correct. The forcings by all hydrostatic waves including both Rossby waves and gravity waves are expressed with EP flux divergence in the TEM equation based on the primitive equation system. The border wavelength between unresolved or resolved waves is determined by the grid spacing of the data. So as to clarify the treatment of the theoretical TEM equation for the analysis of data with different resolutions, the order of the first several paragraphs in section 2 has been changed with needed modification, again as mentioned above.
    The minimum resolved wavelengths for 1.2 degree or 1.25 degree output grid of the reanalysis data used in our study are about 500 km (4 grid points). Note that this is optimistic estimation and actual wavelength that can be examined by this grid spacing would be longer than 500km. Geller et al. (2013) including the results from the model used in Sato et al. (2009) treated GWs whose horizontal wavelength typically shorter than 1000 km. Thus, we can compare our result of $\Psi_{GW}$ with the results of Geller et al. (2013) and Sato et al (2009). The original model resolution for the reanalysis is higher than the output grid spacing used in the present study. So, one may consider that the gravity waves with moderately long wavelengths may be resolvable. However, even such GWs may not be fully resolved because of coarse vertical resolution of the model. These gravity waves can be subgrid "vertical" scale phenomena and should be included in the parameterization. In fact, $\Psi_{INC}$ is reflected by phenomena unresolved in the model, and we consider that a plausible candidate is unresolved GWs. The four sentences have been added in the last part of the 5th paragraph of section 6.

5) Figure 10: The reason not to calculate JJ in JRA-55 is better to be included in the figure caption.
The caption has been revised following the suggestion.

[revised manuscript text omitted]

---

## Author Response (AR3)

We greatly appreciate careful and critical reading and valuable comments by the reviewer and editor and concrete suggestions by the editor regarding the reviewer's comments.

My interpretation of these points raised by the referee are as follows. In cases (1) and (2) I have proposed some straightforward (and, I think, justifiable) changes.

(1) The GW contribution as defined is actually a contribution from 'unresolved' waves and the 'resolved' waves may include a contribution from GWs. The referee suggests that the resolved waves are divided into RW and GW contributions on the basis of zonal wavenumber. A simply alternative alternative would be for you to make it absolutely clear at key points in the paper (abstract, introductory sections, conclusions) that your distinction is between resolved and unresolved waves and that the resolved waves may contain some gravity waves.

Following the suggestion, the 2nd sentence in Abstract (ll.8-9, p.1), the 4th sentence in the 3rd paragraph of Section 2 (ll.5-7, p.7), and the 3rd sentence in Section 7 (ll.23-24, p.19) have been added.

(2) The residual in the momentum equation (and the corresponding contribution to the mass stream function) cannot be reliably interpreted as the GW (or unresolved wave) contribution since the assimilation increment, which contributes to the residual, is influenced by many different aspects of model error against observation. Your argument against that is that many of the characteristics of the GW contribution that you identify are common to all four reanalysis datasets. The differences between the datasets are quite difficult to identify from figures such as Figure 2. Quantitative comparison is easier in Figures 3-5, where it is clear that estimates of the GW contribution differ by up to a factor of 2. Therefore, again at important points of the paper, please emphasise that whilst there is qualitative agreement between the results from the different reanalyses, there is significant quantitative uncertainty (as illustrated by Figure 3-5). Your current statement on l28 p19 'First the results from the four reanalysis [datasets] were quite similar except for minor features' -- for example -- seems much too strong. (A factor of 2 disagreement is not minor.)

We agree to the editor that this statement is too strong. The quantitative difference in the GW potential contribution is large for 70 hPa but not for 10 hPa. We consider that the 70 hPa level is close to the tropopause whose expression may depend on the model. However, the detailed examination of this point is beyond the scope of our study. The last sentence of the last paragraph of Section 3.1 (l.17, p.12) and the 3rd last sentence of the 3rd paragraph of Section 7 (l.13, p.20) have been revised.

(3) I'm not completely clear on this point. Certainly Geller et al (2013) seem to imply that GM momentum flux is significantly overestimated by parametrizations compared to observations -- 'The faster fall off with height of the gravity wave momentum fluxes derived from satellite measurements than in models is the most severe disagreement between measured and model fluxes shown in this paper.' -- though you have argued against this in a previous response. (Perhaps you were arguing specifically against the suggestion that the disagreement was caused by 'tuning' of the parametrisation?) But it is not clear to me why that implies you are overestimating the GW contribution --

unless the inclusion of the GW parametrization in the model has an effect on the assimilation as a whole (more than simply changing the assimilation increment -- e.g. if it somehow affected the estimate of the RW contribution). Please consider once again whether this general point requires any change -- e.g. addition of a sentence indicating uncertainty.

Descriptions regarding the discussion on GW parameterizations given in Geller et al. (2013) have been added in the newly-added 5th paragraph of Section 1 (ll.10-24, p.3).

Given that I am going to accept the paper in a situation where a referee still has significant doubts, I have looked again myself at the paper as a whole and have noted the points below. Please make changes to address these (there may be other typographical errors that I haven't spotted) -- or provide brief justification why no change is needed.

p1 l8: 'data' > 'datasets'

Thank you for this indication. Revision of similar usage of "data" has been made in other parts of the manuscript.

p3 l21: 'hardly' doesn't make sense -- 'poorly'?

We have revised the expression following the suggestion (l.1, p.4).

p4 l21: 'Randel (2008), as described above' -- wasn't clear to me what your 'as described above' referred to -- be more specific. I think that this is your 'second' method on p3.

The 3rd last sentence in the 5th last paragraph has been added (ll.1-3, p.5).

p4 l30: Your use of 'potential' here represents an important definition (which would not be clear, for example, to someone reading the abstract -- who would not know that 'potential' was being used in a certain sense).

We understand this comment. However, we think the word "potential" is a suitable word to express this term. There was another phrase using "potential" in the abstract, which is in the last sentence describing the deficiencies of current GW parameterizations (i.e., "potential deficiencies …"). We have replaced this with "plausible" in the revised manuscript.

p4 l32: This justification of a broader use of 'potential' is very confusing. The first use is required because of dynamics (the response to a force is both an acceleration and a meridional circulation). This is completely distinct from uncertainty due to mismatch between model and data.

We agree. The sentence "Another reason to use the term "potential" is that the limited performance of data assimilation may cause contamination of estimates of the GW contributions." has been removed (l.14, p.5).

p7 l14: 'function by' > 'function driven by'

The word "driven" has been added (l.7, p.8).

p8 l8: 'However, the similarity among Psi_GW (phi,z) estimated ... from the four reanalysis datasets ... MUST show real dynamics' -- it is this sort of statement that I think continues to trouble the referee -- their comment (2). I could, imagine, for example, that the time and space sampling of the data used (which is presumably pretty much the same for each of the re-analyses) relative to the high (and regular) space-time resolution of each of the models being used for the reanalysis leads to a significant contribution to the data assimilation increment that is common to all the models.

We understand that this expression is too strong. We consider that current GW parameterizations are not perfect and have significant deficiency as described in the (newly included) 5th paragraph of Section 1 referring to Geller et al. (2013). Thus, a part of increment of the reanalysis reflects the parameterization deficiency. We think that this part may be quite large. There is a missing wavenumber range which can be resolved by the models with the high (and regular) space-time resolution but cannot be seen in the sampled spatial grids of the reanalysis datasets. In our method, this missing wavenumber range is also included for the analysis. The word "must" has been replaced with "may" (l.21, p.8). The 2nd sentence in Abstract (ll.8-9, p.1), the 4th sentence in the 3rd paragraph of Section 2 (ll.5-7, p.7), and the 3rd sentence in Section 7 (ll.23-24, p.19) have been added.

p8 l31: 'the zonal mean zonal wind tendency term ... is evaluated as the radiation effect' -- personally I'd question whether this term can reliably be associated with radiation alone, but if you want to make this working assumption then it is important that your introduction of the term 'radiation effect' is presented to the reader here as a definition -- and that definition is properly referred to later -- e.g. on p19 l19 you use the term 'RW and radiation contributions' and the reader needs to be reminded that you have a particular justification for using the term 'radiation contribution'. Again -- personally if I were writing your paper I would continue use 'pa u/pa t contribution', because that relates directly to what you have calculated, rather than requiring an extra assumption about the physics.

The expression of "radiation contribution" has been revised following the suggestion (ll.13-14, p.9; ll.14-15, p.14; ll.22-23, p.19).

p9 l16-18: You should give more details on the levels in the models. (These details are surely as important as those on horizontal resolution, which you do give.)

The 2nd last sentence has been added in the last paragraph of Section 2 (ll.2-3, p.10).

p17 l28: Geller et al 2013 is cited here, but as far as I can tell is not included in the reference list.

Thank you for the indication. It seems that we misused Mendeley to make the reference list. The reference is included manually.

Appendix A: This doesn't necessarily require action on your part, but I find some of your statements in this Appendix puzzling. You are noting that using a vertical integration of vbarstar vs a horizontal integration of wbarstar to deduce Psi can give different answers. You are taking vbarstar and wbarstar from the re-analysis dataset. To me the difference between the two calculations potentially arises from a combination of non-zero divergence of the (vbar,wbar) field, or from inconsistency between the numerical integration/differentiation that you are using in your calculation vs that which is used in the reanalysis calculations. I don't see how you draw any conclusions about e.g. the role of wave forcing in the stratosphere/mesosphere (l25-29 on p21) or with the existence of GW forcing in the mesosphere (l30 p21 - l3 p22).

Thank you for your interest in this part. We have revised the 3rd and 4th paragraphs of Appendix A (ll.8-20, p.22).

We have not replied to the following comments by the reviewer directly because we have replied to these comments indirectly through the revision following the editor's comments as described above. We really appreciate the reviewer's comments and the editor's treatment.

Comments on "The climatology of Brewer-Dobson circulation and the contribution of gravity waves" by Sato and Hirano

The reviewer appreciates the authors' efforts on improving the manuscript during several revision processes. The reviewer could understand better in the current manuscript what authors want to say. Nevertheless, there are some points, mainly underlain assumptions of the manuscript, which are still unacceptable to the reviewer. Considering the impacts of the current results to potential readers, the reviewer would like for the following points to be clarified before the manuscript is accepted to ACP.

1.        In the TEM equation (1), the EDP represents forcing by the all "resolved waves" which includes both planetary waves and resolved GWs (mainly inertial GWs": IGWs). This is not only for high resolution GCMs, but the ones used in the current study with horizontal resolution of about 1 degree in latitude/longitude. The only difference is that horizontal scales of IGWs resolvable can be smaller for high resolution GCMs. That is, the EPD calculated in the present study should not be solely from Rossby waves. Accordingly, GWF should be from the "parameterized subgrid-scale" GWs, which cannot be resolved from data grids. Theoretically, sources of GWs for any GWD parameterization should be less than the model grid spacing, as done in orographic GWD and convective GWD schemes. Therefore, if model resolution is relatively coarse (that is, resolved GWs are less abundant), the parameterized subgrid-scale GWF should be larger (based mostly on tuning), in general, in order for the momentum balance.

If authors really want to put all scales of GWs, including both resolved GWs and parameterized GWs, to the potential GWF, the EDP should be calculated separately based on the zonal wavenumber (e.g., say k < 20 are Rossby waves and k > 20 are resolved GWs, where k is the zonal wavenumber), and then sum of the EPD by the resolved GWs and parameterized GWD can be a total GWF.

2.      In the present study, GWF is estimated from the residual of the TEM equation, which is similar to some previous studies before the parameterized GWD output was provided from the reanalysis data sets. The residual term is similar to the sum of the parameterized GWD and assimilation increment. The assimilation increment is simply a model error against observation, and there is no way to isolate individual process to induce the error. In order for the resolved grid values of wind and temperature to be similar among the reanalysis data sets, as in the current study represented by EPD comparison, there might be a significant assimilation increment for each data sets (as shown from MERRA and MERRA-2 in Fig. 15), which should be different from each data sets that were produced using different GCM models. Note again that the assimilation increment cannot be solely by the uncertainty of the GWD parameterization of the model. Therefore, the residual of the TEM equation cannot be considered as potential GWF.

3.      The GWD output from each reanalysis data set, represented by pGW forcing in the current manuscript, is purely model output, because there are no observational data to assimilate it. In current GCMs, parameterized GW momentum flux is significantly overestimated in the middle atmosphere, as shown from Geller et al. (2013, JCL) in the stratosphere (at z = 40 km, Figs. 4 & 5 of Geller et al., 2013). Considering this situation, contribution of GWs to the BDC suggested from the current study, which is estimated by the residual of the TEM equation (pGW forcing + assimilation increment), is highly overestimated.

[revised manuscript text omitted]